# A spatiotemporal transcriptomic atlas of porcine (*Sus scrofa*) female early gonadal development
Pengcheng He[1,3], Wenzhe Xia[1,3], Tianzhi Chen [1,3], Yaxuan Yan[1,3], Dengfeng Gao[2,3], Yadi Teng[1], Ting Zhao[1], Xinze Chen[1], Zhiqiang Feng[1], Runbo Li[1], Meng Wang[1], Yuwen Ke[1] ✉ & Jianyong Han [1] ✉

The early gonads of mammals contain primordial germ cells (PGCs) and gonadal somatic cells. In females, the gonadal somatic cells promote the specification of PGCs into oogonia and their entry into meiosis, which is crucial for oogenesis. Although single-cell transcriptome sequencing technology holds significant advantages in cell type identification, its inability to resolve spatial positional information substantially hampers research on communication mechanisms between germ cells and adjacent somatic cells. Here, we utilized high-resolution spatial transcriptomic technology to dissect the spatial dynamics of various cell types during the specification of PGCs into oogonia and their entry into meiosis in porcine gonads. We clarified the spatial localization of two waves of granulosa cells in the supporting cell lineage and their roles in regulating germ cell development. Furthermore, we found that interstitial and endothelial cells were predominantly located in the medullary region of the early gonads. Notably, cell-cell communication analysis revealed the critical role of BMP signaling (BMP2, BMP4 and GDF5) in driving the specification of PGCs into oogonia and their subsequent entry into meiosis. Our study provides a spatially resolved understanding of the PGC-to-oogonia specification in vivo and offers critical resources for reconstituting oogenesis in vitro.

In female mammals, the germ cell lineage originates from primordial germ cells (PGCs) that undergo a multi-step differentiation process to form oocytes[1]. Elucidating the molecular mechanisms governing PGC specification into oogonia and the initiation of meiosis under physiological conditions is critical for the faithful reconstitution of gametogenesis in vitro[2]. Comparative analyses have highlighted significant interspecies differences in cellular composition, molecular signatures, and developmental trajectories, particularly between rodent models and humans[3,4]. Pigs have emerged as a superior model for studying human development and regenerative medicine due to their conserved embryonic development and shared epigenetic dynamics during germ cell specification[5–8]. Here, we utilized pigs as a large-animal model to investigate the progression from PGC specification through oogonia differentiation to meiotic initiation.

Porcine PGCs (pPGCs) exhibit remarkable evolutionary conservation with human PGCs (hPGCs), making pigs a valuable model for uncovering fundamental mechanisms of human germ cell development[5,9]. In hPGCs specification, *SOX17* and *BLIMP1* play dual roles in establishing germline identity while repressing somatic programs—unlike in mice, where *SOX17* is dispensable[10]. In pigs, PGC specification occurs between embryonic days 12–14 (E12–E14), driven by BMP signaling and the sequential upregulation of *SOX17* and *BLIMP1*[11,12]. Following specification, pPGCs initiate hindgut migration at E15 and colonize the genital ridges by E22. Between E28 and E42, these cells express oogonia markers such as *DAZL* and *DDX4* and undergo extensive proliferation before entering meiosis at E45[13]. Early epigenetic reprogramming in pre-migratory pPGCs is characterized by global DNA demethylation and reduced H3K9me2 levels[5,9]. Upon gonadal colonization, pPGCs establish functional niches through interactions with somatic cells, coordinating their developmental progression.

During the process of PGC specification into oogonia, gonadal somatic cells play a key role. In mouse models, the origin of gonadal somatic cells has been clearly elucidated, where the gonadal primordium first appears on the coelomic surface of the mesonephros at around E10. The coelomic epithelium is a source of supporting cell precursors for the male and female gonads. Granulosa cells are derived from supporting cell precursors in

[1]Frontiers Science Center for Molecular Design Breeding (MOE), State Key Laboratory of Animal Biotech Breeding, College of Biological Sciences, China Agricultural University, Beijing, China. [2]State Key Laboratory of Swine and Poultry Breeding Industry, College of Animal Science and Technology, Sichuan Agricultural University, Chengdu, China. [3]These authors contributed equally: Pengcheng He, Wenzhe Xia, Tianzhi Chen, Yaxuan Yan, Dengfeng Gao. ✉e-mail: keyw@cau.edu.cn; hanjy@cau.edu.cn

undifferentiated genital ridges[14]. Female gonadal development involves coordinated activation of the Rspo1/Wnt4/β-catenin signaling axis and *Foxl2* expression to drive granulosa cell commitment[15]. In females, granulosa cells promote the specification of PGCs into oogonia and directly initiate meiosis[16]. This underscores the critical role of the germ cell microenvironment in fate determination.

Single-cell omics technologies have well characterized the transcriptomic profiles of distinct cell types during gonadal development[17]; however, the spatial localization features and developmental trajectories of these cellular populations remain to be elucidated. Here, we employed high-resolution spatial RNA sequencing (RNA-seq) technology to map the dynamic changes in cellular populations during gonadal development[18,19]. We delineated the migration trajectory of PGCs during their specification into oogonia and entry into meiotic initiation, while simultaneously defining the spatial signatures of distinct supporting cell types and further elucidating their significant roles in regulating germ cell development through key molecular mechanisms. Additionally, we found that mesenchymal cells are predominantly accumulated at the gonad-mesonephros interface. Concurrently, intercellular communication has identified BMP signaling as playing a critical role in both oogonia specification and meiotic initiation. This study provides a theoretical foundation for in vitro reconstitution of oogenesis in humans and large animal models, thereby advancing progress in reproductive.

## Results

### Decoding a near single-cell resolution spatial transcriptomic atlas of porcine female gonadal development

To elucidate the spatiotemporal transcriptional dynamics of female gonadal development, we performed high-definition spatial RNA sequencing (hdST-seq) at a resolution of 10 μm to establish a high-resolution transcriptional atlas across five critical embryonic stages (E24, E27, E30, E35, and E50)[17,18] (Fig. 1a). Biological replicates (2–3 tissue sections per stage) were processed to control biological variability while preserving cellular heterogeneity. Following stringent quality control, 50,630 high-quality spatial pixels were retained for constructing the porcine female gonadal spatiotemporal transcriptomic atlas (Fig. 1a and Supplementary Data 1). We applied uniform manifold approximation and projection (UMAP) to visualize the distribution of cells within gonadal samples and projected the data back into spatial coordinates by mapping cell identities onto a spatial grid, preserving barcode-based positional information for visualization (Fig. 1b, c and Supplementary Fig. 1a–d).

To investigate cell types and spatial information within the gonads, we systematically classified cellular populations into eight distinct subclusters based on previously reported marker genes in mouse, pig, and human[3,5,20] (Fig. 1d, e). Specifically, cluster 8 corresponds to female fetal germ cells (FGCs) expressing canonical markers (*KIT, BLIMP1* and *DDX4*). During gonadal development from E24 to E50, FGCs continue to proliferate and migrate to accumulate in the gonadal cortical region (Fig. 1d, e and Supplementary Fig. 1e, f). This cortical migratory pattern mirrors observations in human fetal gonad, suggesting that this aspect of germ cell behavior might be conserved between pigs and humans[3,21]. Cluster 1, characterized by ovarian surface epithelium cells (OSECs) markers (*LHX9, KRT19*), was predominantly localized to the outer cortical region throughout development[4,22] (Fig. 1d and Supplementary Fig. 1e). Spatial visualization revealed a distinct stratified organization in the gonads at E24–E27. Specifically, the OSECs occupied the outer cortical region, with cluster 3 adjacent to it medially, and cluster 2 localized to the medullary region (Fig. 1e and Supplementary Fig. 1f). Subsequent characterization identified cluster 2 as Pregranulosa cells-I (PreGCs-I) through *WNT6* expression, while cluster 3 was classified PreGCs-II based on *TOX3* expression[23–25] (Fig. 1b–d and Supplementary Fig. 1e). Together, three heterogeneous cell types, OSECs, PreGCs-I, and PreGCs-II, were identified and arranged in a stratified organization extending from the outer cortical region to the inner central zone, where they predominantly reside. As development proceeds, PreGCs-I gradually translocate to the medullary region.

In early gonads (E24 and E27), endothelial cells (ECs), interstitial cells (ICs) and mesenchymal cells (MCs) were predominantly located opposite to the regions containing the OSECs, PreGCs-I and PreGCs-II. Clusters 4 and 5 both expressed the interstitial cells marker *NR2F1* and *PDGFRA*, while cluster 7 were identified as endothelial cells due to *PECAM1* and *COL15A1* expression[20,26] (Fig. 1d and Supplementary Fig. 1e). During early gonad development (E24 and E27), interstitial cells comprised only a minor fraction of the total cell population (Fig. 1c). However, as development progressed to the E50 stage, their abundance increased markedly, and they became primarily localized to the medullary region (Fig. 1e and Supplementary Fig. 1f). Meanwhile, mesenchymal cells expressing *ARX* and *NR2F2* were scarcely detectable in the gonad during the later developmental stages (E30–E50)[26] (Fig. 1d, e and Supplementary Fig. 1e, f).

In summary, we constructed the high-resolution spatiotemporal transcriptomic atlas of porcine early gonads and elucidated the spatial distribution patterns of distinct cell types.

### Unveiling spatial patterns and molecular shifts during porcine PGC-to-Oogonia development

To characterize the transcriptional dynamics underlying the specification of PGCs into Oogonia, we extracted FGCs and performed UMAP-based clustering. Three distinct FGC subpopulations were identified: Mitotic PGCs, Retinoic Acid-responsive Oogonia (RA Oogonia), and Meiotic Oogonia (Fig. 2a, b). Transcriptional analysis revealed that Mitotic PGCs retained pluripotency markers (*OCT4, NANOG*) alongside core PGC markers (*TFAP2C, BLIMP1*)[11] (Fig. 2c). RA Oogonia exhibited upregulation of oogonia markers (*DAZL, DDX4*) and RA pathway components (*ZGLP1, STRA8*)[27] (Fig. 2c). Meiotic Oogonia were distinguished by meiotic gene expression, including *SYCP1, DMC1,* and *SPO11*[20] (Fig. 2c). Stage-specific immunofluorescence staining of BLIMP1 (early PGC marker) and DAZL (oogonia marker) confirmed these transitions: BLIMP1+/DAZL− at E24, BLIMP1High/DAZLLow at E30, BLIMP1Low/DAZLHigh at E35, and BLIMP1−/DAZL+ at E50 (Fig. 2d). We also reconstructed the differentiation trajectory of FGCs including Mitotic PGCs, RA Oogonia, and Meiotic Oogonia and computed pseudotime using Monocle 3 (Supplementary Fig. 2a). These results indicated that the developmental window spanning stages E24–E50 encompasses the specification process from PGCs to oogonia. Differential expression analysis identified 1335, 199, and 1456 genes uniquely enriched in each cluster, respectively (Supplementary Fig. 2b and Supplementary Data 2). Gene Ontology (GO) enrichment confirmed Meiotic Oogonia cluster specificity in "meiotic nuclear division" and "meiotic cell cycle processes"[28] (Supplementary Fig. 2c and Supplementary Data 2). To evaluate meiotic initiation during gonadal development, we next performed immunofluorescence at E35 and E50 for the oogonia marker DAZL as well as the leptotene and zygotene stage marker γH2AX, respectively[29]. E35 DAZL+ cells lacked γH2AX signal, while robust expression was evident in E50 DAZL+ cells (Fig. 2e). Together, our results demonstrate that E24-E50 encompasses FGCs undergoing three stages: PGCs colonization of gonads, their specification into oogonia, and the initiation of meiotic entry.

Having defined the transcriptional landscape of FGC specification, we next sought to characterize the accompanying epigenetic reprogramming events. Epigenetic reprogramming is integral to germ cell ontogeny, encompassing DNA demethylation, histone modification dynamics, and chromatin remodeling[30–32]. After understanding the transcriptional changes, we then explored the epigenetic aspects. Active demethylation involves ten-eleven translocation enzymes (*TET1, TET2, TET3*) that oxidize 5 mC to 5-hydroxymethylcytosine (5hmC), which are often coupled with thymine-DNA glycosylase (TDG)—mediated base excision repair[33]. Immunofluorescence staining revealed the absence of 5 mC and 5 hmC in germ cells but high expression of 5 mC and 5 hmC in surrounding somatic cells at E24, E30, and E50, consistent with previous reports[5] (Supplementary Fig. 2d, e). Alongside low 5mC and 5hmC levels, DNA methyltransferase 3A (*DNMT3A*), *TET1* and *TET3* expression declined, while the base excision repair (BER) pathway was upregulated, suggesting that active DNA demethylation had been completed, yet epigenetic reprogramming seemed

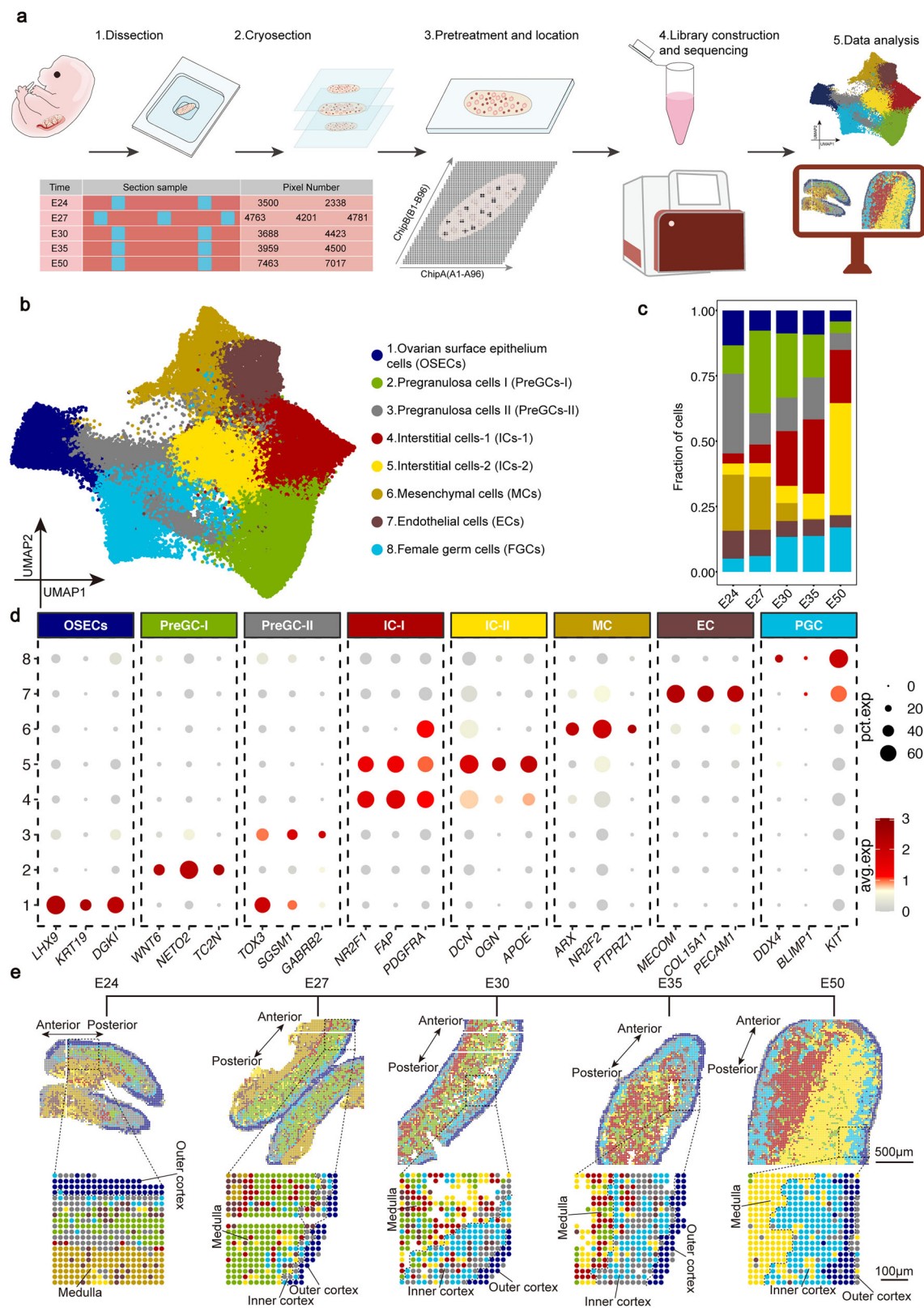

to persisted via the BER pathway[5,34–36] (Supplementary Fig. 2f and Supplementary Data 3). Interestingly, the trimethylation of histone H3 at lysine 27 (H3K27me3), a repressive histone modification, produced a stronger signal in Mitotic PGCs but sharply declined by E30, becoming undetectable by E35 and E50 (Supplementary Fig. 3a). The core components of the Polycomb Repressive Complex 2 (PRC2) members *EZH2, SUZ12*, and *EED* are highly expressed in early gonadal pPGCs[37]. PRC2 catalyzes H3K27me3 and plays a key role in gene expression regulation[33] (Supplementary Fig. 3b and Supplementary Data 3). Furthermore, during the differentiation of PGCs into oogonia, the expression of subunits *SMARCA5* and *HLTF* from the SWI/SNF chromatin remodeling complex, which functionally antagonizes H3K27me3-mediated regulation, is gradually upregulated[5,38]

**Fig. 1 | High-resolution spatial transcriptomics atlas of porcine female gonads.**
**a** Schematic diagram of experimental workflow: Gonadal samples were collected at embryonic days 24, 27, 30, 35, and 50 (E24–E50) to construct a spatiotemporal molecular atlas of female porcine gonad development. **b** UMAP visualization of single-cell clustering showing distinct cell populations of germ and somatic cells in developing female porcine gonads. **c** Bar plot showing proportional changes of major cell types across developmental stages (E24–E50), with color-coding indicating cell type identity. **d** Dot plot illustrating marker gene expression: The average expression levels of marker genes in various cell types of porcine female gonads are shown.

These levels are obtained from the logarithmically scaled normalized counts of high-resolution spatial transcriptomics. The percentage of cells expressing a given gene is represented by the dot size, and the expression intensity is color-coded. **e** Spatial mapping of cell populations: Cellular distributions were reconstituted on the microarray coordinate system through HDst-seq barcode alignment. The dashed box indicates the enlarged view of the corresponding area. The outer cortical, inner cortical, and medullary regions of the gonads are delineated by dashed lines. The image shown is from a single representative sample (the replicate is presented in the Supplementary Fig. 1e). Scale bar: 500 μm (zoomed-out), 100 μm (zoomed-in).

(Supplementary Fig. 3b and Supplementary Data 3). Collectively, these results reveal dynamic epigenetic changes during the specification of Mitotic PGCs into Meiotic Oogonia, including the disappearance of 5mC/5hmC and the rapid downregulation of H3K27me3.

To map the spatial distribution of FGC subtypes, we visualized their localization across five developmental stages. At E24 and E27, Mitotic PGCs were widely dispersed in a random arrangement (Fig. 2f and Supplementary Fig. 3c). From E30 to E35, RA Oogonia progressively increased in number, correlating with the migration of FGCs toward the cortical region (Fig. 2f and Supplementary Fig. 3c). By E50, Mitotic PGCs had completely disappeared, while Meiotic Oogonia had expanded to levels comparable to RA Oogonia and were predominantly localized in the cortex (Fig. 2f and Supplementary Fig. 3c, d). Quantitative analysis revealed a progressive decline in the distance between FGCs and the cortical surface from E24 to E50 (Supplementary Fig. 3d). Interestingly, the spatial organization of E50 germ cell clusters shows that meiotic oogonia are predominantly located in the internal region compared to the RA oogonia (Supplementary Fig. 3e, f). To evaluate the conservation of germ cell spatial distribution patterns, we performed a cross-species comparative analysis using spatial transcriptomics. Based on the key events in germ cell development in human and porcine fetuses, we selected corresponding developmental stages: the gonads from human female fetuses at 14 postconceptional weeks (Garcia-Alonso et al.) and porcine female fetuses at E50[3,5,39] (Fig. 2g). The results suggested a high degree of conservation in the spatial distribution of germ cells between the two species, with cells predominantly localized in the cortical region and exhibiting a gradual outward migration pattern (Fig. 2h and Supplementary Fig. 3f). However, we did not detect any Mitotic PGCs in the porcine E50 samples, whereas Mitotic PGCs in humans persisted in the outer cortical region until 20 weeks[3]. These observations could point to divergent mechanisms governing the formation of the germ cell pool in porcine and human.

In summary, these results demonstrated that the specification of PGCs into oogonia between stages E24 and E50 is marked by a progressive reduction in H3K27me3. Furthermore, a spatial distribution pattern conserved in humans is observed by E50, with germ cells predominantly localized to the cortical region.

## Spatiotemporal migratory dynamics of supporting cell lineages in porcine female gonadal development

The specification of the supporting cell lineages, originating from the coelomic epithelium, into granulosa and theca cells in the ovary is essential for germ cell development[40,41]. In pigs, PGCs colonize the porcine gonads at roughly E22, and guided by female supporting cells, start their differentiation into oogonia at roughly E28[13]. To elucidate supporting cell development, we first performed spatial visualization of *GATA4* (a zinc finger transcription factor) and *WT1* (Wilms tumor 1 protein), key markers of the supporting cell lineage, which revealed their predominant localization in the OSECs, PreGCs-I and PreGCs-II region[42,43] (Fig. 1e and Supplementary Fig. 4a). Protein-level validation through immunofluorescence staining of E24–E27 frozen sections confirmed GATA4[+] WT1[+] supporting cells primarily within these same regions (Figs. 1e, 3a and Supplementary Fig. 4a, b). Interestingly, the porcine mesonephros differs morphologically from that of mice, exhibiting a larger size and predominantly a tubular structure. Unlike the widespread expression of WT1 in the mice mesonephros, WT1 protein

expression in pigs is not widespread throughout the mesonephric region but is restricted to isolated clusters in a central subregion; both the lateral mesonephric area and the region adjoining the gonad are WT1-negative[42,43] (Supplementary Fig. 4c, d).

We used Monocle3 to analyze the developmental trajectories and fate specification of three putative supporting cell populations in the developing female gonad, which include OSECs, PreGCs-I, and PreGCs-II. Pseudo-temporal analysis revealed that OSECs and Pre-GCs I reside at earlier developmental stages (Fig. 3b). This finding may be consistent with observations in mice and humans, where OSECs and PreGCs-I have been reported to share a common coelomic epithelial origin[44–46]. Nevertheless, we observed distinct gene expression profiles between PreGC-I and both OSECs and PreGCs-II (Supplementary Data 4). PreGCs-I were characterized by high *WNT6* expression and the absence of *LHX9* and *KRT19*. In contrast, OSECs and PreGCs-II shared a transcriptional similarity but were distinguished by the lower expression of *LHX9* and *KRT19* in PreGCs-II (Fig. 3c and Supplementary Fig. 4e, f). This progressive similarity, coupled with their pseudotemporal ordering, suggests that PreGCs-II likely differentiates directly from OSECs, while PreGCs-I may represent a divergent, parallel branch of development. Together, these results indicate that the three cell types collectively form the supporting cell lineage during early gonadal development in pigs. Differential gene expression analysis revealed that the PreGCs-I were enriched for GO terms for "WNT signaling pathway" and "regulation of WNT signaling pathway", suggesting that WNT signaling may play a critical role in regulating granulosa cell differentiation[47] (Supplementary Fig. 4g, h and Supplementary Data 4). Given the established role of RSPO1/WNT4-β-catenin signaling in female gonad development in mice and humans, we interrogated the expression of pathway components[3,48]. Notably, key mediators of WNT signaling activation, including *LEF1*, *AXIN2*, and *RSPO1*, exhibited significant upregulation specifically at the PreGCs-I stage (Fig. 3d). These results suggested that the well-established role of WNT signaling as a pivotal regulator of supporting cell lineage specification during early differentiation, previously shown in mice and humans, is conserved in the pig model.

To delineate the spatial characteristics of OSECs, PreGCs-I, and PreGCs-II during porcine female gonad development, we performed spatial visualization analysis on these cell populations (Fig. 3e and Supplementary Fig. 4i). In E24 and E27, these cells exhibited a distinct stratified distribution pattern, with OSECs positioned in the outer cortical region, PreGCs-II located medially adjacent to the OSECs, and PreGCs-I located to the medullary region. Notably, from E30 to E50, PreGCs-I were exclusively localized to the medullary region, while PreGCs-II showed limited inward migration towards the inner cortical region during this period (Fig. 3e and Supplementary Fig. 4i). To validate the spatial transcriptomic data, we performed immunofluorescence staining. The results showed that GATA4 (a supporting cell lineage marker) and KRT19 (a marker for OSECs) colocalized in the inner cortex of the fetal gonad, consistent with the spatial transcriptomics visualization (Fig. 3f and Supplementary Fig. 4f). Having established the spatial dynamics of the porcine supporting cell lineage, we next assessed its evolutionary conservation through a cross-species analysis. To this end, we utilized a publicly available spatial transcriptomics dataset of human ovaries at 14 postconceptional weeks (Garcia-Alonso et al.) and compared it with our porcine E50 data, stages chosen for their comparable level of gonadal differentiation[3]. The results showed that the supporting cell

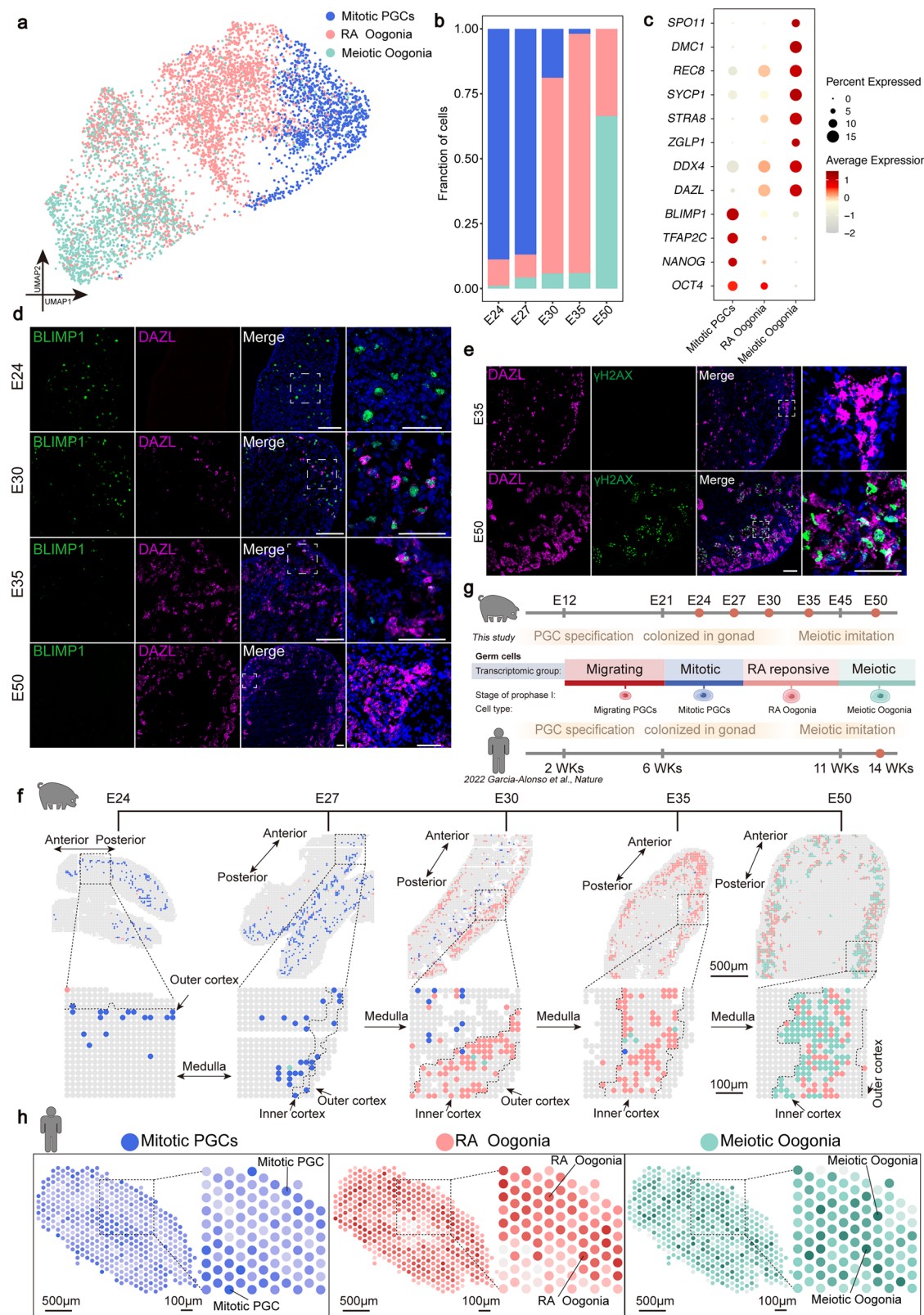

lineages in female gonads of both pigs (E50) and humans (14 weeks) exhibit a highly conserved spatial distribution: OSECs are restricted to the outer cortical region, PreGCs-II cells are distributed throughout the inner cortex, and the less abundant PreGCs-I population resides specifically within the medullary region (Fig. 3g).

Taken together, our study unequivocally revealed the stratified characteristics of supporting cell lineage subtypes and their migratory dynamics and demonstrated that two waves of granulosa cells were present as early as E24. Moreover, the spatial distribution of this lineage is highly conserved in both pigs and humans.

**Fig. 2 | The spatiotemporal transcriptional characteristics of three FGC subtypes. a** UMAP analysis reveals three fetal germ cell subpopulations: Mitotic PGCs, RA Oogonia, and Meiotic Oogonia. **b** Temporal dynamics of germ cell proportions. Bar plot quantifies the relative abundance of germ cell subpopulations across developmental stages (E24-E50), color-coded by cell state. **c** Dot plot presents the characteristic genes expressed by germ cells at the three different stages. The size of the dots represents the percentage of cells expressing each gene, and the color coding is used to indicate the gene expression level. **d** Immunofluorescence images of BLIMP1 (green) and DAZL (purple) in porcine female gonadal sections from different timepoints. Nuclei were counterstained with DAPI (blue). The dashed box indicates the enlarged view of the corresponding area. Scale bars, 100 μm (zoomed-out), 50 μm (zoomed-in). **e** Immunofluorescence images of DAZL (purple) and γH2AX (green) in porcine female gonadal sections from different timepoints. Nuclei were counterstained with DAPI (blue). The dashed box indicates the enlarged view of the corresponding area. Scale bars, 100 μm (zoomed-out), 50 μm (zoomed-in). **f** Spatial visualization of the three porcine FGC subpopulations, with each subpopulation color-coded. The dashed box indicates the enlarged view of the corresponding area. The outer cortical, inner cortical, and medullary regions of the gonads are delineated by dashed lines. The image shown is from a single representative sample (the replicate is presented in the Supplementary Fig. 3c). Scale bar: 500 μm (zoomed-out), 100 μm (zoomed-in). **g** Timeline of oogenesis and sampling strategy for cross-species analysis in pigs and humans. **h** Spatial localization of three human FGC subgroups at 14 gestational weeks. The dashed box indicates the enlarged view of the corresponding area. Scale bars: 500 μm (zoomed-out), 100 μm (zoomed-in).

## Spatial transcriptomic profiles of mesenchymal, interstitial, and endothelial cells in early gonads

In early gonad sections (E24 and E27), mesenchymal cells constitute a substantial fraction of the cell population; yet, they contribute minimally to gonadogenesis in later developmental stages. To explore their potential functions, we performed an in-depth analysis of mesenchymal cells and uncovered extensive heterogeneity. Through iterative clustering, we identified two distinct subpopulations with divergent distribution patterns: Mesenchymal cells 1 (MC-1) exhibit high expression of *ARX* and *MAN1A1*, whereas Mesenchymal cells 2 (MC-2) express *NR2F1* and *RARB*[49,50] (Fig. 4a, b). Spatial visualization revealed a stratified distribution pattern, with MC-1 localizing to the gonadal tissue boundary region and MC-2 positioned directly beneath them (Fig. 4c and Supplementary Fig. 5a). Given that the gonad-mesonephros interface is a site of active tissue remodeling and cell migration and considering prior evidence that *MAN1A1* silencing can promote cell migration and *RARB* knockout enhances EMT, we hypothesized that these subpopulations might possess differential migratory capacities[50]. Supporting this notion, GO enrichment analysis showed that MC-2 was enriched in terms like "mesenchyme development," while MC-1 was enriched in both "mesenchyme development" and the "negative regulation of cell migration" (Supplementary Fig. 5b, c and Supplementary Data 5). Collectively, our findings suggest the presence of two mesenchymal subpopulations residing at the periphery of the developing gonad, which appear to exhibit relatively distinct transcriptional profiles and spatial distributions. Their positioning along the tissue boundary, together with their divergent gene expression signatures, provides a plausible framework for further studies into their potential functions at the gonadal–mesonephric interface, which may involve roles in regulating cell migration.

During the early gonadal stages (E24 and E27), both Interstitial cell-1 (IC-1) and Interstitial cell-2 (IC-2) populations were limited in number. As development advanced to E30–E50, the interstitial cell population progressively expanded, primarily localizing within the medullary region (Fig. 4f). Spatial visualization revealed that IC-1 cells were initially located near the gonadal-mesonephric interface and subsequently disseminated throughout the gonad over time. In contrast, IC-2 cells displayed a sparse and dispersed distribution at early stages, but their numbers increased substantially by E50, accompanied by a more clustered spatial organization. Notably, the distribution patterns of IC-1 and IC-2 varied between sections at E50. In one section, they occupied separate regions, while in another, IC-2 was positioned closer to the inner cortex, whereas IC-1 remained largely confined to the medulla (Fig. 4f). The observed spatial heterogeneity is likely attributable to the specific location of the sections within the gonad, particularly differences between anterior and posterior regions. To further investigate their functional divergence, we performed differential gene expression analysis (Supplementary Data 5). GO enrichment analysis indicated that IC-1 was significantly associated with processes such as "muscle tissue development", "response to steroid hormone", and "extracellular matrix organization"[26]. Conversely, IC-2 was enriched for pathways involved in high metabolic activity, including "aerobic respiration", "ATP biosynthetic process", and "ribosome biogenesis" (Supplementary Fig. 5b, c and Supplementary Data 5). Taken together, these findings reveal two spatially and transcriptionally distinct interstitial cell populations (IC-1 and

IC-2), which appear specialized for divergent functions: IC-1 may be specialized for structural and steroid-responsive roles, whereas IC-2 appears to be in a state of high metabolic activity, potentially supporting proliferative demands.

Ovarian function is dependent on the establishment and continual remodeling of a complex vascular system[51]. We therefore profiled the expression of angiogenesis-related genes in endothelial cells and interstitial cells (Fig. 4g). Among the key factors identified, *PDGFB* and *PDGFD*, both known to promote endothelial proliferation, were detected in ECs[51,52]. ECs also highly expressed the junctional protein *CDH5* (VE-cadherin), which is critical for vascular integrity, as well as *VEGFC* and *PECAM1*, central regulators of angiogenesis[53] (Fig. 4g and Supplementary Fig. 5d). To delineate the spatial relationship between endothelial cells and germ cells, we performed spatial visualization at E50 (Fig. 4h). Interestingly, spatial analysis at E50 revealed clusters of endothelial cells near the inter cortical region, positioned adjacent to germ cells. Immunofluorescence staining further suggested that PECAM1⁺ cells formed structures exhibiting a morphology reminiscent of vessels and were localized in proximity to germ cells (Fig. 4i). In summary, these results suggested that the distribution of interstitial cells in the medulla and endothelial cells near the inner cortex supports their potential functional collaboration, which may be critical for supporting female germ cell development.

## Spatiotemporal regulation of gonadal somatic cell niches in oogonia development

During the specification of PGCs into oogonia, FGCs progressively migrate toward the cortical regions. To identify somatic cell types critical for oogonia specification, we analyzed the spatial distribution of gonadal somatic cells surrounding three FGC subtypes at distinct developmental stages (Fig. 5a and Supplementary Fig. 6a). During the E24 to E50 developmental period, the germ cell microenvironment is predominantly composed of supporting cell lineages and interstitial cells (Fig. 5b and Supplementary Fig. 6b). At E24, Mitotic PGCs are predominantly surrounded by PreGCs-I, with only minimal presence of OSECs and PreGCs-II. As development proceeds to E35–E50, PreGCs-II become increasingly localized around germ cells, suggesting a progressive migration of germ cells toward the cortical region (Fig. 5a and Supplementary Fig. 6a). PreGCs-II increased markedly during oogonia specification and supporting cell markers *LHX9* and *WNT6* were localized to cortical regions adjacent to FGCs in E50 gonads[44,54] (Fig. 5c and Supplementary Fig. 6c, d). Mesenchymal cells are rarely distributed around germ cells, whereas endothelial cell numbers remain relatively constant throughout this period. Notably, the population of interstitial cells increases over developmental time, suggesting possible functional interactions with germ cells[55] (Fig. 5b and Supplementary Fig. 6b). Consistent with these observations, immunofluorescence staining reveals that PDGFRA-positive interstitial cells are predominantly localized in the medullary region, with some cells situated around oogonia in the cortical region (Fig. 5d). These results suggested that supporting cell lineages and interstitial cells are predominantly localized around germ cells and play potential critical roles in the specification of PGCs into oogonia.

In addition to the spatial distribution of supporting cells, we also investigated the expression of hormone- and matrix-related genes in the

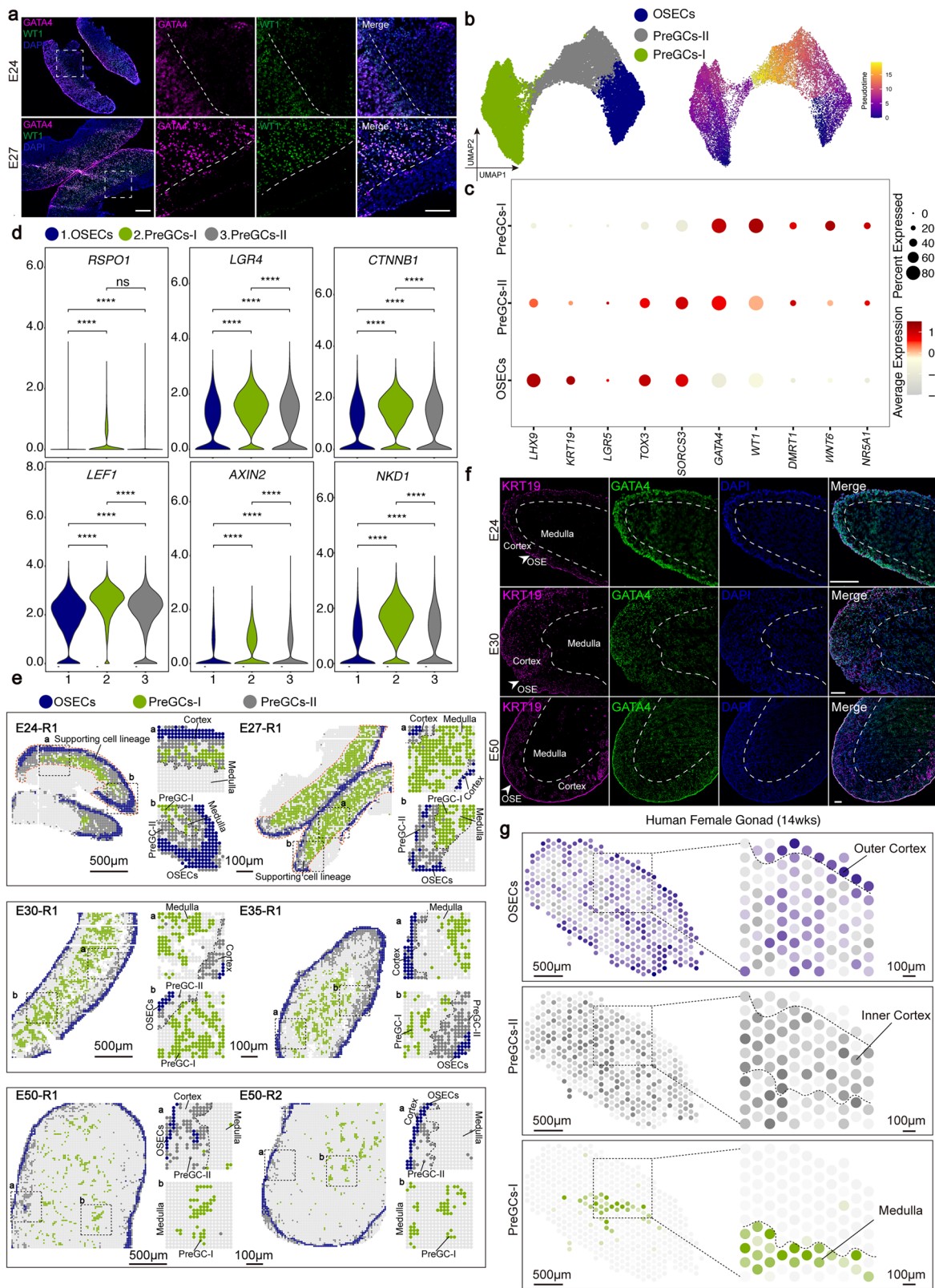

Communications Biology| (2026)9:487

developing gonads. We further observed that hormone- and matrix-related genes, including *NRG1* and *LAMA1*, were highly expressed in developing gonads (Fig. 5e and Supplementary Fig. 6e). *NRG1*, essential for oocyte maturation in mice, was enriched in the cortical regions of germ cells from E24 to E50, suggesting its potential role in oogonia differentiation[56,57] (Supplementary Fig. 6d). *LAMA1*, a key extracellular matrix (ECM)

component critical for early embryonic development and organogenesis, exhibits a dynamic localization pattern[46,58,59]. Spatial analysis revealed that *LAMA1* was confined to the gonadal medullary region at early stages (E24 and E27) but later expanded to both cortical and medullary regions (E30–E50) (Supplementary Fig. 6e). Notably, immunofluorescence staining at E50 showed that LAMA1 completely encapsulated numerous FGCs,

**Fig. 3 | Dynamic gene expression and spatial patterns of cells in the supporting cell lineages. a** Immunofluorescence images of GATA4 (purple) and WT1 (green) in porcine female gonadal sections from different timepoints. Nuclei were counterstained with DAPI (blue). The dashed box indicates the enlarged view of the corresponding area. The dashed line demarcates a region positive for both GATA4 and WT1 (left) from a region negative for these markers (right). Scale bar, 100 μm (zoomed-out), 50 μm (zoomed-in). **b** Monocle3 was applied to perform UMAP dimensionality reduction on spatial transcriptome data from three types of supporting cell lineages, and pseudotemporal trajectory analysis was used to reconstruct the developmental pathway. **c** Dot plot displays the average expression levels of representative genes in the supporting cell types in porcine female gonads. The average expression levels are obtained from the logarithmically scaled normalized counts of single-cell spatial transcriptomes. The percentage of cells expressing each gene is indicated by the size of the dots, and the color shade encodes the expression intensity. **d** Violin plot illustrates the differential expression patterns of downstream effector molecules of the WNT pathway in the supporting cell lineage subpopulation. Results are analyzed by wilcox test. *adjp < 0.05; **adjp < 0.01; ***adjp < 0.001; ****adjp < 0.0001; ns not significant, adjP ≥ 0.05. **e** Spatial visualization illustrates the distribution of three distinct subgroups of porcine supporting cell lineage at different developmental stages, each color-coded for identification. **a**, **b** show magnified views of different regions in the figure. The cortical and medullary regions of the gonads are delineated by dashed lines. The image shown is from a single representative sample (the replicate is presented in the Supplementary Fig. 4g). Scale bars: 500 μm (zoomed-out), 100 μm (zoomed-in). **f** Immunofluorescence images of KRT19 (purple) and GATA4 (green) in porcine female gonadal sections from different timepoints. The dotted line demarcates the boundary between the gonad cortex and medulla; the white arrow indicates the ovarian surface epithelium (OSE). Nuclei were counterstained with DAPI (blue). Scale bars, 100 μm. **g** Spatial localization of three human supporting cell lineage subgroups at 14 gestational weeks. The dashed box indicates the enlarged view of the corresponding area. Scale bar: 500 μm (zoomed-out), 100 μm (zoomed-in).

forming an enclosure within which female germline cysts develop[60] (Fig. 5f). The encapsulation of germline cysts by this structure suggests a potential role in providing a supportive niche for germ cells during development. Meanwhile, we also observed that *LAMA1* was expressed in various cell types (Fig. 5e). In summary, these results indicate that the supporting cell lineages and interstitial cells interact with germ cells in a spatial context, suggesting that this cell population may play a more critical role during the specification of PGCs into oogonia (Fig. 5g). Furthermore, LAMA1, as a key component of the ECM, provides structural support by enveloping the germ cells.

## BMP signaling pathway critically regulates oogonia specification and meiotic initiation

Given that FGCs are surrounded by both supporting cell lineages and interstitial cells, we investigated how these cell types collectively promote PGC specification into oogonia. The ligand-receptor interaction analysis performed with CellChat evaluated interactions between germ cells and both their surrounding supporting cell lineages and interstitial cells. Given the established role of WNT signaling in germ cell and gonadal somatic cell differentiation, we evaluated WNT pathway activation in fetal gonads (Fig. 6a, b; Supplementary Fig. 7a, b and Supplementary Data 6). We identified a significant interaction involving WNT ligands (WNT3A, WNT5A, and WNT6) from PreGCs-I and PreGCs-II, and WNT receptors (FZD3 and FZD8), expressed in Mitotic PGCs and RA Oogonia (Fig. 6a, b and Supplementary Fig. 7a, b). The TCF/LEF family, which mediates canonical WNT signaling by facilitating nuclear CTNNB1 (β-catenin) interactions with DNA, was also analyzed[61,62]. To assess canonical WNT pathway activation during the PGC-to-oogonia transition, we examined CTNNB1 localization. High levels of membrane-associated CTNNB1 were detected in somatic cells and FGCs of gonads at various developmental stages. However, weak nuclear CTNNB1 was detected exclusively in E35 oogonia, suggesting that the canonical WNT signaling mechanism may be transiently active only in E35 Oogonia (Fig. 6c).

The BMP signaling pathway not only regulates PGCs specification but has also been shown in recent studies to play a pivotal role in oogonia specification[63,64]. We noticed that BMP ligand-receptor interactions between FGC subtypes and surrounding somatic cells revealed progressively increasing BMP ligand (BMP2, BMP4, and GDF5) activity from E27 to E50 (Fig. 6d, e; Supplementary Fig. 7a, b and Supplementary Data 6). At E50, BMP2 and BMP4 are predominantly expressed by PreGCs-II and IC-1, with minimal contribution from OSECs and PreGCs-I to germ cell interactions; however, no significant interactions were found between IC-2 and germ cells (Fig. 6d). Furthermore, GDF5 is secreted exclusively by PreGCs-II (Fig. 6d). Particularly during the E50 stage, the BMP signaling pathway exhibits a significant bidirectional role between germ cells and PreGC-II/IC-1, suggesting that this pathway not only serves as a mediator through which supporting cells influence germ cells, but may also function as an important tool for germ cells to provide feedback and regulate their microenvironment

(Fig. 6d). During the transition from RA oogonia to meiotic oogonia, BMP signaling activity progressively intensifies (Fig. 6d, e and Supplementary Fig. 7a, b). In contrast, BMP signaling in human and mouse is predominantly active and its function is largely restricted to the RA oogonia stage[20,27,65]. These findings suggest that the BMP signaling pathway may remain active in porcine germ cells as they initiate meiosis. Meiotic oogonia exhibited a significant upregulation in the expression of *ID* genes, which are established downstream targets of the BMP signaling pathway, compared to RA oogonia and Mitotic PGCs[66] (Supplementary Fig. 7c). This suggests that BMP signaling activation plays an important role in the PGC-to-oogonia transition. In the single-cell transcriptomic data of porcine female gonads from E45 to E75, we similarly observed that the BMP downstream target genes, the *ID* family and *ZGLP1*, were predominantly expressed in meiotic oogonia and oocytes[21] (Supplementary Fig. 7d–f). This further suggests that BMP signaling may play a sustained role during oocyte development. We examined the activity of the BMP signaling pathway and found that at E24 and E27, it was primarily enriched at the gonadal-mesonephros interface. As development progressed to E50, BMP signaling became predominantly localized in the inner cortical region, while its expression level decreased on the side adjacent to the mesonephros (Fig. 6f, g and Supplementary Fig. 7g). RA, a known key inducer of meiosis, has been reported in mice to act synergistically with BMP2 in promoting meiotic initiation[67]. Therefore, we further analyzed the activity of the RA signaling pathway and observed that, similar to BMP signaling, it was also enriched at the gonadal-mesonephros interface during stages E24 and E27. However, by E50, RA signaling showed high expression on both sides of the cortical region (Fig. 6f and Supplementary Fig. 7g). These results suggested a clear spatial separation between the active regions of the BMP and RA signaling pathways, with no colocalization observed. Since a hallmark of BMP pathway activation is the nuclear translocation of phosphorylated (p) SMAD proteins, we assessed pSMAD1/5 localization in fetal gonads[68]. At E24, no pSMAD1/5 expression was detected in BLIMP1+ cells; at E35, pSMAD1/5 was exclusively localized to the cytoplasm of DDX4+ cells; whereas by E50, distinct nuclear expression was observed in DDX4+ cells (Fig. 6h and Supplementary Fig. 7h). Our analysis demonstrated that the BMP signaling pathway (BMP2, BMP4, and GDF5) is critical for the specification of PGCs into oogonia, as indicated by the upregulation of *ID* genes and nuclear translocation of pSMAD1/5.

Beyond WNT and BMP signaling, we explored additional ligand–receptor interactions between FGC subpopulations and supporting cell lineage, particularly within the RTK signaling pathway (Supplementary Fig. 8a–d and Supplementary Data 6). Key cytokines (KIT, EGF, and FGF) exhibited dynamic expression patterns. KIT was highly expressed in FGCs at E24 and E30 but showed reduced expression in a subset of DDX4+ germ cells at E50, indicating a gradual decline in KIT function during the PGC-to-oogonia transition[69,70] (Supplementary Fig. 8e, f). During the oogonia specification process, we noticed significant interactions between NOTCH ligands (DLK1) from IC-2 and NOTCH1/2 expressed by oogonia at E50, suggesting that NOTCH signaling may be essential for oogonia

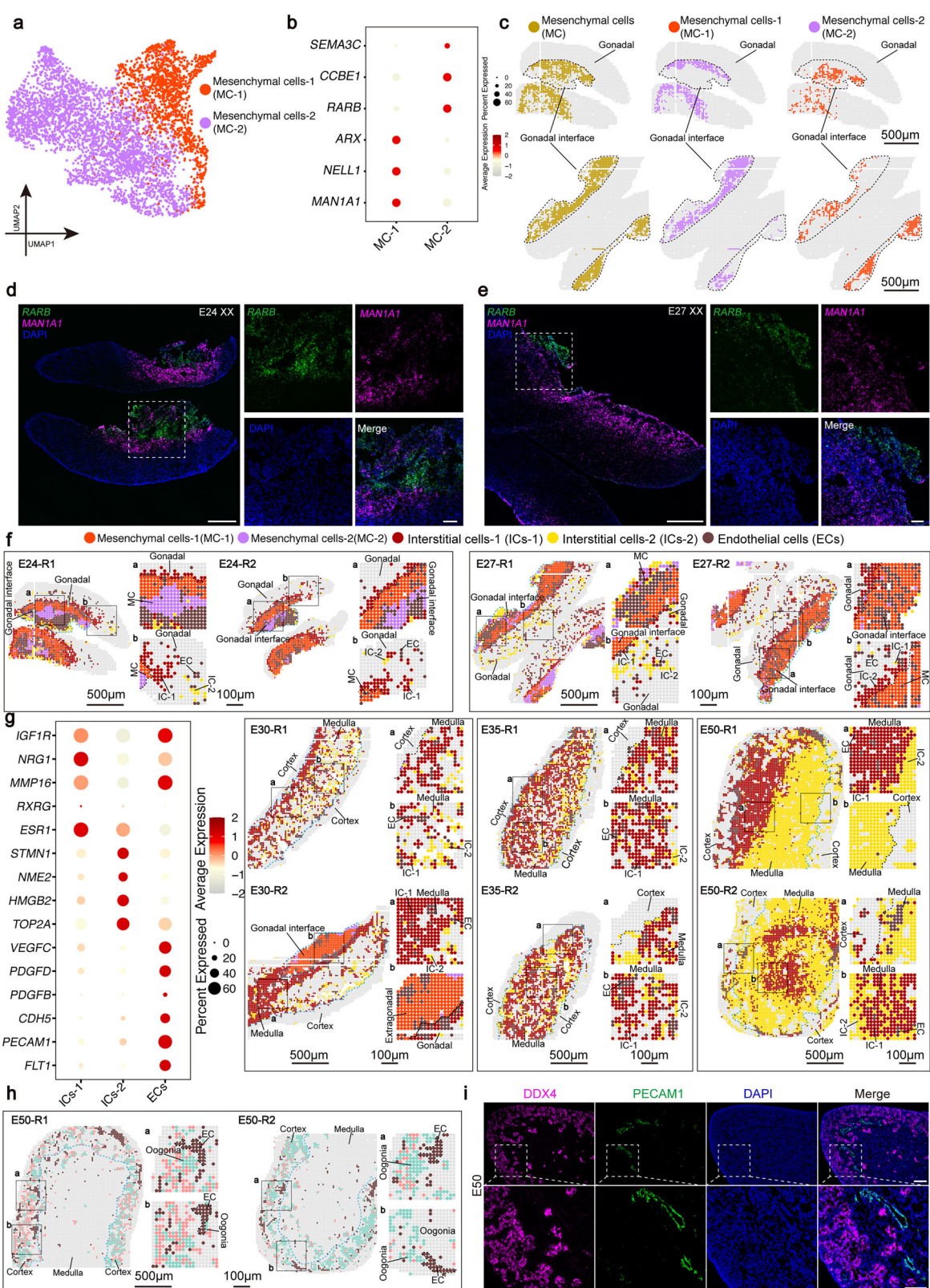

specification[71,72] (Supplementary Fig. 8f and Supplementary Data 6). Concurrently, the upregulation of neural-associated molecules, including axon guidance components NCAM1 homodimers, suggests potential mechanistic parallels with neural developmental pathways during the PGC-oogonia transition (Supplementary Fig. 8a–d). Ligand-receptor analysis further revealed critical ECM involvement, with the LAMA1-DAG1 complex and

COL1A1-(ITGA2 + ITGB1) interactions emerging as key functional modules, suggesting that ECM components via coordinated secretion and functional integration establish a regulatory network essential for oogonia specification[73] (Supplementary Fig. 8a–d).

In summary, BMP signaling plays a critical role in regulating PGC commitment to the oogonia and meiotic initiation through a regulatory

**Fig. 4 | Heterogeneity and spatial organization of mesenchymal, interstitial and endothelial cells in the developing porcine gonad. a** UMAP plot shows that the MCs are divided into two subpopulations. **b** Dot plot displays the characteristic gene expression in two mesenchymal cell subgroups. The diameter of each dot representing a gene indicates the proportion of cells expressing the gene, and the color gradient corresponds to the expression intensity. **c** Spatial visualization illustrates the distribution of two distinct subgroups of mesenchymal cells at different developmental stages, each color-coded for identification. The dashed line indicates the gonad-mesonephros interface. Scale bar: 500 μm. **d** RNA in situ hybridization for *RARB* (green, probe) and *MAN1A1* (purple, probe) in porcine female gonadal sections from E24. The cell nuclei were counterstained with DAPI (blue). The dashed box indicates the enlarged view of the corresponding area. Scale bar: 250 μm (zoomed-out), 50 μm (zoomed-in). **e** RNA in situ hybridization for *RARB* (green, probe) and *MAN1A1* (purple, probe) in porcine female gonadal sections from E27. The cell nuclei were counterstained with DAPI (blue). The dashed box indicates the

enlarged view of the corresponding area. Scale bar: 250 μm (zoomed-out), 50 μm (zoomed-in). **f** Spatial visualization of endothelial, mesenchymal, and interstitial cells in the porcine gonad at different developmental stages. Each cell type is color-coded for identification. **a**, **b** Show magnified views of different regions in the figure. The cortical and medullary regions of the gonads are delineated by dashed lines. Scale bar: 500 μm (zoomed-out), 100 μm (zoomed-in). **g** Dot plot shows the expression of angiogenesis-related genes in endothelial and interstitial cells. The dot diameter indicates the proportion of expressing cells, and the color gradient reflects the expression level. **h** Spatial visualization of endothelial and germ cells in the porcine gonad at E50. Each cell type is color-coded for identification. Scale bar: 500 μm (zoomed-out), 100 μm (zoomed-in). **i** Immunofluorescence images of DDX4 (purple) and PECAM1 (green) in porcine female gonadal sections in E50 gonad. Nuclei were counterstained with DAPI (blue). The dashed box indicates the enlarged view of the corresponding area. Scale bar: 100 μm (zoomed-out), 50 μm (zoomed-in).

mechanism primarily mediated by the PreGCs-II. Furthermore, the NOTCH, ECM, and neuro-related ligand-receptors may collectively orchestrate this complex developmental transition.

## Discussion

Although single-cell transcriptome sequencing technology holds notable advantages in the identification of cellular subtypes within porcine gonadal tissues, the spatial localization of distinct cell types within early-stage gonads has not been widely investigated[17]. Here, we applied hdST-seq to female porcine embryos at five developmental stages (E24, E27, E30, E35, and E50) and systematically elucidated the spatiotemporal dynamics of heterogeneous cell populations in the developing gonads. This study bridges gaps in our understanding of early gonadogenesis by refining spatial maps of key cell types and uncovering dynamic cell-cell interactions critical for oogonia specification[17,21].

From E24 to E50, we identified three distinct germ cell stages—Mitotic PGCs, RA Oogonia, and Meiotic Oogonia—and characterized their spatial transitions from an initially dispersed distribution to progressive cortical localization. Previous studies have reported similar migratory patterns in human fetal ovaries, where mitotic germ cells localize to the outer cortex, meiotic germ cells predominate in the inner cortex, and FIGLA-positive oocytes are positioned within the medulla[3,21,38]. We integrated spatial transcriptomic data from porcine E50 female gonads and published human 14-week female gonads, which consistently revealed that Meiotic oogonia are located more internally within the gonadal tissue compared to RA oogonia. This corticomedullary gradient in early oogenesis supports the notion of spatially regulated germ cell differentiation[74]. The distributional differences of mitotic PGCs in the early gonads also reflect distinct molecular mechanisms in the formation of the female germ cell pool between pigs as a polytoeous multivalent species and humans as a monotocous species. At E50, porcine oogonia were organized into clusters within the cortical region. This spatial distribution is notably similar to that of the female germline cysts in mice, which comprise approximately 24 nurse cells and ultimately yield about 6 oocytes[75]. This similarity suggests that pigs may employ a comparable mechanism to ensure an ample cellular reserve for subsequent oogenesis. At the epigenetic level, the complete loss of 5 mC and 5 hmC by E24 indicates that demethylation likely begins during PGC migration to the gonads[5,35]. In contrast, H3K27me3 dynamics from E24 to E50 reveal a progressive decline: it is highly expressed in Mitotic PGCs at E24 and E27, partially retained at E30, and nearly absent by E35 and E50. This reduction in H3K27me3 suggests its crucial role in the specification of PGCs into oogonia.

In early gonads, OSECs, PreGCs-I, and PreGCs-II exhibit a stratified spatial distribution, functioning as key supporting cell lineages that contribute to PGC specification and intercellular communication with FGC subtypes. Beyond supporting cell lineages, two types of mesenchymal cells localized at the gonadal-mesonephros interface exhibit a stratified distribution pattern. While participating in ECM development, MC-1 may primarily serve as a boundary structure of the gonad, regulating cell

migration between the mesonephros and gonad. Endothelial cells and interstitial cells exhibit different cell migration characteristics. The transition of endothelial cells from an initially random distribution to forming an organized structure around germ cells by E50 suggests that they undergo directed migration. In contrast, the widespread distribution of interstitial cells is more likely the result of the combined effects of cell migration and proliferation. At E50, the number of both ICs significantly increases, providing not only structural support for gonadal tissues but also establishing a necessary microenvironment for germ cell development. This finding suggests that, besides granulosa cells, ICs may also play a critical role in the in vitro reconstituted of ovaries. Furthermore, in the female gonad at E50, endothelial cells form vascular-like structures and localize adjacent to germ cells, thereby supplying ample nutritional support for the expansion of oogonia.

Our analysis of ligand-receptor interactions between germ cells and surrounding somatic cells highlights the pivotal role of BMP signaling (BMP2, BMP4, and GDF5) in PGC specification into oogonia. Notably, in both mice and humans, the expression of *ZGLP1*, a downstream target gene of the BMP signaling pathway, plays a critical role primarily in the RA-responsive oogonia stage prior to meiosis; whereas in pigs, *ZGLP1* remains active not only during the RA-responsive stage but also continues to function throughout meiotic stages[27,65]. This observation suggests that the temporal action of BMP signaling during germ cell development may exhibit species-specific differences. Additionally, we identified NOTCH signaling components, neural-related pathways, and matrix-associated ligand-receptor interactions as critical regulators of the PGC-to-oogonia transition. These findings provide a valuable resource for understanding in vivo germ cell specification and establish a theoretical framework for optimizing in vitro oogenesis reconstitution. We propose that the systematic application of these molecular interactions could enhance xenogeneic reconstituted ovaries in large animals, facilitating complete oogonia differentiation and meiotic entry. Notably, recent studies indicate that hPGCLCs can undergo epigenetic reprogramming independent of gonadal somatic cell support[76]. This discovery not only advances efforts to develop gonadal somatic cell-independent human oogenesis systems but also offers alternative progenitor cell sources for future oogenesis reconstitution.

In summary, this study provides a comprehensive spatiotemporal atlas of porcine embryonic gonads, highlighting PreGCs-II as a critical regulator of oogonia specification and meiotic entry. Through cell-cell communication analysis, we define BMP2, BMP4 and GDF5 as key mediators of this developmental cascade. By leveraging large animal models, our findings refine the understanding of human germ cell specification and offer insights into improving in vitro oogenesis in mammals.

## Methods
### Pig embryos and gonads collection
All experimental procedures involving animals were conducted in accordance with the guidelines and regulations for the care and use of laboratory animals, and were approved by the Animal Ethics Committee of China

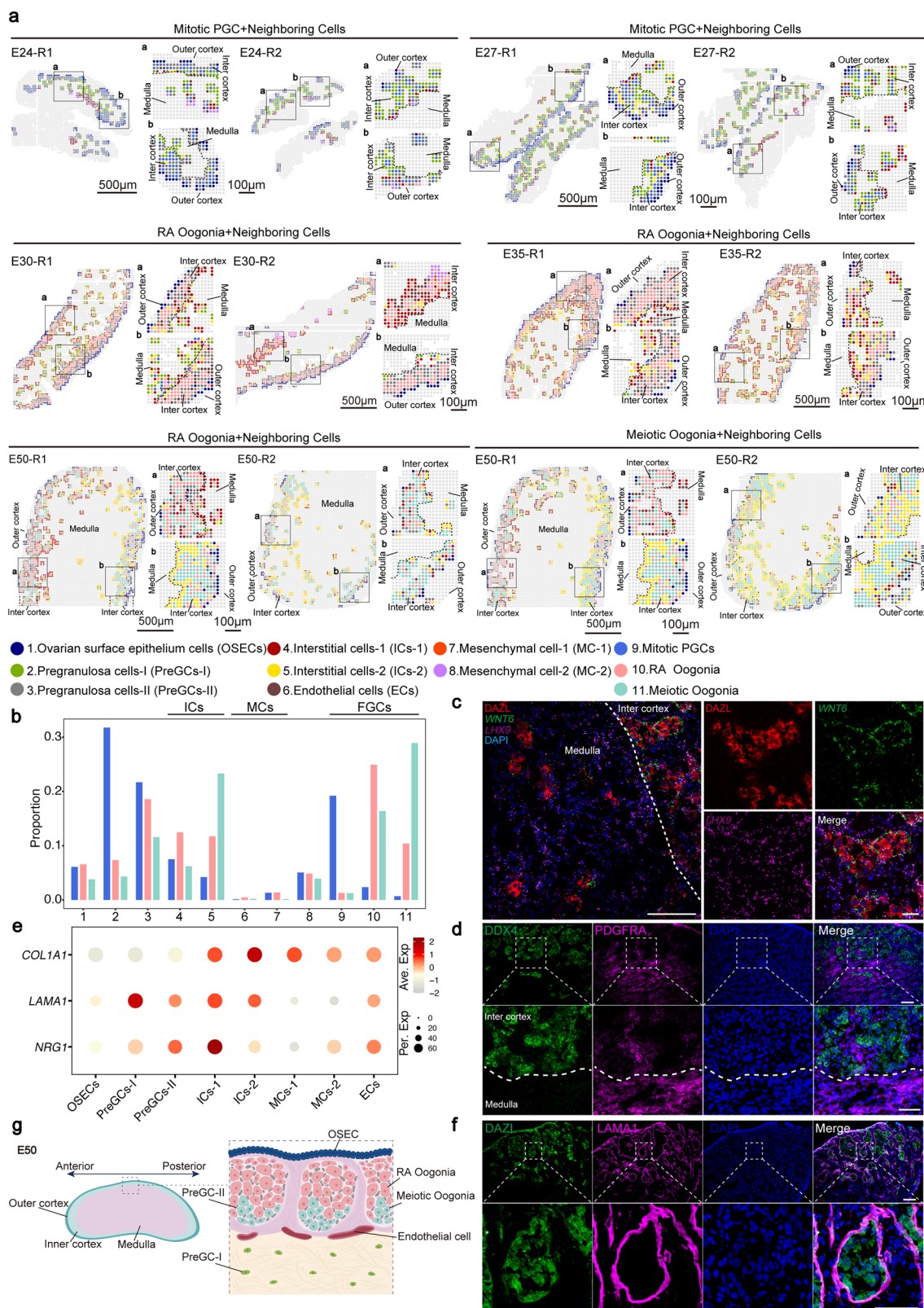

Agricultural University (Approval No: AW02113202-3-1). We have complied with all relevant ethical regulations for animal use. Porcine embryos were obtained from five pregnant sows (Large White breed, approximately 18 months old) sourced from local commercial breeding facilities. The sows were inseminated via standard artificial insemination procedures at their home facilities and were transported to the experimental facility under controlled conditions to minimize stress. At the designated gestational time points (24, 27, 30, 35, and 50 days post-fertilization, E24–E50), sows were humanely euthanized to permit the collection of embryos. Fetal gonads were harvested and rinsed in ice-cold PBS supplemented with 1% fetal bovine serum to remove residual blood, embedded in OCT compound (Sakura, 4583), and snap-frozen on dry ice. Prior to cryosectioning, embryonic sex

**Fig. 5 | Spatiotemporal analysis of three FGC subsets niches in porcine gonadal development. a** Spatial visualization showing niches surrounding three female fetal germ cell subsets from E24 to E50. Mitotic PGCs, RA Oogonia, Meiotic Oogonia. Different cell types are color-coded. **a**, **b** show magnified views of different regions in the figure. The outer cortical, inner cortical, and medullary regions of the gonads are delineated by dashed lines. Scale bars: 500 μm (zoomed-out), 100 μm (zoomed-in). **b** The bar chart represents the proportional distribution of germ cell subpopulations at different periods. The color code corresponds to the cell developmental stage. **c** Immunofluorescence images of DAZL (red), WNT6 (green, probe) and LHX9 (purple, probe) in porcine female gonadal sections of E50 gonads. Nuclei were counterstained with DAPI (blue). The dotted line separates the inner cortex from the medulla, with the left side representing the medullary region and the right side the inner cortical region. Scale bar: 100 μm (zoomed-out), 25 μm (zoomed-in). **d** Immunofluorescence images of DDX4 (green) and PDGFRA (purple) in porcine

female gonadal sections of E50 gonads. Nuclei were counterstained with DAPI (blue). The dashed box indicates the enlarged view of the corresponding area. The dashed line represents the boundary between the inner cortex and the medulla, with the area superior to it corresponding to the inner cortical region and the area inferior to it corresponding to the medullary region. Scale bar: 100 μm (zoomed-out), 50 μm (zoomed-in). **e** Dot plot displays the characteristic gene expression in various cell types. Dot size represents the percentage of cells expressing the gene, and the color intensity represents the average expression level. **f** Immunofluorescence images of DDX4 (green) and LAMA1 (purple) in porcine female gonadal sections of E50 gonads. Nuclei were counterstained with DAPI (blue). The dashed box indicates the enlarged view of the corresponding area. Scale bar: 100 μm (zoomed-out), 50 μm (zoomed-in). **g** Schematic illustration of a porcine fetal ovary at E50. The dashed box indicates the enlarged view of the corresponding area.

was determined by PCR to ensure only female gonads were processed. For sex identification, genomic DNA was extracted from embryonic tissue and subjected to PCR using specific primers for the porcine SRY and ZFX genes. The primer sequences were as follows:

SRY forward: GGGAAAGGCTCCTCACTATTT,
SRY reverse: AGGGATACATCCTCTCCTCTAC;
ZFX forward: GTGCTGCTTTGTCTTGGAATG,
ZFX reverse: GAGGGAGTTAGGTCTGGATACT.

Samples exhibiting only the ZFX amplification product were identified as female, while those showing both ZFX and SRY products were identified as male. Cryosectioning was performed at 8–10 μm thickness using a cryostat, with sections visualized under a fluorescence microscope and stored at −80 °C until further analysis.

### Sample preparation
Frozen tissue section slides were removed from the −80 °C freezer and placed on an ultra-clean platform to air dry for approximately 10 min. The OCT compound on the tissue surface was removed using PBS with RNase Inhibitor (Enzymatics). The tissue was then fixed with 4% PFA for approximately 20 min and washed three times with PBS[19].

### Permeabilization and reverse transcription
The fixed tissue was permeabilized using 0.05% Triton X-100 in PBS for 20 min and washed twice with 0.5× PBS containing RNase Inhibitor (Enzymatics). A reverse transcription mixture (24.9 μL 30% PEG6000, 12 μL 5× RT buffer, 12 μL 50 μM RT primer with an affinity tag, 7.5 μL 200 U/μL Maxima H Minus Reverse Transcriptase, 3 μL 10 mM dNTP, and 0.6 μL 40 U/μL RNase Inhibitor) was then added. The sample was incubated at room temperature for 30 min, followed by incubation in a 42 °C humidified chamber for 90 min. After reverse transcription, 50 μL NIB with 1 μL 0.5 M EDTA was added and incubated at room temperature for 3 min. The slides were then washed thoroughly with DEPC-treated water.

### Microfluidic chip processing
A PDMS microfluidic chip (Suzhou Cchip Scientific Instrument Co., Ltd) was placed over the tissue section, ensuring the chip channels aligned with the tissue. Each microwell was loaded with 3 μL barcode (Sangon Biotech (Shanghai) Co., Ltd) mixture (115.8 μL 1× NEB buffer 3.1 with 1% RI (Enzymatics), 81 μL RNase-free water, 27 μL 10× T4 Ligase Buffer, 2.7 μL 10% Triton X-100, 2.2 μL T4 DNA Ligase (2000 U/μL), 2.2 μL RNase Inhibitor, and 0.7 μL Superase in RNase Inhibitor (Ambion)) along with 1 μL of a well-specific barcode (Supplementary Data 1). The chip was incubated at 37 °C for 30 min. Subsequently, 3 μL blocking solution A was added to each microwell. The channels were washed with 1× NEB buffer, evacuated, and the PDMS chip was removed. The slides were washed again with DEPC-treated water. The second round of microfluidic chip processing was performed using the same procedure. After removing the second chip, the slides were washed, air-dried, and aligned with a circularly perforated PDMS piece. Lysis buffer (2 mg/mL proteinase K, 10 mM Tris (pH 8.0), 200 mM NaCl, 50 mM EDTA, and 2% SDS) was added, and lysis

was carried out at 55 °C for 2 h. The lysate was collected and stored at −80 °C.

### cDNA purification and amplification
The cDNAs in the lysate were purified using streptavidin beads (Dynabeads MyOne Streptavidin C1, Thermo Fisher). The beads (40 μL) were washed three times with 1× B&W buffer (5 mM Tris pH 8.0, 1 M NaCl, 0.5 mM EDTA) with 0.05% Tween-20 and stored in 100 μL of 2× B&W buffer containing 2 μL SUPERase In RNase Inhibitor. Tissue lysates were diluted with an equal volume of water to reduce SDS concentration before purification. The purified DNA was eluted using the DNA Clean and Concentrator kit (D4014, Zymo). For purification, 100 μL of MyOne C1 bead suspension was added to the sample and incubated at room temperature for 60 min with gentle rotation. Beads with cDNA were washed twice with 1× B&W buffer containing 0.05% Tween-20 and 1 μL SUPERase In RNase Inhibitor, followed by one wash with STE buffer with gentle rotation.

### Template switching and cDNA amplification
Bead-bound cDNAs were cleaned and resuspended in a template switch reaction mix containing 44 μL 5× Maxima RT buffer (Thermo Fisher), 44 μL 20% Ficoll PM-400 (Sigma), 22 μL 10 mM dNTPs (Thermo Fisher), 5.5 μL RNase Inhibitor (Enzymatics), 11 μL Maxima H Minus Reverse Transcriptase (Thermo Fisher), and 5.5 μL template switch primer (100 mM). The reaction was incubated at room temperature for 30 min, followed by incubation at 42 °C for 90 min. After template switching, the beads were washed once with 400 μL STE buffer and once with 400 μL RNase-free water. The supernatant was removed using a magnetic stand. Bead-bound cDNAs were amplified in a 55 μL PCR mix (1× Kapa HiFi PCR mix, 400 nM P7 primer, and 400 nM RNA PCR primer). The PCR conditions were: 95 °C for 3 min, followed by 6-10 cycles of 98 °C for 30 s, 65 °C for 45 s, and 72 °C for 3 min.

### Library preparation and sequencing
After amplification, the sample was centrifuged at 10,000 × g for 1 min. The supernatant was transferred to a new tube, purified using 0.8× SPRIselect beads, and eluted in 36 μL RNase-free water. The cDNA concentration was quantified using a Qubit fluorometer (Thermo Fisher). For each sample, 50 ng of cDNA was fragmented in a 50 μL tagmentation mix (35 μL cDNA, 10 μL 5× tagmentation buffer L, 2 μL custom-embedded Tn5, and 3 μL RNase-free water) at 55 °C for 10 min. An equal volume of 8 M guanidine hydrochloride was added, and the DNA was purified using 2× SPRIselect beads and eluted in 20.5 μL RNase-free water. Purified cDNA was mixed with tagmentation PCR mix (25 μL NEBNext High-Fidelity 2× PCR Master Mix, 2.5 μL 10 μM P7 primer, and 2.5 μL 10 μM Ad1 primer). PCR conditions were: 72 °C for 5 min, 98 °C for 30 s, followed by 7-9 cycles of 98 °C for 10 s, 65 °C for 30 s, and 72 °C for 1 min. The amplified library was purified using 0.7× SPRIselect beads and eluted in 12 μL RNase-free water. Library quality was assessed using a Bioanalyzer (Agilent) and Qubit (Thermo Fisher). Sequencing was performed on an Illumina NovaSeq 6000 platform with 150-bp paired-end reads.

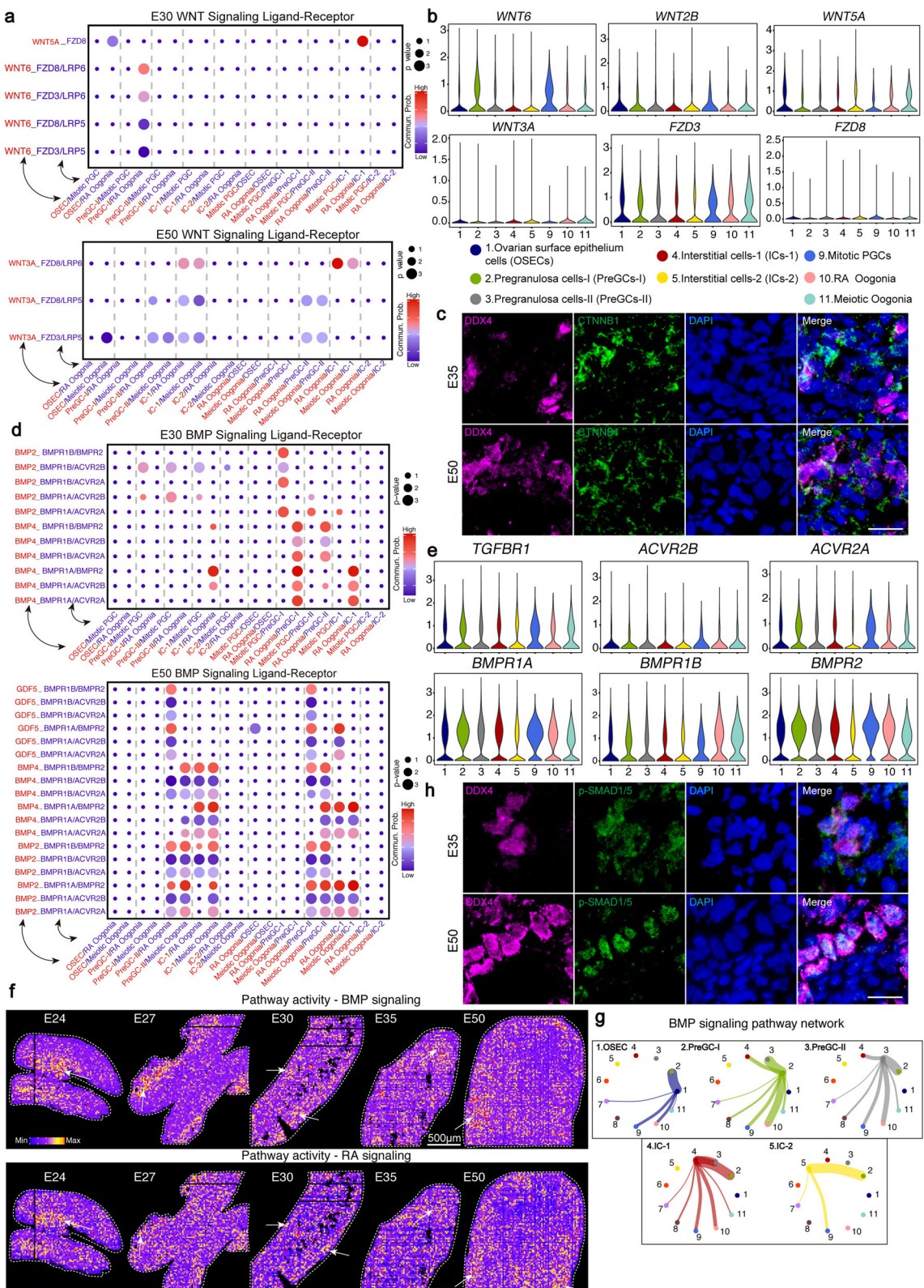

## Spatial RNA-seq data processing

This study follows a single-cell transcriptomics analysis workflow: Raw sequencing data were first processed with cutadapt (v1.18)[77] to remove adapter sequences using default parameters. Molecular barcode processing and alignment-based quantification were performed using umi-tools (v1.1.0)[78] and ST pipeline (v1.8.1)[79], generating a gene expression matrix.

For human RNA-seq data, clean reads were aligned to the GRCh38/hg38 reference genome assembly[80]. For porcine RNA-seq data, clean reads were aligned to the Sscrofa11.1 reference genome assembly[81]. The gene annotation files used in this study were downloaded from the Ensembl database. To improve data coverage, we retained reads mapping to both exonic and intronic regions during sequence alignment. The gene expression matrix

**Fig. 6 | Spatiotemporal crosstalk between three FGCs subtypes and supporting lineages in porcine gonadal development. a** WNT ligand-receptor interaction networks among fetal germ cells (Mitotic PGCs, RA oogonia, Meiotic oogonia), supporting cell lineages (OSECs, PreGCs-II, and PreGCs-I), and interstitial cells (IC-1 and IC-2) at E30 (top) and E50 (bottom). **b** Violin plots of selected WNT-associated ligands and receptors in female germ and somatic clusters. **c** Immunofluorescence images of DDX4 (green) and CTNNB1 (purple) in porcine female gonadal sections from different timepoints. Nuclei were counterstained with DAPI (blue), Scale bar: 20 μm. **d** BMP ligand-receptor interaction networks among fetal germ cells (Mitotic PGCs, RA oogonia, Meiotic oogonia), supporting cell lineages (OSECs, PreGCs-II, and PreGCs-I), and interstitial cells (IC-1 and IC-2) at E30 (top) and E50 (bottom). **e** Violin plots of selected BMP-associated ligands and receptors in female germ and somatic clusters. **f** Spatial visualization of BMP and RA pathway activity in the developing female pig gonad at different time points. The white arrows indicate the enriched regions of BMP/RA signaling activation. The image shown is from a single representative sample (the replicate is presented in the Supplementary Fig. 7d). Max, maximum; min, minimum. Scale bar: 500 μm. **g** Circle plot visualizing the intercellular communication network mediated by the BMP signaling pathway among different cell types. Outer Sectors (Nodes): Each node (sector) identifies a specific cell type included in the analysis, labeled by its color and name. Arrow Direction: Indicates the direction of signal flow, representing communication from the sending cell population (source) to the receiving cell population (target).Edge Thickness: Represents the total strength or number of links of BMP signaling exchange between the pair of cell populations. Thicker lines denote stronger communication. Line Color: Is consistent with the color of the sending cell type (the source sector). **h** Immunofluorescence images of DDX4 (purple) and p-SMAD1/5 (green) in porcine female gonadal sections from different timepoints. Nuclei were counterstained with DAPI (blue), scale bar: 20 μm.

was then processed and integrated using Seurat (v4.4.0). Normalization was performed with the SCTransform() function, and multiple samples were merged using the merge() function. To reduce dimensionality, we applied principal component analysis (PCA) via RunPCA() to capture major sources of variation, followed by batch effect correction with Harmony (v1.2.0), which significantly enhanced data consistency across samples. The set of highly variable genes was selected based on genes commonly expressed across all samples. For cell type annotation, differentially expressed genes (DEGs) were identified, and cell types were assigned based on classical marker genes and their spatial expression patterns. Ultimately, five developmental stage-specific cell populations were precisely annotated.

### Spatial transcriptomics data processing[82]
To construct a Seurat object for spatial transcriptomics analysis, raw count matrices from multiple samples were loaded and preprocessed. Each dataset was imported as a gene-by-pixel matrix, with pixel coordinates extracted from row names. The data was transformed into a Seurat object using CreateSeuratObject(), and sample metadata, including channel, developmental stage, and spatial coordinates, was annotated accordingly. A filtering step was applied to remove low-quality pixels based on total transcript counts, ensuring robust downstream analysis. The remaining pixels underwent SCTransform() normalization with 3000 variable features, standardizing expression values while preserving biologically relevant variations. The processed Seurat objects for each sample were returned for integration and further analysis.

### Single-cell RNA-seq data integration and clustering analysis
To integrate multiple single-cell RNA-seq datasets and identify distinct cellular clusters, we processed the data using Seurat (v4.4.0) and RPCA for batch correction. First, highly variable genes were selected from the SCT-normalized expression matrix, ensuring consistency across datasets. Principal component analysis (PCA) was performed using RunPCA(), and the elbow plot was examined to determine the number of significant principal components for downstream analysis.

To correct batch effects across samples, IntegrateLayers() was applied, aligning cells based on their original dataset identities. Dimensionality reduction was subsequently performed using RunUMAP(), employing the corrected harmony embeddings. A shared nearest neighbor (SNN) graph was constructed using FindNeighbors(), followed by FindClusters() to assign cells to clusters. The clustering resolution was adjusted to 0.3, balancing granularity and biological relevance. The results were visualized through UMAP projections, with clusters and sample distributions highlighted using DimPlot().

For differential expression analysis, marker genes were identified using FindAllMarkers() under both RNA and SCT normalization frameworks. Only genes with a minimum expression fraction of 10% and a log2 fold change ≥0.5 were considered significant. The top five marker genes per cluster were extracted and ranked by fold change, and heatmaps were generated using DoHeatmap() to visualize their expression patterns across clusters. The final marker gene lists were saved as CSV files for further

interpretation. For cell type annotation, differentially expressed genes (DEGs) were identified, and cell types were assigned based on classical marker genes and their spatial expression patterns. Ultimately, five developmental stage-specific cell populations were precisely annotated.

### Cluster spatial distribution analysis
To analyze the spatial distribution of cell clusters across different samples, we extracted cluster identity information and mapped barcode coordinates. Cells were grouped based on their cluster assignments, and barcode positions were parsed from cell identifiers. To ensure consistency, barcode indices were factorized and ordered accordingly. Spatial distribution patterns were visualized using ggplot2 in R[83], with each cluster assigned a distinct color from a predefined palette. The scatter plots were generated with geom_point(), displaying barcode positions while maintaining aspect ratio with coord_equal(). To enhance readability, x- and y-axis labels were formatted using scale_x_continuous() and scale_y_reverse(), and plot aesthetics were refined by removing grid lines and setting a minimal background. Each sample was processed individually, and the resulting figures were exported as PDF files for further interpretation.

### Clustering and subtype identification of fetal germ cells (FGCs)
To further refine the classification of FGCs, we performed additional clustering analysis. First, cells belonging to a specific FGC cluster were extracted for downstream analysis. The subset of cells was then subjected to graph-based clustering using the FindNeighbors() and FindClusters() functions in Seurat. To improve subtype identification, we applied a low clustering resolution (resolution = 0.1) and further isolated a specific subpopulation. To refine subtype annotations, cells expressing SYCP3 were manually reassigned to a new cluster. The final cluster labels were updated accordingly to reflect the new classification. All clustering steps were performed in Seurat (v4.4.0 or later) using PCA-based nearest neighbor graph construction and Louvain clustering. The reclassified subtypes were then used for downstream analysis.

Gene Module Scoring and Visualization: To investigate the activity of gene modules related to DNA demethylation and histone modification, predefined gene sets were compiled based on prior biological knowledge. Gene module scores were computed using AddModuleScore() in Seurat to normalize the scores. For visualization, violin plots were generated using VlnPlot(), displaying module scores across different cell populations. Statistical summaries (median values) were overlaid on the plots to facilitate interpretation. Boxplots were added to enhance clarity by summarizing score distributions while excluding outliers. Custom color palettes were applied to maintain consistency across figures, and axis labels were adjusted to improve readability.

### Cluster spatial distribution analysis
To analyze the spatial distribution of cell clusters across different samples, we extracted cluster identity information and mapped barcode coordinates. Cells were grouped based on their cluster assignments, and barcode positions were parsed from cell identifiers. To ensure consistency, barcode

indices were factorized and ordered accordingly. Spatial distribution patterns were visualized using ggplot2 in R[83], with each cluster assigned a distinct color from a predefined palette. The scatter plots were generated with geom_point(), displaying barcode positions while maintaining aspect ratio with coord_equal(). To enhance readability, x- and y-axis labels were formatted using scale_x_continuous() and scale_y_reverse(), and plot aesthetics were refined by removing grid lines and setting a minimal background. Each sample was processed individually, and the resulting figures were exported as PDF files for further interpretation.

### Clustering and subtype identification of fetal germ cells (FGCs)

To further refine the classification of FGCs, we performed additional clustering analysis. First, cells belonging to a specific FGC cluster were extracted for downstream analysis. The subset of cells was then subjected to graph-based clustering using the FindNeighbors() and FindClusters() functions in Seurat. To improve subtype identification, we applied a low clustering resolution (resolution = 0.1) and further isolated a specific subpopulation. To refine subtype annotations, cells expressing SYCP3 were manually reassigned to a new cluster. The final cluster labels were updated accordingly to reflect the new classification. All clustering steps were performed in Seurat (v4.4.0 or later) using PCA-based nearest neighbor graph construction and Louvain clustering. The reclassified subtypes were then used for downstream analysis.

Gene module scoring and visualization: To investigate the activity of gene modules related to DNA demethylation and histone modification, predefined gene sets were compiled based on prior biological knowledge. Gene module scores were computed using AddModuleScore() in Seurat to normalize the scores. For visualization, violin plots were generated using VlnPlot(), displaying module scores across different cell populations. Statistical summaries (median values) were overlaid on the plots to facilitate interpretation. Boxplots were added to enhance clarity by summarizing score distributions while excluding outliers. Custom color palettes were applied to maintain consistency across figures, and axis labels were adjusted to improve readability.

### Human gonad cell type deconvolution of spatial transcriptomic data

Human gonadal Cell type deconvolution of spatial transcriptomic data was performed using spacexr[84]. A SpatialRNA object was first constructed by integrating the spatial expression matrix with the corresponding spatial coordinates. Subsequently, an RCTD object was created by combining the spatial data with a reference single-cell RNA-seq dataset using the create.RCTD() function. Model fitting and deconvolution were carried out with the run.RCTD() function in full mode, which estimates both single-cell and doublet compositions within each spatial capture spot. This configuration allows the algorithm to model complex cellular mixtures and achieve robust deconvolution across heterogeneous tissue regions. The final outputs included the estimated cell type proportions for each spatial spot, which were further used for downstream visualization and spatial correlation analyses.

### Developmental trajectory inference

To reconstruct lineage differentiation trajectories, we utilized Monocle3 (v1.3.1)[85] with default parameters. Notably, to avoid potential algorithmic differences affecting visualization, we did not directly use Monocle3's UMAP output. Instead, we displayed trajectory inference results within a unified UMAP embedding generated using Seurat's RunUMAP() function. This approach ensured visualization consistency by integrating trajectory inference with the existing dimensional reduction pipeline.

### Functional annotation and pathway enrichment analysis

GO enrichment analysis. For functional annotation of DEGs, we conducted Gene Ontology (GO) enrichment analysis using clusterProfiler (v4.11.0)[86] and the human gene annotation database org.Hs.eg.db. The enrichGO() function was used to annotate cellular components (CC), molecular functions (MF), and biological processes (BP). To identify significantly enriched terms, a p-value < 0.05 and FDR-adjusted q-value < 0.1 were applied. The enriched terms were visualized using barplot() and dotplot() to highlight core functional modules, providing insights into the subcellular localization, molecular activities, and regulatory networks of DEGs.

KEGG pathway analysis. To explore the biological pathways associated with DEGs, we performed KEGG pathway enrichment analysis using the enrichKEGG() function. Pathways related to metabolism and signal transduction were systematically analyzed. Significant pathways (p-value < 0.05, q-value < 0.1) were further examined using hierarchical clustering and pathway topology analysis. Key pathways were visualized with heatmaps and bar plots, elucidating the roles of DEGs in dynamic cellular processes.

Cell-cell communication analysis. We performed cell-cell communication network analysis using CellChat (v1.6.1)[87]. First, the gene expression matrix was used to construct a CellChat object via createCellChat(). Cell types were annotated based on known classifications or clustering results. We then calculated cell-cell communication probabilities using computeCommunProb(), constructing an interaction network. To identify key signaling pathways and receptor-ligand interactions, we performed network analysis with netAnalysis.compute() and determined the signaling role of each pathway using netAnalysis.signalingRole(). The cell-cell communication network was visualized using customized plot() functions, allowing us to explore intercellular interactions and their potential biological implications.

Neighborhood cell selection. To identify neighboring cells, we first extracted cells from the target cell population along with their meta.data using WhichCells(), retrieving their x and y coordinates. We then extracted all cells' meta.data and coordinate information. For each orig.ident, we iterated through the target cells, identifying neighboring cells within their vicinity (excluding the central point) and collecting their cell IDs into final_cells. After removing duplicates, we added the target cells to final_cells as well. Finally, we subsetted the original Seurat object using subset(), extracting the target cell population and its neighboring cells to generate the final_seurat object.

Distance-Based FGC depth calculation. Consecutive sections of the gonads were prepared, and central section containing intact medulla and continuous cortical layers were selected as samples for spatial transcriptomics sequencing. This selection criterion ensured that all samples were derived from a consistent anatomical context, specifically with regard to the relative proportions of cortex and medulla. To quantify FGC depth, we established a measurement based on Euclidean distance. The outermost pixels of the female gonad were manually labeled as reference points, and for each pixel within the tissue, the minimum distance to the reference point was calculated. The depth of each pixel within the layer was determined by its relative position between boundaries. Specifically, depth was calculated as: $Depth_i = dist_i/D$, where dist_i represents the minimum distance from a given pixel to the reference point, and D denotes the current diameter of the female gonad.

Immunofluorescence staining. Cryosections of OCT-embedded porcine female gonadal tissues were prepared at 8-10μm thickness using a cryostat at −20 °C. Sections were air-dried at room temperature (RT) for 20 min, followed by fixation in 4% paraformaldehyde (PFA) for 15 min at RT. After three 5-min PBS washes, tissues were permeabilized with 0.5% Triton X-100 in PBS for 15 min. Following another PBS wash cycle, sections were blocked with 3% BSA and 10% donkey serum in PBS for 1.5 h. Primary antibodies (see Table 1) were applied in a humidified

**Table 1 | List of antibodies and RNA-scope probes used in this study**

| Reagent or resource | Source | Identifier |
|---|---|---|
| Rat polyclonal anti-BLIMP1; dilution 1:300 | Thermo Fisher Scientific | 14-5963-80 |
| Rabbit monoclonal anti-DAZL; dilution 1:300 | abcam | ab215718 |
| Rabbit polyclonal anti-DDX4; dilution 1:200 | abcam | ab13840 |
| Mouse monoclonal anti-γH2AX; dilution 1:200 | abcam | ab26350 |
| Mouse monoclonal anti-5mC; dilution 1:500 | Active Motif | 39649 |
| Rabbit polyclonal anti-5hmC; dilution 1:500 | Active Motif | 39791 |
| Rabbit polyclonal anti-H3K27me3; dilution 1:300 | Active Motif | 39155 |
| Rabbit polyclonal anti-KRT19; dilution 1:200 | Proteintech | 10712-1-AP |
| Rabbit polyclonal anti-WT1; dilution 1:200 | Proteintech | 12609-1-AP |
| Mouse monoclonal anti-GATA4; dilution 1:300 | Santa Cruz Biotechnology | sc-25310 |
| Rabbit polyclonal anti-CD31; dilution 1:300 | abcam | ab28364 |
| Rabbit polyclonal anti-LAMA1; dilution 1:500 | Sigma-Aldrich | L9393 |
| Rabbit polyclonal anti-PDGFRA; dilution1:200 | Absin | abs146265 |
| Rabbit polyclonal anti-CTNNB1; dilution 1:300 | abcam | ab6302 |
| Rabbit polyclonal anti-Phospho-SMAD1/5; dilution 1:100 | Cell Signaling Technology | 9516T |
| Rabbit polyclonal anti-c-KIT; dilution 1:300 | abcam | ab32363 |
| RNAscope Probe -Ss-LHX9-C1 (Target region: 201-1706) (Accession Number: XM_005667985.3) | ACDBio | 1807211-C1 |
| RNAscope Probe -Ss-WNT6-C2 (Target region: 63-1552) (Accession Number: XM_021075516.1) | ACDBio | 1573061-C2 |
| RNAscope Probe -Ss-MAN1A1-C3 (Target region: 1090 - 2188) (Accession Number: NM_213885.1) | ACDBio | 1807231-C3 |
| RNAscope Probe -Ss-RARB-C2 (Target region: 976 - 2392) (Accession Number: XM_005669304.3) | ACDBio | 1807241-C2 |
| TSA Vivid Fluorophore 520; dilution 1:1000 | ACDBio | PG-323271 |
| TSA Vivid Fluorophore 570; dilution 1:1000 | ACDBio | PG-323272 |
| TSA Vivid Fluorophore 650; dilution 1:1000 | ACDBio | PG-323273 |
| Donkey anti-Rat IgG (H+L) Highly Cross-Adsorbed Secondary Antibody, Alexa Fluor 488; dilution 1:1000 | Thermo Fisher Scientific | A-21208 |
| Donkey anti-Mouse IgG (H+L) Highly Cross-Adsorbed Secondary Antibody, Alexa Fluor 488; dilution 1:1000 | Thermo Fisher Scientific | A-21202 |
| Donkey anti-Mouse IgG (H+L) Highly Cross-Adsorbed Secondary Antibody, Alexa Fluor 594; dilution 1:1000 | Thermo Fisher Scientific | A-21203 |
| Donkey anti-Rabbit IgG (H+L) Highly Cross-Adsorbed Secondary Antibody, Alexa Fluor 488; dilution 1:1000 | Thermo Fisher Scientific | A-21206 |
| Donkey anti-Rabbit IgG (H+L) Highly Cross-Adsorbed Secondary Antibody, Alexa Fluor 594; dilution 1:1000 | Thermo Fisher Scientific | A-21207 |
| RNAscope Multiplex Fluorescent Reagent Kit v2 | ACDBio | 323100 |
| RNA-Protein Co-Detection Ancillary Kit | ACDBio | 323180 |

chamber at 4 °C overnight. Post-incubation, sections underwent three 5-min PBS washes before incubation with fluorophore-conjugated secondary antibodies (Alexa Fluor 488, 594; Invitrogen) for 40 min at RT. Slides were mounted with Fluoroshield containing DAPI (Sigma) and sealed with nail polish (Table 1).

**RNA-FISH.** RNA-FISH was performed on optimal cutting temperature (OCT)-embedded 8–10 μm gonad sections using the RNA Scope Multiplex Fluorescent V2 Assay kit (ACDBio 323100) according to the manufacturer's protocol (Advanced Cell Diagnostics). Gonad sections were fixed in 4% paraformaldehyde (PFA) for 15 min and dehydrated through a graded ethanol series (50%, 70%, 100%, 100%). Endogenous peroxidase activity was quenched with 3% hydrogen peroxide for 10 min at room temperature, followed by enzymatic digestion with RNA Scope Protease IV for 30 min. Hybridization was conducted with custom-designed probes targeting specific markers in female gonadal somatic cells: Set 1: Pig_LHX9 (C1, 1807211) and Pig_WNT6 (C2, 1573061); Set 2: Pig_RARB (C2, 1807241) and Pig_MAN1A1 (C3, 1807231). Probe hybridization occurred at 40 °C for 2 h, followed by signal amplification using the RNA Scope TSA diluent system. TSA dyes (PG-323271-520, PG-323272-570, PG-323273-690) were diluted 1:1000 in amplification buffer. The positive and negative control probes were used as a control to test for RNA quality and unanticipated background, respectively (see Table 1 for probe target details).

**Combined application of RNA-FISH and immunofluorescence.** Gonad sections were fixed in 4% PFA for 15 min and dehydrated using 50%, 70%, 100% and 100% ethyl alcohol solutions. Endogenous peroxidase activity was quenched with hydrogen peroxide for 10 min at room temperature followed by using RNA Scope hydrogen peroxide solution for 10 min at room temperature. Sections were incubated with the desired primary antibody at 4 °C overnight in a humidified chamber. After washing PBS-T twice, fix it in 10% formalin for 30 min, and then wash it again twice. After treatment with protease IV enzyme, complete the RNA-FISH procedure as described above, and then perform secondary antibody treatment at room temperature for 30 min.

**Combined application of RNA-FISH and immunofluorescence.** Gonad sections were fixed in 4% PFA for 15 min and dehydrated using 50%, 70%, 100% and 100% ethyl alcohol solutions. Endogenous peroxidase activity was quenched with hydrogen peroxide for 10 min at room temperature followed by using RNA Scope hydrogen peroxide solution for 10 min at room temperature. Sections were incubated with the desired primary antibody at 4 °C overnight in a humidified chamber. After washing PBS-T twice, fix it in 10% formalin for 30 min, and then wash it again twice. After treatment with protease IV enzyme, complete the RNA-FISH procedure as described above, and then perform secondary antibody treatment at room temperature for 30 min.

## Statistics and reproducibility

Differential expression genes (DEGs) for each cell type under different conditions were identified using the FindMarkers function in Seurat, which employs a built-in Wilcoxon test with Benjamini–Hochberg correction for p-values. Wilcoxon rank-sum test was used to analyze gene set functional scores and the statistical relationship between germ cells and the distance to the outermost cortex. For spatial transcriptomic data, 2–3 biological replicates were conducted for each developmental stage. All immuno-fluorescence staining and RNA-FISH experiments were independently repeated at least three times using biologically independent samples. Representative images shown in the figures were confirmed across all replicates. Data are presented as significance levels set at $*adjp < 0.05$; $**adjp < 0.01$; $***adjp < 0.001$; $****adjp < 0.0001$; ns not significant, $adjP \geq 0.05$.

## Reporting summary

Further information on research design is available in the Nature Portfolio Reporting Summary linked to this article.

## Data availability

All data from this study have been deposited to GSA with the accession number CRA023499. Raw image files supporting the findings of this study have been deposited to Figshare with the (https://doi.org/10.6084/m9.figshare.31563142). All numerical source data for the charts and graphs in the main figures and Supplementary Figs. directly supporting the findings of this study are included in the Supplementary Data files, with the detailed correspondence as follows: Supplementary Data 1 provides the source data for Fig. 1; Supplementary Data 2 provides the source data for Figs. 2c, 3c, and Supplementary Fig. 2b, c; Supplementary Data 3 provides the source data for Supplementary Fig. 2f; Supplementary Data 4 provides the source data for Supplementary Fig. 4e, f; Supplementary Data 5 provides the source data for Fig. 4b, g, and Supplementary Fig. 5b, c; Supplementary Data 6 provides the source data for Fig. 6, Supplementary Figs. 7 and 8. All other data supporting the findings of this study are available from the corresponding author upon reasonable request.

## Code availability

The pipeline code generated to perform the analysis in this study is available at GitHub (https://github.com/Prince-Xiaaa/pig-fetal-ovary-st-code) and on Zenodo (https://doi.org/10.5281/zenodo.18897620).

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

## Acknowledgements

We thank the State Key Laboratory of Animal Biotech Breeding for the equipment and experimental platform. This work was supported by the National Key R&D Program of China (2022YFD1302201), the Biological Breeding-National Science and Technology Major Project (2023ZD0407503), the Natural Science Foundation of China (32302746, 32370846), the fellowship of China Postdoctoral Science Foundation (2023M743802), Pinduoduo-China Agricultural University Research Fund (PC2024A01001), Hebei Province Science and Technology Plan (23227602Z), and Shijiazhuang City Science and Technology Project (221790262A), the 2115 Talent Development Program of China Agricultural University and High-performance Computing Platform of China Agricultural University.

## Author contributions

J.H., T.C., and Y.K. conceptualized this project and supervised the overall experiments. P.H., T.C., and Y.Y. collected pig embryos and gonads. P.H. prepared the cryosectioned samples. W.X. and T.Z. completed all spatial transcriptomic library construction, performed data analysis of spatial transcriptomics sequencing results, and wrote the experimental methods and analytical protocols for spatial transcriptomics. G.D. and W.X. are responsible for the bioinformatics methodology. P.H., Y.Y., Z.F., and Y.T. performed immunofluorescence (IF) and RNA fluorescence in situ hybridization (RNA-FISH) to validate molecular characterization. X.C. drew the schematic diagram. R.L. participated in the bioinformatics analysis. P.H., Y.Y., W.M., and Y.T. arranged the layout of the images. J.H., Y.K., P.H., T.C., and W.X. performed manuscript writing, review, and editing.

## Competing interests

The authors declare no competing interests.
