## [Transparent Peer Review file · Communications Biology]

A spatiotemporal transcriptomic atlas of porcine (*Sus scrofa*) female early gonadal development

Corresponding Author: Professor Jianyong Han

Version 0:

Reviewer comments:

Reviewer #1

(Remarks to the Author)

In the present study, He and colleagues employ high-definition spatial RNA sequencing (hdST-seq) to investigate the molecular events at the single-cell level occurring during porcine ovarian development. Specifically, they examine five developmental stages, ranging from the newly differentiated ovary to the onset of meiosis in oocytes. They identify 11 distinct cell types and analyze the changes in both their spatial organization and transcriptomic profiles across these stages. They perform a detailed characterization of the dynamic changes in germ cells, supporting cells, and other cell populations, including endothelial and interstitial cells. Finally, they perform a ligand–receptor interaction analysis between supporting cells and germ cells.

General Comments

Although the data presented appear to be of considerable interest, they need to be presented more accurately. Many of the authors' claims cannot be verified by examining the figures. Some figures are not referenced in the main text, and the figure layout often does not follow the order in which they are cited. The authors provide 22 supplementary tables, but there is no reference to them in the text. Specific examples are described below.

In the Results section, the authors focus exclusively on well-known and general aspects of early ovarian development. However, the strength of this type of study lies in its hypothesis-free design. Such an approach has the potential to uncover previously unrecognized features—particularly at the level of cell types and their single-cell signatures—that deviate from current knowledge, even if only slightly. The authors are encouraged to not only highlight the findings that align with existing knowledge, but also to emphasize aspects in which porcine ovarian development diverges from what has been described in other species.

This also applies to the Discussion section. The authors' findings should serve as the central foundation from which to draw conclusions that underscore the novelty and significance of the study. Instead, the authors present a series of paragraphs that could have been written independently of the data presented here. Their results—both spatial and transcriptomic—should be the starting point for drawing conclusions about the genetic pathways operating in the pig, and how these may diverge from those in other species. Specific examples are described below.

Specific comments

Results section

Lines 103-104

“cluster 6 corresponds to female fetal germ cells (FGCs) expressing canonical markers (KIT, BLIMP1 and DDX4)” (Fig. 1c-e; Supplementary Fig. 1d)”

BLIMP1 is not shown in Fig. 1c-e; Supplementary Fig. 1d.

Lines 103-106

“cluster 6 corresponds to female fetal germ cells (FGCs) expressing canonical markers (KIT, BLIMP1 and DDX4), initially exhibited a dispersed distribution but progressively migrated toward cortical regions during gonadal development (Fig. 1c-e; Supplementary Fig. 1d)”

The different cell types are not clearly distinguishable in Figure 1e. The overlap of colors makes it difficult to observe changes in the spatial distribution of individual cell types. While Figure 1e is worth keeping, the authors should consider including supplementary figures in which only the relevant cell type is highlighted. Alternatively, if such figures are presented later in the manuscript, they should be explicitly referenced here.

Lines 110-111

“Cluster 1, characterized by ovarian surface epithelium cells (OSECs) markers (LHX9, KRT19), was predominantly localized to the outermost cortical region throughout development, with a minor population migrating toward the medullary region by E50”

At E50, there appear to be many red cells, corresponding to OSECs, in the central region of the gonad. Is this correct? If so, and if this represents a species-specific feature in pigs compared to other species, it would be advisable to mention it explicitly.

Lines 117-118

Together, three heterogeneous cell types, OSECs, Pre-GCs I, and Pre-GCs II, were detected in the early gonads, exhibiting a stratified distribution pattern.

The authors should improve the description. Specifically, it should be clarified which region is central, which is cortical, which cell types occupy the largest area, etc

Figure 2b is not cited in the text. Supplementary Fig. 2e is cited earlier than the other panels of Supplementary Fig. 2.

Line 153

“oogonial” should be oogonia.

Lines 153-156

“Differential expression analysis identified 1,313, 62, and 1,052 genes uniquely enriched in each cluster, respectively (Supplementary Fig. 2a). Gene Ontology (GO) enrichment confirmed Meiotic Oogonia cluster specificity in meiotic nuclear division and meiotic cell cycle processes³⁰(Supplementary Fig. 2b)”
Please, reference tables with these data.

lines 157-158

“To evaluate meiotic initiation during gonadal development, we next performed immunofluorescence at E35 and E50 for the leptotene-stage meiotic germ-cell marker γ H2AX.”
In Figure 2e the staining is presented as a double immunofluorescence for γ H2AX and DAZL. It would be helpful to clarify in the text that DAZL is included in the assay,

Supp Fig 3 (Lines 164-168)

The Supp Fig 3 could benefit from several improvements to enhance clarity and interpretation. Firstly, the panel labeling and citation order are inconsistent — for example, panels b and c (which include more complex labeling) are cited before panel a (simpler labeling). Reordering or relabeling would improve the narrative flow. Additionally, in panels b, c, and e, the co-expression of markers is difficult to appreciate. This is particularly evident in panel c (E50), where high cell density and signal overlap hinder interpretation. To address this, the authors should consider including a zoomed-in view of the region of interest and displaying channels with DAPI separately to better visualize marker co-localization.

Lines 170-172

“Alongside low 5mC and 5hmC levels, TET1 and TET3 expression declined, while the base excision repair (BER) pathway was upregulated, suggesting that active DNA demethylation had been completed, yet epigenetic reprogramming persisted via the BER pathway”

Inspection of Supp Fig 3a indicate that neither TET3 nor PARP1 is appreciably altered. To substantiate their claims the author are encouraged to reference the quantitative data already provided in the supplementary tables—specifically the log₂ fold-change (logFC) and adjusted P-value (padj) for each gene discussed. Presenting these values in a concise summary table or figure will (i) demonstrate the magnitude of change for TET1, TET3, and individual BER genes, and (ii) clarify which components of the pathway are truly differentially expressed.

Lines 173.174

“H3K27me3 exhibited high levels in mitotic PGCs at E24–E27 but sharply declined by E30, becoming undetectable by E35 and E50 (Supplementary Fig. 3e).”
Whereas Fig. 2b identifies germ cells with DAZL and BLIMP1, the supplementary figure switches to DDX4 without

explaining why this marker is preferred or describing its spatiotemporal pattern, leaving it unclear whether the same cell population is being tracked. The figure also contains unexplained arrows, omits the identity of the markers used at E35, and is shown at such low magnification that one cannot judge whether every DDX4-positive cell is truly H3K27me3-negative—particularly at E35 and E50, where red and green signals appear to overlap.

Lines 174-177

The authors state that additional chromatin regulators exhibited dynamic expression changes, citing upregulation of RBBP5, BRDT, and HLTf (Supplementary Fig. 2d; Fig. 3d). However, the reference to Supplementary Fig. 2d is vague and not clearly integrated into the interpretation—its relevance is not explained, and key patterns are left for the reader to infer. This figure should be repositioned as Supplementary Fig. 3 for consistency with the narrative. Furthermore, the authors should go beyond the general statement of “dynamic changes” and provide more specific examples to clarify the trends observed. For instance, within the PRC2 complex, RBBP7 appears to increase while EZH2 decreases—are these changes consistent with known regulatory dynamics? Similarly, HLTf and SMARCA4 show divergent expression patterns. A clearer interpretation of such differences is needed. The authors are encouraged to perform a gene expression differential analysis to determine whether the chromatin regulators as a group show statistically significant coordinated changes. This would strengthen the biological relevance of the observations and move the interpretation beyond isolated gene-level comments.

Figure 2g is mentioned before 2f

Figure 3b (lines 203-206)

It is advisable to clearly distinguish between the gonad and the mesonephros, perhaps by using dashed lines. Additionally, orienting all gonads consistently across panels would improve clarity and facilitate comparison.

Line 201-202

“WT1 (Wilms tumor 1 protein), key markers of the coelomic epithelium, which revealed their predominant localization in the gonadal region with minimal mesonephric expression”

In mice, WT1 is also expressed in the mesonephros (Liu/Yao 2016, Sasaki/Saitou 2021, Mayere/Nef 2022). Is different in pigs. This is important to clarify, as several conclusion depends on the expression of this marker.

Lines 208-210

The authors report co-expression of KRT19 and GATA4 localized to the cortical region of fetal gonads, with RNA-FISH confirming GATA4 expression in WNT6⁺ pre-granulosa cells at E50 (Fig. 3c, e). However, the images do not clearly delineate the cortical region, making it difficult to assess the spatial context of the staining. Additionally, the biological significance of the double staining—KRT19 and WNT6—should be explained to clarify the developmental relationship and functional implications of these markers during gonadal differentiation.

Lines 214-220

“To investigate the heterogeneity among OSECs, Pre-GCs I, and Pre-GCs II, we performed comparative analysis of their transcriptional profiles”
Where are shown the results of such analyses (Supp Table ??)

Lines 214-218

“This analysis indicated dynamic gene expression gradients across these cell types: Pre-GCs II exhibited downregulation of canonical OSECs markers (LHX9, KRT19) while upregulating TOX3, whereas OSECs maintained high LHX9 and KRT19 expression with minimal TOX3 expression¹⁵. Pre-GCs I showed a progressive decline in LHX9 and KRT19 while acquiring granulosa lineage markers such as WNT6.”

The paragraph should be revised to clearly explain the rationale behind selecting these markers and the biological implications of their expression patterns. Specifically, the authors should clarify why LHX9 and KRT19 are considered canonical OSEC markers and how their downregulation in Pre-GCs II reflects a shift away from the OSEC identity. Similarly, the upregulation of TOX3 and acquisition of granulosa lineage markers like WNT6 in Pre-GCs I should be linked to the progression toward granulosa cell differentiation.

Figure 1d is mentioned after figure 1f.

Lines 231-233

The statement that “WNT signaling is a pivotal regulator of supporting cell lineage specification during early differentiation” is well established in other species. The authors should explicitly acknowledge this prior knowledge and clarify that their results demonstrate that the same regulatory role of WNT signaling is conserved in the pig model.

Supplementary figure 4 is not mentioned in the text

Supplementary Figure 4a is referenced after the entire figure description

Line 250

“To validate these spatial relationships, we conducted RNA-FISH targeting the mesonephric marker RARB and interstitial marker MAN1A1.”

The authors should clarify the biological roles of RARB and MAN1A1 and provide relevant references to support their selection as markers for the mesonephric and interstitial cell populations, respectively.

Line 252-255

“RNA-FISH revealed differential localization patterns, with RARB predominantly localized to the mesonephros region, while MAN1A1 expression was restricted to the adjacent interstitial region. These results indicated stratified distribution patterns of

multiple cell types at the early gonad-mesonephros interface (Fig. 4b, c)”

The figure would benefit from clearer annotation: please label the gonad and mesonephros regions to help interpret the spatial expression patterns. The purpose of the dashed square is unclear—if it indicates a zoomed area, this should be clearly shown; otherwise, consider removing it. Finally, please elaborate briefly on the biological implications of these differential expression patterns beyond their distinct localization.

Lines 255_258

“transcriptomic analysis identified several genes (EBF1, ITGA9, HBE1, and NAV3) commonly expressed across endothelial cells, interstitial cells, and mesonephric, indicating shared transcriptional features (Fig. 4f)”

Please clarify the biological relevance of the shared gene expression—do these findings suggest common function, origin, or interaction between the cell types?

Supp 5 b mentioned after c.

fig 4e mentioned after 4g.

Lines 309-314

“Between E24 and E35, Mitotic PGCs are predominantly enveloped by supporting cells, while interstitial cells, endothelial cells, and stromal progenitor cells show minimal involvement (Fig. 5a, b; Supplementary Fig. 6a). In E30–E50 gonads, the proportion of supporting cells surrounding RA Oogonia decreases, potentially due to the substantial expansion of FGCs during this phase. By E50, Meiotic Oogonia are once again primarily surrounded by supporting cells (Fig. 5a, b; Supplementary Fig. 6a, c).”

The current images make it difficult to distinguish cell types, as the colored dots lack clear contrast. Please provide a simplified image with two clearly distinguishable colors highlighting only FGCs and supporting cells, excluding other germ cells. A separate image showing the remaining somatic cell types would also improve clarity. In this context, Supplementary Fig. 6a appears redundant and could be removed.

Fig 6 e

A higher-magnification image is needed to clearly show protein co-expression, displaying only the relevant channels. Additionally, please clarify why the LAMA1 expression pattern appears markedly different between Figs. 6d and 6e, despite both corresponding to E50 samples.

Line 334-335

“Meanwhile, we also observed that LAMA1 was expressed in various cell types (Supplementary Fig. 6d). “

Figure 6e does not allow the reader to evaluate expression intensity and cell-type identity simultaneously. A higher-magnification inset of the dashed-square region for expression levels may help.

Line 338-340

“Overall, these results collectively underscore the central role of supporting cells in oogonia specification, with hormonal signaling and extracellular matrix remodeling likely contributing to this process.”

This conclusion is overly generic and echoes long-standing literature rather than the author’s data. Please anchor the statement to the specific markers and experimental findings presented here—for example, detail how your supporting-cell signatures (e.g., X, Y) and ECM/hormonal readouts (e.g., Z) directly link to oogonia specification.

Line 347

(Fig. 6a; Supplementary Fig. 7d).

If panel 6d is the first cited, it should also be presented first in the figure to maintain a logical and clear order. Please reorder the panels accordingly.

Line 349-351

“We identified a significant interaction between WNT-ligand WNT3A, expressed by Pre-GCs II, and WNT receptors FZD3, expressed in Mitotic PGCs and RA Oogonia (Fig. 6a, b)”

WNT3A is not shown Fig 6b.

Lines 354-358

“High levels of membrane-associated CTNNB1 were detected in somatic cells and FGCs of gonads at various developmental stages. However, weak nuclear CTNNB1 was detected exclusively in RA oogonia, suggesting that the canonical WNT signaling mechanism may be transiently active only in RA Oogonia (Fig. 6d).”
Figure 6d does not allow evaluation of CTNNB1 localization; neither the cytoplasmic nor the nuclear signal is discernible. Please include higher-magnification images of the same region, add a DAPI channel, and provide clear overlays so that nuclear versus cytoplasmic CTNNB1 can be unambiguously assessed.

Line 365-368

“Meiotic oogonia exhibited significantly upregulated ID gene expression relative to RA oogonia and Mitotic PGCs, suggesting an important role for BMP signaling in the PGC-to-oogonia transition”
Please, explain the role of ID genes (with references) in this process.

Line 359-365

The BMP signaling pathway not only regulates PGCs specification but has also been shown in recent studies to play a pivotal role in oogonia specification⁵⁸. We noticed that BMP ligand-receptor interactions between FGC subtypes and surrounding Pre-GCs II revealed progressively increasing BMP ligand (BMP2, GDF5) activity from E24 to E50. At E50, the ligand BMP2 from Pre-GCs II exhibited significant interactions with RA Oogonia and Meiotic Oogonia, whereas GDF5 functioned exclusively in Meiotic Oogonia (Fig. 6a).

Figure 6a is too small to resolve or compare the reported BMP ligand-receptor interactions. Please supply (i) a higher-magnification supplementary panel that clearly shows these contacts and (ii) a separate figure that tracks the interaction within a single cell type across the developmental time points.

Line 368-372

Since a hallmark of BMP pathway activation is the nuclear translocation of phosphorylated (p) SMAD proteins, we assessed pSMAD1/5 localization in fetal gonads. At E24, no pSMAD1/5 expression was detected in BLIMP1+ cells; at E35, pSMAD1/5 was exclusively localized to the cytoplasm of DDX4+ cells; whereas by E50, distinct nuclear expression was observed in DDX4+ cells⁶⁰ (Fig. 6e; Supplementary Fig. 7e)

The present images do not permit reliable assessment of pSMAD1/5 subcellular localization—E35 and E50 panels appear essentially identical. Please supply higher-magnification images with a DAPI counterstain so that nuclear versus cytoplasmic pSMAD1/5 can be clearly distinguished.

Line 414 and FIGLA-positive oocytes are positioned within
414the medulla^{3, 21, 38} NO encuentro donde fue estudiado figla

Please explain what are FIGLA-positive oocytes. Please provide reference.

Discussion

Lines 415-418

At E50, we observed female germline yst formation, similar to mouse ovarian cysts containing ~6 oocytes supported by 24 nurse cells, suggesting that cortical expansion of porcine germ cells ensures a sufficient pool for subsequent oogenesis. Where is this shown?

Lines 439-453

This paragraph should be grounded in the author's own results. Please revise it to explicitly reference data generated in your study that support these conclusions; if your experiments do not provide such evidence, the paragraph should be removed. This is extensive to other paragraphs of the discussion.

Methods Line 46-52

PDMS microfluidic chip: Which manufacturer? Please describe in more detail. If no commercial brand is used, the origin should be specified.

well-specific barcode. ¿Which manufacturer? Please describe in more detail. If no commercial brand is used, the origin should be specified.

Line 532-533 Please change "Statistically analyze" for "Statistical analysis"

Reviewer #2

(Remarks to the Author)

Dear authors,

This paper provided an interesting dataset of spatial transcriptomics with relatively low-resolution regarding bin size and molecular signature of pig fetal female gonads during a period of development (between E24-E50). The data has shortcomings, but it is important and unique to have a broader picture of female gonadogenesis in mammals.

The clusters provided are general, the genes chosen rather standard (any new things compared to human or mouse?), the clusters are not well defined and restricted (because the resolution bins are relatively large and so each bin is a mixture of cells/signatures). For example, the OSE cluster is clearly broader than just OSE. It further provides standard ligand-receptor analysis and spatial neighbourhood analysis, which is adequate. The included validation using immunofluorescence (IF) is valuable, but the images are of low quality morphology (do you have access to paraffin sections?).

The analysis is standard, but the dataset is worth providing as resource, whereas the interpretation of the data needs some improvement and claims need to be refined.

Point of improvement:

1. The evidence for preGCs I and II is not robust. Could the authors provide better validation and higher magnification/better quality IF to separate between the two populations? In particular there are no germ cells in the spatial location of preGC I, these cells seem rather stromal to me. What would preGC I be in the middle of the gonad far away from germ cells? Could this cluster currently include several different cell types?
2. Why are OSE cells inside the E50 gonad (Fig S4A)? I think the OSE clusters should be further subclustered to understand what those cells are.
3. Having cells expressing similar genes does not mean they share a common origin. Hence the claim that interstitial cells are derived from the mesonephros is not well evidenced. Can the authors provide more convincing data for common origin or perhaps tone down that claim.
4. Comparison with humans should be more detailed regarding similarities and specifically differences between the different compartments/cell types.
5. Cluster 1 in the DEG table has a low quality set of genes (lots of RPL genes). Are you sure this cluster does not represent low quality cells? I am a bit concerned about the thresholds used for filtering the data. The DEGS from several clusters are unexpected and the usual suspects are absent. Can the authors mention this as limitation of the technology used?
6. I think the resolution (bin-size) is a severe limitation and this should also be explicit in the text.
7. Title: remove 'high definition'

Reviewer #3

(Remarks to the Author)

Oogenesis proceeds in the ovary through the interaction of various cell types. Cell-cell communication between adjacent cells is essential for cell differentiation. In this study, the authors present a spatial transcriptomic dataset that covers the early stage of ovarian development in a porcine fetus. The authors generated a dataset capturing the development of several ovarian cells, including oocytes, supporting cells, endothelial cells, and a part of mesonephric cells. They also demonstrate cell-to-cell communication between supporting cells and oocytes. This dataset is informative and useful resource for the field of reproductive biology. Therefore, it is important that the data are shared publicly. However, the current version of the manuscript needs to be restructured to make data easier to follow and interpret.

Major points to address

1. The authors demonstrated the differentiation and spatial distribution of ovarian cells. However, it is unclear whether this is completely novel or if related findings have been reported before. Please include the information about the knowledge shown in mouse and explain the similarities and differences. This will highlight the significance of this paper and relationship in relation to previous studies.
2. Based on Fig. 1e, the authors discuss the migration of several cell types. However, it is difficult for this reviewer to understand the authors' argument only on images of whole region. Could the authors provide magnified or cell-type specific images to make the data easier to interpret?
3. Even though the authors have the spatial transcriptomics dataset, visualizing gene expression levels spatially is limited. In the mouse, it is well known that the onset of meiosis exhibits an anterior-posterior wave because the mesonephros is the major source of retinoic acid. Therefore, it is useful if the authors can show the such kind of transcriptomic shift. Additionally, discussion about the link between somatic cell differentiation and anterior-posterior wave is also interesting.

4. Based on Fig. 6 a and b, the authors discuss about the ligand-receptor expression and cell-to-cell communication, focusing mainly on the effect of ligand from supporting cells on oocytes. However, the effect of oocyte-derived ligand on supporting cells, such as BMP2 at E50, are also observed. The authors should discuss this and explain why the authors have ignored it to help readers more easily follow their logic.

5. Line 349-350, the authors discuss about the function of WNT3A in cell-to-cell communication between supporting cell and oocytes. The authors showed WNT6, WNT2B, FZD8, WNT5A, ROR2 and FZD3 violin plot but the violin plot for WNT3A is missing. Could the authors provide WNT3A violin plot?

6. Line 365-367, the authors demonstrated the upregulation of ID family genes in meiotic oogonia. In mouse, as reported in Miyauchi et al., EMBO (2017), Nagaoka et al., Science (2020) and others, BMP signals affect oocytes prior to meiotic initiation. Is this due to differences in species or problems in identifying the cell type? Please discuss about this. In addition, please provide the expression of ZGLP1 that is known as the critical downstream factor of BMP signal.

Minor points to address

1. Line 158, γ H2AX is not a specific marker for leptotene stage of meiotic cells, it also marks the zygotene stage of meiotic cells. Therefore, "leptotene stage of meiotic..." should be replaced with "leptotene and zygotene stage of meiotic..."
2. Line 180-192, the figures should be labelled in the order in which they appear in the main text.
3. There is no reference about the BMP function on oocyte differentiation. The authors should refer the paper such as Wu et al. PLoS Biology (2016), Miyauchi et al., EMBO (2017), Nagaoka et al., Science (2020) and Cheung et al., Development (2025).

Reviewer #4

(Remarks to the Author)

In this manuscript by He and colleagues, the authors produced a spatial transcriptome dataset of the developing pig ovary at high spatial and transcriptome resolution. This dataset spans multiple developmental timepoints from E24 to E50 that are relevant for ovary development, including PGC specification and entry of oogonia into meiosis. The authors also identified in a spatially resolved manner the two waves of pre-granulosa cells. With this dataset, the authors interrogated the dynamics of While the data are high quality and this manuscript promises to become an important resource for the field of ovarian biology, this remains a transcriptome dataset that does not provide functional evidence for the authors' sometimes lofty conclusions, and there are several instances where the manuscript could be improved.

Suggestions to authors

I have listed below specific suggestions and recommendations to improve the quality of the manuscript and to clarify the impact of the data presented.

Major comments / limitations

- In the abstract and manuscript the authors claim that they "clarified the roles of the two waves of pre-granulosa cells in regulating germ cell development" and that they "identified the mesonephros as the source of endothelial and interstitial cells along with their specific contributions to gonadal development." They further claim that "cell-cell communication analysis revealed the critical role of BMP signaling (BMP2, GDF5) in driving the specification of PGCs into oogonia and their subsequent entry into meiosis." I have several problems with these claims:

-- The data presented are descriptive, based on expression of specific transcripts (and some protein level validation) in wild-type pig ovaries. These data are merely correlations, and do not allow such strong conclusions on the role of specific cell populations or pathways in development of the ovary. I recommend the authors either provide functional evidence for these conclusions (using in vitro systems with small molecule inhibitors or cell ablation methods for example), or that they tone down their conclusions and clearly identify in the results and discussion that these data provide correlations that may indicate function, but this would need to be tested using functional experimental approaches. This should be addressed throughout the text. For example line 301 to 303, the summary suggests functional involvement of angiogenesis, ECM remodeling and myogenesis, but this was not functionally tested, just suggested by transcript expression. Same on line 338-340, with the additional caveat that this has already been shown in other animal models.

-- Some of these conclusions (2 waves of granulosa cells, mesonephric contribution to vasculature, role of BMP2 in oocyte development) have already been made using functional experiments in mouse ovaries. The fact that these same pathways and developmental events may be conserved between mouse and pig and function in similar ways is novel and exciting, and I would argue that the authors could do a better job recognizing these similarities and using them to make more solid conclusions.

- I did not see a materials and methods section in the manuscript, it seems all methods were pushed to supplemental. I find this extremely confusing and does not favor data accessibility and reproducibility. I recommend putting the materials and methods in the main manuscript.

What is the spatial resolution (size and spacing of spots)?

- Lines 93-95: "Following stringent quality control, 50,630 high-quality spatial pixels were retained for constructing the porcine female gonadal spatiotemporal transcriptomic atlas". The authors should provide more detail about the QC steps, what was defined as "high quality" pixels, what is the difference between the QC-filtered and the raw datasets?

- Lines 126-127 "both endothelial cells and interstitial cells shared a corticomedullary migration pattern". This conclusion seems like a stretch, how did the authors determine this was a migration pattern vs proliferation of endothelial and interstitial cells toward the medulla? These two possibilities should be addressed and/or discussed.
- Line 166, the authors investigate the methylation of DNA in germ cells. This experiment is interesting but could be better introduced. In its current form it is not immediately clear why this assay is relevant. Authors should also define 5mC and 5hmC. Similar comment for line 172, why is it relevant to investigate H3K27me3 marks?
- Line 187: how did authors calculate the distance between FGCs and the cortex, and how did they normalize for dorsal-ventral localization of the section plane. I.e. if the section is closer to the surface of the ovary, the distance will be smaller than if the section is taken on the dorsal side of the ovary which is mostly medulla.

Figures

- The authors state in line 93, that 2-3 biological replicates were sequenced, but they only show one section per stage. Were these integrated together on a single spatial map as shown in Fig. 1, or is fig 1e depicting a single replicate? I recommend clarifying this in the text and legend, and showing images of the transcriptome for all biological replicates. A discussion of the variability in spatial transcriptome between individuals would be interesting as well.
- For the spatial images in Fig. 1e, I recommend adding some indication of the orientation of the tissue (anterior / posterior pole, cortex vs medulla) to better orient the reader to these images.
- I recommend using colorblind-friendly color schemes in the IF images instead of red and green, to increase accessibility of the manuscript.
- A few typos: Fig 1a4, "library"; Fig1a3, chip orientation should say "B1-B96" and A1-A96; Fig 1c y-axis title, "fraction";
- Fig 2D and E, where in the tissue were these panels taken? I recommend showing the entire section in addition to the zoom up to show distribution of BLIMP1, DAZL, H2AX cells across the tissue. Same comment for Fig3c and fig6d

Secondary / minor comments

- Gene / protein annotation seems inconsistent. For example, line 147 IF for BLIMP1 and DAZL, should not be italicized since this is looking at protein.
- What method was used for pseudotime analysis? This should be mentioned in the results at line 149.
- What method was used for ligand-receptor analysis? This should be mentioned in the results at line 344.

Version 1:

Reviewer comments:

Reviewer #1

(Remarks to the Author)

The authors have successfully addressed the major concerns from the previous round, significantly improving the quality and scientific rigor of the study. The new comparative analyses and structural reorganization are commendable.

Minor issues:

Line 111: "These FGCs initially exhibited a dispersed distribution but progressively migrated toward cortical regions during gonadal development"

Comment: In the figures, the FGCs appear to be always located close to the cortical region. Please verify this description or the figures.

Line 117: "Spatial visualization revealed a stratified organization of clusters 2 and 3 relative to OSECs at E24–E27"

Comment: The term "stratified organization" is unclear in this context. Please describe better where and how this stratification occurs, as it is not immediately evident.

Line 119: "Subsequent characterization identified cluster 2 as pre-granulosa cells II (Pre-GCs II) through TOX3 expression, while cluster 3 was classified Pre-GCs I based on WNT6 expression."

Comment: There is a contradiction here. In other parts of the text/figures, Cluster 2 is identified as Pre-GCs I and Cluster 3 as Pre-GCs II. Please verify and correct the identification.

Line 121: "(Fig. 1d, e)."

Comment: Based on the context, this citation should likely be Fig. 1b-d.

Line 129: "...interstitial cells marker NR2F1 and PDGFRA, while clusters7"

Comment: Typo. It should read "cluster 7".

Line 185: "trimethylation of histone H3 at lysine 27 (H3K27me3), a repressive histone modification, produced a stronger signal in Mitotic PGCs but sharply declined by E30, becoming undetectable by E35 and E50"

Comment: Please cite the corresponding figure panel immediately after this sentence to support the statement.

Line 232: "GATA4 (a zinc finger transcription factor) and WT1 (Wilms tumor 1 protein), key markers of the coelomic epithelium."

Comment: In this context, it would be more precise to indicate that these are markers of the supporting cell lineage, rather than just the coelomic epithelium.

Line 266: "We observed a distinct stratified distribution pattern of these cells in E24 and E27."

Comment: Similar to the previous comment, please describe this pattern more clearly (where and how the stratification is observed).

Line 271: "The results showed that GATA4 (a coelomic epithelial marker) and KRT19 (a marker for OSECs and Pre-GCs II cells)"

Comment: This statement seems to contradict the figures (or standard literature). Please check if the markers or cell types are swapped, or provide references that support this specific co-expression/identity.

Line 279: "The results showed that the supporting cell lineages in female gonads of both pigs (E50) and humans (14 weeks) exhibit a highly conserved spatial distribution: OSECs are restricted to the outer cortical region, Pre-GCs II cells are distributed throughout the inner cortex, and the less abundant Pre-GCs I population resides specifically within the medullary region"

Comment: It is difficult to distinguish these specific regions in the human gonad images provided. Please improve the annotation or visualization to clearly differentiate the cortical vs. medullary regions.

Line 361: "Mitotic PGCs are predominantly surrounded by Pre-GCs I"

Comment (General Issue): The abbreviation "GC" is used confusingly for both Germ Cells and Granulosa Cells throughout the manuscript. For example, here "Pre-GCs" refers to granulosa, but elsewhere "GC" implies germ cells. The authors must strictly differentiate these abbreviations.

Line 389: "...which female germline cysts develop (Fig. 5e)"

Comment: This citation appears incorrect and likely refers to Fig. 5f. Additionally, Panel 5g is not mentioned in the text.

Supplementary Material: "Fig. 2g"

Comment: The text refers to a Figure 2g, but this panel appears to be missing.

Figure 6c (Upper):

Comment: In the merged panel, the purple and green channels appear to be swapped compared to the individual single-channel panels. Please verify.

Reviewer #2

(Remarks to the Author)

Dear authors,

The manuscript has gained in quality and clarity.

I now support publication.

Reviewer #3

(Remarks to the Author)

The authors have addressed the comments raised by this reviewer, making the current version of the manuscript is easier to follow. Only minor concerns remain as follow.

1. Line 180-181, is "DNA methylation (DNMT3A)" not typographical error for "DNA methyltransferase 3a (DNMT3A)"?

2. Line 413-415, the authors cannot not definitively state that these cells are RA oogonia based on these immunofluorescence images. Therefore, this reviewer suggests that the authors revise this point to refer to the embryonic day in order to ensure accuracy.

Reviewer #4

(Remarks to the Author)

The authors have done a very thorough job revising the manuscript and incorporating all of the reviewer comments. I have no further comments, the authors have addressed my concerns to satisfaction.

Version 2:

Reviewer comments:

Reviewer #1

(Remarks to the Author)

The authors have incorporated all my suggestions. The manuscript has improved considerably.

Point-by-point response to reviewers' comments:

Reviewer #1 (Remarks to the Author):

In the present study, He and colleagues employ high-definition spatial RNA sequencing (hdST-seq) to investigate the molecular events at the single-cell level occurring during porcine ovarian development. Specifically, they examine five developmental stages, ranging from the newly differentiated ovary to the onset of meiosis in oocytes. They identify 11 distinct cell types and analyze the changes in both their spatial organization and transcriptomic profiles across these stages. They perform a detailed characterization of the dynamic changes in germ cells, supporting cells, and other cell populations, including endothelial and interstitial cells. Finally, they perform a ligand–receptor interaction analysis between supporting cells and germ cells.

General Comments

Although the data presented appear to be of considerable interest, they need to be presented more accurately. Many of the authors' claims cannot be verified by examining the figures. Some figures are not referenced in the main text, and the figure layout often does not follow the order in which they are cited. The authors provide 22 supplementary tables, but there is no reference to them in the text. Specific examples are described below.

Response: We sincerely thank you for your insightful comments and highly constructive suggestions. We have systematically verified all figure and table citations to ensure that each key conclusion in the text explicitly references the corresponding figure, table, or supplementary material. In addition, we have rearranged the figures to follow the sequential order of their citation in the manuscript.

In the Results section, the authors focus exclusively on well-known and general aspects of early ovarian development. However, the strength of this type of study lies in its hypothesis-free design. Such an approach has the potential to uncover previously unrecognized features—particularly at the level of cell types and their single-cell signatures—that deviate from current knowledge, even if only slightly. The authors are encouraged to not only highlight the findings that align with existing

knowledge, but also to emphasize aspects in which porcine ovarian development diverges from what has been described in other species.

Response: We fully concur with this perspective. Accordingly, we performed a re-analysis of the published spatial transcriptomic data from human embryos at 14 PCW and compared it with the porcine data from this study. In the revised manuscript, particular emphasis was placed on highlighting the conserved spatial distribution patterns of germ and supporting cell lineages between humans and pigs, while also noting the differences in the distribution of Mitotic PGCs between the two species (**see details in lines 209-217 and 274-283 of the revised manuscript**).

This also applies to the Discussion section. The authors' findings should serve as the central foundation from which to draw conclusions that underscore the novelty and significance of the study. Instead, the authors present a series of paragraphs that could have been written independently of the data presented here. Their results—both spatial and transcriptomic—should be the starting point for drawing conclusions about the genetic pathways operating in the pig, and how these may diverge from those in other species. Specific examples are described below.

Response: We fully concur with your perspective. In response, we have removed the extraneous content related to the differentiation of pluripotent stem cells into gametes and refined the section of *Discussion* to more closely align with the core spatial transcriptomics data. Furthermore, we have emphasized the timing of BMP signaling downstream target *ZGLP1* during the specification of primordial germ cells into oogonia, noting the species-specific differences between mice and humans (**see details in lines 534-538 of the revised manuscript**).

Specific comments

Results section

Lines 103-104

“cluster 6 corresponds to female fetal germ cells (FGCs) expressing canonical markers (KIT, BLIMP1 and DDX4)” (Fig. 1c-e; Supplementary Fig. 1d)” **BLIMP1 is not shown in Fig. 1c-e; Supplementary Fig. 1d.**

Response: Thanks for your comment. *BLIMP1* (also known as *PRDM1*) is a marker gene

for early primordial germ cells (PGCs) and is expressed in early PGCs of mice¹, humans², and pigs³. Due to an oversight in the original figure, the gene “*BLIMP1*” was erroneously designated by its alternative name “*PRDM1*.” To maintain terminological accuracy and consistency with the text, the label has now been corrected to “*BLIMP1*.” We again extend our appreciation for bringing this to our attention (**see details in Fig. 1d and Fig. 2c of the revised manuscript**).

References:

1. Ohinata, Y. et al. A Signaling Principle for the Specification of the Germ Cell Lineage in Mice. *Cell* **137**, 571-584 (2009).
2. Irie, N. et al. SOX17 Is a Critical Specifier of Human Primordial Germ Cell Fate. *Cell* **160**, 253-268 (2015).
3. Zhu, Q. et al. Specification and epigenomic resetting of the pig germline exhibit conservation with the human lineage. *Cell Rep* **34**, 108735 (2021).

Lines 103-106

“cluster 6 corresponds to female fetal germ cells (FGCs) expressing canonical markers (KIT, BLIMP1 and DDX4), initially exhibited a dispersed distribution but progressively migrated toward cortical regions during gonadal development (Fig. 1c-e; Supplementary Fig. 1d)”

The different cell types are not clearly distinguishable in Figure 1e. The overlap of colors makes it difficult to observe changes in the spatial distribution of individual cell types. While Figure 1e is worth keeping, the authors should consider including supplementary figures in which only the relevant cell type is highlighted. Alternatively, if such figures are presented later in the manuscript, they should be explicitly referenced here.

Response: We sincerely appreciate your valuable suggestion. We fully agree that in the original Figure 1e, the overlapping colors of different cell types made it difficult to clearly track the spatiotemporal distribution of any single cell type. To thoroughly address this issue, we have refined the color scheme to enhance contrast. Additionally, magnified views along with annotations of the anterior and posterior, as well as the cortical and medullary

regions, have been included to better illustrate the spatial architecture (**Response Figure 1**). We believe these additions and refinements significantly improve the visual presentation and strengthen the support for our findings.

Response Figure 1: Spatial mapping of cell populations: Cellular distributions were reconstituted on the microarray coordinate system through HDst-seq barcode alignment. The dashed box indicates the enlarged view of the corresponding area. The outer cortical, inner cortical, and medullary regions of the gonads are delineated by dashed lines. The image shown is from a single representative sample (the replicate is presented in the **Supplementary Fig. 1e**). Scale bar: 500 μm (zoomed-out), 100 μm (zoomed-in) (as shown in **Fig. 1e** of the revised manuscript).

Lines 110-111

“Cluster 1, characterized by ovarian surface epithelium cells (OSECs) markers (LHX9, KRT19), was predominantly localized to the outermost cortical region throughout development, with a minor population migrating toward the medullary region by E50”

At E50, there appear to be many red cells, corresponding to OSECs, in the central region of the gonad. Is this correct? If so, and if this represents a species-specific feature in pigs compared to other species, it would be advisable to mention it explicitly.

Response: Thank you for your valuable feedback and insightful observation regarding the localization of Cluster 1 (OSECs) at E50. We agree that this point warrants careful consideration and verification. We have diligently revisited our clustering methodology and critically evaluated our approach to batch correction. We found that the initial clustering

using Harmony might have led to over-correction (overfitting), potentially obscuring genuine biological differences.

To address this, we performed a thorough comparison:

1. Clustering samples from different developmental stages independently.
2. Comparing various batch correction strategies.

We ultimately determined that using Reciprocal PCA (RPCA) for batch correction yielded the most robust and biologically faithful result. RPCA works by matching cells based on the principal components in a low-dimensional space, effectively mitigating the influence of high-dimensional noise. Crucially, compared to Harmony, RPCA is a gentler approach that is better at preserving genuine biological variance during the batch integration process. The resulting map, based on RPCA integration, now provides a more accurate and reliable representation of the cellular architecture. We have updated our figures and analysis accordingly (**see details in Fig. 1 b-e and Supplementary Fig. d, e of the revised manuscript**). We appreciate you raising this critical point, as it has allowed us to significantly improve the robustness and confidence in our final cellular map.

Lines 117-118

Together, three heterogeneous cell types, OSECs, Pre-GCs I, and Pre-GCs II, were detected in the early gonads, exhibiting a stratified distribution pattern.

The authors should improve the description. Specifically, it should be clarified which region is central, which is cortical, which cell types occupy the largest area, etc

Response: We fully endorse the suggestions provided. To help readers more accurately comprehend the spatial distribution patterns of OSECs, Pre-GC I, and Pre-GC II, we have carefully refined the wording to more clearly emphasize their distinct localization within the early gonadal stages.

The modification is as follows (see details in lines 121-125 of the revised manuscript):

Together, three heterogeneous cell types, OSECs, Pre-GCs I, and Pre-GCs II, were identified and arranged in a stratified organization extending from the outer cortical region to the inner central zone, where they predominantly reside. As development proceeds, Pre-GCs I gradually translocate to the medullary region.

Figure 2b is not cited in the text. Supplementary Fig. 2e is cited earlier than the other

panels of Supplementary Fig. 2.

Response: We have checked and reorganized the order in which the panels of Fig. 2 are cited in the main text. All panels (a, b, c, d, e) are now referenced in accordance with their logical sequence of appearance in the Results section, ensuring a coherent narrative flow. We have also verified that the panel order within the supplementary figure itself aligns with the citations in the text.

Line 153

“oogonial” should be oogonia.

Response: Thank you for your suggestion. We have revised the content accordingly to ensure logical coherence, as requested.

The modification is as follows (see details in lines 156-158 of the revised manuscript):

These results indicated that the developmental window spanning stages E24 to E50 encompasses the specification process from PGCs to oogonia.

Lines 153-156

“Differential expression analysis identified 1,313, 62, and 1,052 genes uniquely enriched in each cluster, respectively (Supplementary Fig. 2a). Gene Ontology (GO) enrichment confirmed Meiotic Oogonia cluster specificity in meiotic nuclear division and meiotic cell cycle processes³⁰(Supplementary Fig. 2b)”

Please, reference tables with these data.

Response: Thank you for your suggestion. We have revised the text to explicitly reference the **Supplementary tables 2**, which contain the detailed data on the number of differentially expressed genes and the results of the GO enrichment analysis (**see details in lines 158-162 of the revised manuscript**).

lines 157-158

“To evaluate meiotic initiation during gonadal development, we next performed immunofluorescence at E35 and E50 for the leptotene-stage meiotic germ-cell marker γ H2AX.”

In Figure 2e the staining is presented as a double immunofluorescence for γ H2AX and DAZL. It would be helpful to clarify in the text that DAZL is included in the assay,

Response: We fully agree with your suggestion.

The modification is as follows (see details in lines 163-166 of the revised manuscript):

To evaluate meiotic initiation during gonadal development, we next performed immunofluorescence at E35 and E50 for the oogonia marker DAZL as well as the leptotene and zygotene stage marker γ H2AX, respectively.

Supp Fig 3 (Lines 164-168)

The Supp Fig 3 could benefit from several improvements to enhance clarity and interpretation. **Firstly, the panel labeling and citation order are inconsistent — for example, panels b and c (which include more complex labeling) are cited before panel a (simpler labeling). Reordering or relabeling would improve the narrative flow.**

Response: Thank you very much for your thorough review of our manuscript and your highly constructive feedback regarding Supplementary Figure 3. We have rearranged the panel sequence to ensure a clearer narrative flow and full alignment with the order of citation in the main text.

Additionally, in panels b, c, and e, the co-expression of markers is difficult to appreciate. This is particularly evident in panel c (E50), where high cell density and signal overlap hinder interpretation. To address this, the authors should consider including a zoomed-in view of the region of interest and displaying channels with DAPI separately to better visualize marker co-localization.

Response: Regarding the difficulty in identifying co-expression markers: In panels with high cell density and significant signal overlap, we have included higher-magnification insets to allow more detailed observation of specific cellular regions (**Response Figure 2a-c**). Additionally, we now provide separate images for each fluorescence channel, including DAPI nuclear staining (**Response Figure 2a-c**). This will help readers clearly distinguish the specific localization of each marker and better assess co-expression patterns.

Response Figure 2: Epigenetic dynamics during pPGC development **a**, Immunofluorescence images of BLIMP1/DAZL (purple) and 5mC (green) in porcine female gonadal sections from different timepoints. Nuclei were counterstained with DAPI (blue). Scale bars, 50 μ m. **b**, Immunofluorescence images of BLIMP1/DDX4 (purple) and 5hmC (green) in porcine female gonadal sections from different timepoints. Nuclei were counterstained with DAPI (blue). Scale bars, 50 μ m. **c**, Immunofluorescence images of BLIMP1/DDX4 (purple) and H3K27me3 (green) in porcine female gonadal sections from different timepoints (E30, E35, and E50). Nuclei were counterstained with DAPI (blue). At E24, germ cells show co-localization of BLIMP1 and H3K27me3; at E30, only a very small number of DDX4-positive germ cells exhibit co-localization with H3K27me3; and at E35 and E50, no co-localization is observed between DDX4 and H3K27me3 in germ cells. Scale bars, 50 μ m (as shown in **Supplementary Fig. 2d, e** and **Supplementary Fig. 3a**).

Lines 170-172

“Alongside low 5mC and 5hmC levels, TET1 and TET3 expression declined, while the base

excision repair (BER) pathway was upregulated, suggesting that active DNA demethylation had been completed, yet epigenetic reprogramming persisted via the BER pathway”

Inspection of Supp Fig 3a indicate that neither TET3 nor PARP1 is appreciably altered. To substantiate the their claims the author are encouraged to reference the quantitative data already provided in the supplementary tables—specifically the log₂ fold-change (logFC) and adjusted P-value (padj) for each gene discussed. Presenting these values in tin a concise summary table or figure will (i) demonstrate the magnitude of change for TET1, TET3, and individual BER genes, and (ii) clarify which components of the pathway are truly differentially expressed.

Response: Thanks for your suggestions. Your point regarding the need to support our conclusions on *TET1/TET3* and BER pathway expression with quantitative data is particularly important and well taken. We have re-examined the statistical results in the Supplementary Tables, and the data show that although the downregulation of *TET3* is modest in magnitude (logFC = -0.319), it does reach statistical significance (padj = 3.5e-09). Therefore, in accordance with your feedback, we have maintained the original statement concerning *TET3* downregulation. Additionally, as suggested, we have introduced a new **Supplementary Table 3** that lists the complete logFC and adjusted p-values (padj) for all epigenetic related genes discussed, thereby providing direct quantitative support for this conclusion (**see details in lines 180-184 of the revised manuscript**).

Lines 173.174

“H3K27me3 exhibited high levels in mitotic PGCs at E24–E27 but sharply declined by E30, becoming undetectable by E35 and E50 (Supplementary Fig. 3e).”

Whereas Fig. 2b identifies germ cells with DAZL and BLIMP1, the supplementary figure switches to DDX4 without explaining why this marker is preferred or describing its spatiotemporal pattern, leaving it unclear whether the same cell population is being tracked.

Response: Thank you very much for your insightful and detailed comments. Due to the fact that DAZL and H3K27me3 antibodies are raised in the same host species, co-immunofluorescence staining is not amenable. Given that the progressive expression of

DAZL and *DDX4* serves as a pivotal indicator of early primordial germ cell differentiation into oogonia, both markers can be jointly utilized to identify oogonia at this developmental stage¹⁻³ (see details in Fig. 2c and Lines 144-150 of the revised manuscript). Therefore, we ultimately opted for co-staining of *DDX4* and H3K27me3 to investigate the dynamic changes in histone modifications during oogonial development.

The figure also contains unexplained arrows, omits the identity of the markers used at E35, and is shown at such low magnification that one cannot judge whether every *DDX4*-positive cell is truly H3K27me3-negative—particularly at E35 and E50, where red and green signals appear to overlap.

Response: To provide clearer observation of cellular localization, we have now included higher-magnification insets along with the individual DAPI channel. These new images clearly demonstrate a complete absence of H3K27me3 signal within the nuclei of individual *DDX4*-positive germ cells, while the surrounding somatic nuclei remain positive. This provides strong evidence that the loss of H3K27me3 is a germ cell-specific phenomenon. Regarding the meaning of the arrows in the figure, it has now been clearly clarified in the figure caption.

References:

1. Li, L. et al. Single-Cell RNA-Seq Analysis Maps Development of Human Germline Cells and Gonadal Niche Interactions. *Cell Stem Cell* **20**, 858-873.e854 (2017).
2. Saitou, M. & Hayashi, K. Mammalian in vitro gametogenesis. *Science* **374**, eaaz6830 (2021).
3. Zhu, Q. et al. Specification and epigenomic resetting of the pig germline exhibit conservation with the human lineage. *Cell Rep* **34**, 108735 (2021).

In the Supplementary Fig. 3a legend, the white arrows are described in detail as follows: At E24, germ cells show co-localization of *BLIMP1* and H3K27me3; at E30, only a very small number of *DDX4*-positive germ cells exhibit co-localization with H3K27me3; and at E35 and E50, no co-localization is observed between *DDX4* and H3K27me3 in germ cells (**Response Figure 3c**).

The authors state that additional chromatin regulators exhibited dynamic expression changes, citing upregulation of RBBP5, BRDT, and HLTF (Supplementary Fig. 2d; Fig. 3d). However, the reference to Supplementary Fig. 2d is vague and not clearly integrated into the interpretation—its relevance is not explained, and key patterns are left for the reader to infer. This figure should be repositioned as Supplementary Fig. 3 for consistency with the narrative. Furthermore, the authors should go beyond the general statement of “dynamic changes” and provide more specific examples to clarify the trends observed. For instance, within the PRC2 complex, RBBP7 appears to increase while EZH2 decreases—are these changes consistent with known regulatory dynamics? Similarly, HLTF and SMARCA4 show divergent expression patterns. A clearer interpretation of such differences is needed. The authors are encouraged to perform a gene expression differential analysis to determine whether the chromatin regulators as a group show statistically significant coordinated changes. This would strengthen the biological relevance of the observations and move the interpretation beyond isolated gene-level comments.

Response: Thank you for your valuable and insightful feedback. We acknowledge that the initial descriptions in this section may not have been sufficiently clear. To improve readability and ensure better understanding, we have carefully revised and refined the relevant content. According to published literature, the level of H3K27me3 gradually decreases during the differentiation of primordial germ cells (PGCs) into oogonia in both mice and humans^{1, 2}. In our revised manuscript, we have further supplemented the functional association of H3K27me3 with the Polycomb Repressive Complex 2 (PRC2) and the SWI/SNF chromatin remodeling complex. Among these, PRC2 is the key complex responsible for catalyzing the production of H3K27me3 and regulating gene expression, while the SWI/SNF complex functionally antagonizes this process and is involved in the regulation of corresponding gene expression^{3, 4}. Correspondingly, core components of PRC2 (such as *EZH2*, *SUZ12*, and *EED*) are predominantly expressed during early PGC stages, whereas members of the SWI/SNF complex (such as *SMARCA5* and *HLTF*) exhibit more prominent expression in oogonia upon entry into meiosis⁵.

The modification is as follows (see details in lines 184-194 of the revised manuscript):

Interestingly, the trimethylation of histone H3 at lysine 27 (H3K27me3), a repressive histone modification, produced a stronger signal in Mitotic PGCs but sharply declined by E30, becoming undetectable by E35 and E50. As evidenced by immunostaining results, the core components of the Polycomb Repressive Complex 2 (PRC2) members *EZH2*, *SUZ12*, and *EED* are highly expressed in early gonadal pPGCs. PRC2 catalyzes H3K27me3 and plays a key role in gene expression regulation. Furthermore, during the differentiation of PGCs into oogonia, the expression of subunits *SMARCA5* and *HLTF* from the SWI/SNF chromatin remodeling complex, which functionally antagonizes H3K27me3-mediated regulation, is gradually upregulated.

Additionally, as you suggested, we have now included a **Supplementary Table 3**, which provides the full list of logFC and adjusted p-values (padj) for the genes discussed, offering direct quantitative support for this conclusion.

References:

1. Hill, P.W.S. et al. Epigenetic reprogramming enables the transition from primordial germ cell to gonocyte. *Nature* **555**, 392-396 (2018).
2. Borensztein, M. et al. Contribution of epigenetic landscapes and transcription factors to X-chromosome reactivation in the inner cell mass. *Nat Commun* **8**, 1297 (2017).
3. Macrae, T.A., Fothergill-Robinson, J. & Ramalho-Santos, M. Regulation, functions and transmission of bivalent chromatin during mammalian development. *Nat Rev Mol Cell Biol* **24**, 6-26 (2023).
4. Cenik, B.K. & Shilatifard, A. COMPASS and SWI/SNF complexes in development and disease. *Nat Rev Genet* **22**, 38-58 (2021).
5. Zhu, Q. et al. Specification and epigenomic resetting of the pig germline exhibit conservation with the human lineage. *Cell Rep* **34**, 108735 (2021).

Figure 2g is mentioned before 2f

Response: We have checked and reorganized the order in which the panels of Fig. 2 are cited in the main text. All panels are now referenced in accordance with their logical sequence of appearance in the Results section, ensuring a coherent narrative flow. We

have also verified that the panel order within the supplementary figure itself aligns with the citations in the text.

Figure 3b (lines 203-206)

It is advisable to clearly distinguish between the gonad and the mesonephros, perhaps by using dashed lines. Additionally, orienting all gonads consistently across panels would improve clarity and facilitate comparison.

Response: Thank you very much for your valuable suggestions, which are helpful to improve the quality and clarity of the illustrations in our paper. Following your advice, we have used dashed lines to delineate the gonad or the mesonephros in all relevant schematic diagrams. This revision ensures clear visual distinction between the two key areas. Additionally, to better illustrate the methodology underlying our findings, we have incorporated anterior and posterior markers in the figures (**see details in Fig. 1e, Fig. 3a and Fig. 4c of the revised manuscript**).

Line 201-202

“WT1 (Wilms tumor 1 protein), key markers of the coelomic epithelium, which revealed their predominant localization in the gonadal region with minimal mesonephric expression”
In mice, WT1 is also expressed in the mesonephros (Liu/Yao 2016, Sasaki/Saitou 2021, Mayere/Nef 2022). Is different in pigs. This is important to clarify, as several conclusion depends on the expression of this marker.

Response: Thanks for your suggestion. We have carefully reviewed the cited literature and confirmed that, as you indicated, the WT1 is expressed in both the gonads and the mesonephros of mice. To directly address your question regarding species differences in WT1 expression, we conducted additional experiments involving detailed cryosectioning and immunofluorescence analysis of the gonadal-mesonephric complex in E24 pig embryos (**Response Figure 4a**).

We first observed that the porcine mesonephros exhibits significant morphological differences from that of the mouse, being larger in size and predominantly displaying tubular structures (**Response Figure 3a**). Notably, WT1 protein expression is not widespread throughout the mesonephric region but is restricted to isolated clusters in a central subregion; both the lateral mesonephric area and the region adjoining the gonad

are WT1-negative (**Response Figure 3b**). This pattern indicates a high degree of cellular heterogeneity within the porcine mesonephros. To further characterize the properties of these WT1-negative cells, we conducted additional transcriptomic analysis. The results revealed that cells in this region highly express the mesenchymal markers *NR2F2* and *ARX*, while lacking *WT1* signal. The gonad-mesonephric interface serves as the site of extensive tissue remodeling during early gonadal development, regulating cell migration and vascularization. Based on the morphological and molecular evidence described above, we have defined this WT1-negative mesenchymal region as the "gonadal-mesonephric interface," to distinguish it from the conventional mesonephric tissue, and this definition will be used in subsequent analyses (**see details in lines 290-312 of the revised manuscript**). We believe these supplementary experiments have significantly enhanced the rigor and scientific value of the paper. Once again, we sincerely thank you for guiding us through this important work.

Response Figure 3: Immunofluorescence staining identification of gonadal-mesonephric tissues. **a**, Bright-field microscopy of E24 gonadal-mesonephric tissue. The black dashed line denotes the gonads. Scale bars, 1 mm. **b**, Immunofluorescence images of WT1 (red) in porcine female E24 gonadal sections. Nuclei were counterstained with DAPI (blue). Scale bar, 100 µm. The dotted line indicates the demarcation between the gonad and the mesonephros; the left side shows the mesonephros, and the right side shows the gonad. The left arrow indicates WT-positive cells, and the right arrow indicates WT-negative cells.

References:

1. Liu, C., Rodriguez, K. & Yao, H.H. Mapping lineage progression of somatic progenitor cells in the mouse fetal testis. *Development* **143**, 3700-3710 (2016).
2. Sasaki, K. et al. The embryonic ontogeny of the gonadal somatic cells in mice and monkeys. *Cell Rep* **35**, 109075 (2021).
3. Mayère, C. et al. Origin, specification and differentiation of a rare supporting-like lineage in the developing mouse gonad. *Sci Adv* **8**, eabm0972 (2022).

Lines 208-210

The authors report co-expression of KRT19 and GATA4 localized to the cortical region of fetal gonads, with RNA-FISH confirming GATA4 expression in WNT6⁺ pre-granulosa cells at E50 (Fig. 3c, e). **However, the images do not clearly delineate the cortical region, making it difficult to assess the spatial context of the staining.**

Response: We sincerely thank you for these valuable comments. We have incorporated low-magnification panoramic images to provide a more comprehensive background of the gonadal tissue structure. Additionally, dashed lines have been added to the images to clearly delineate the cortical and medullary regions (**Response Figure 4**). We hope these modifications will help readers better visualize the expression patterns of these markers in the cortical region of the gonad.

Response Figure 4: Immunofluorescence images of KRT19 (purple) and GATA4 (green) in porcine female gonadal sections from different timepoints. Nuclei were counterstained with DAPI (blue). The dotted line demarcates the boundary between the gonad cortex and

medulla; the white arrow indicates the ovarian surface epithelium (OSE). Scale bars, 100 μm (as shown in Fig. 3f).

Additionally, the biological significance of the double staining—KRT19 and WNT6—should be explained to clarify the developmental relationship and functional implications of these markers during gonadal differentiation.

Response: Thank you for your suggestion. We would like to clarify that the dual fluorescence staining performed in our study actually involved GATA4 and KRT19, rather than KRT19 and WNT6. In accordance with your advice, we further elaborated on the biological rationale for selecting GATA4 and KRT19 for the co-localization experiments. To further investigate the developmental relationship among OSEC, Pre-GC I, and Pre-GC II, we performed pseudotime analysis. The results indicated that OSEC and Pre-GC I are positioned at relatively early developmental stages (**Response Figure 5**). Given that both cell types in mice and humans originate from the coelomic epithelium, we speculate that pigs may share a similar mechanism of origin¹⁻⁴. On the other hand, Pre-GC II is positioned closer to OSEC along the pseudotime trajectory and resides at a later developmental stage (**Response Figure 5**). Its transcriptional and spatial distribution features are highly conserved with those of OSEC, suggesting that Pre-GC II may primarily originate from OSEC^{2, 4}. To validate the spatial transcriptional features, we selected *GATA4* as a marker for coelomic epithelial cells and *KRT19* as a marker for both OSEC and Pre-GC II, and performed conventional immunofluorescence staining for verification. We have revised the relevant text to clearly state that the co-localization results of GATA4 with KRT19 serve as key experimental validation for the spatial transcriptomic analysis in this study (**Response Figure 4**). These independent morphological findings corroborate our spatial transcriptomic data, further ensuring the reliability and rigor of our analytical approach.

The modification is as follows (see details in lines 270-273 of the revised manuscript):

To validate the spatial transcriptomic data, we performed immunofluorescence staining. The results showed that GATA4 (a coelomic epithelial marker) and KRT19 (a marker for OSECs and Pre-GCs II cells) colocalized in the inner cortex of the fetal gonad, consistent with the spatial transcriptomics visualization.

Response Figure 5: Monocle3 was applied to perform UMAP dimensionality reduction on spatial transcriptome data from three types of supporting cell lineages, and pseudotemporal trajectory analysis was used to reconstruct the developmental pathway (as shown in Fig. 3b).

References:

1. Niu, W. & Spradling, A.C. Two distinct pathways of pregranulosa cell differentiation support follicle formation in the mouse ovary. *Proc Natl Acad Sci U S A* **117**, 20015-20026 (2020).
2. Mayère, C. et al. Origin, specification and differentiation of a rare supporting-like lineage in the developing mouse gonad. *Sci Adv* **8**, eabm0972 (2022).
3. Sasaki, K. et al. The embryonic ontogeny of the gonadal somatic cells in mice and monkeys. *Cell Rep* **35**, 109075 (2021).
4. Saitou, M., Nagano, M. & Mizuta, K. Mechanisms of human germ cell development. *Nat Rev Mol Cell Biol* (2025).

Lines 214-220

“To investigate the heterogeneity among OSECs, Pre-GCs I, and Pre-GCs II, we performed comparative analysis of their transcriptional profiles” **Where are shown the results of such analyses (Supp Table ??)**

Response: Thank you for your comment. In accordance with your recommendation, we have supplemented the corresponding **Supplementary Table 4 (see details in lines 241-245 and 252-256 of the revised manuscript)**.

Lines 214-218

“This analysis indicated dynamic gene expression gradients across these cell types: Pre-

GCs II exhibited downregulation of canonical OSECs markers (LHX9, KRT19) while upregulating TOX3, whereas OSECs maintained high LHX9 and KRT19 expression with minimal TOX3 expression¹⁵. Pre-GCs I showed a progressive decline in LHX9 and KRT19 while acquiring granulosa lineage markers such as WNT6.”

The paragraph should be revised to clearly explain the rationale behind selecting these markers and the biological implications of their expression patterns. Specifically, the authors should clarify why LHX9 and KRT19 are considered canonical OSEC markers and how their downregulation in Pre-GCs II reflects a shift away from the OSEC identity. Similarly, the upregulation of TOX3 and acquisition of granulosa lineage markers like WNT6 in Pre-GCs I should be linked to the progression toward granulosa cell differentiation.

Response: We sincerely appreciate your valuable suggestions. Regarding the selection of markers, we referred to published studies on gonadal somatic cell development in mice, humans, and pigs, and ultimately chose *LHX9* and *KRT19* as marker molecules for ovarian surface epithelial cells (OSECs)¹⁻⁴. We further validated their spatial localization using spatial transcriptomic data, confirming that their distribution aligns with the known anatomical features of OSECs.

We agree with your point that gene expression changes alone are insufficient to directly infer cellular relationships. To address this, we have supplemented our analysis with temporal data, which indicates that both OSECs and Pre-GC I belong to earlier developmental stages, while Pre-GC II cells are more likely derived from OSECs (**Response Figure 5**). This developmental timeline is highly conserved in both mice and humans⁶⁻⁸. Based on these analyses, we have carefully revised the relevant sections to ensure clearer expression and more rigorous logic. Once again, we thank you for your valuable guidance.

The modification is as follows (see details in lines 238-252 of the revised manuscript):

We used Monocle3 to analyze the developmental trajectories and fate specification of three putative supporting cell populations in the developing female gonad, which include OSECs, Pre-GCs I, and Pre-GCs II. Pseudotemporal analysis revealed that OSECs and Pre-GCs I reside at earlier developmental stages. This finding may be consistent with observations

in mice and humans, where OSECs and Pre-GCs I have been reported to share a common coelomic epithelial origin. Nevertheless, we observed distinct gene expression profiles between Pre-GC I and both OSECs and Pre-GCs II. Pre-GCs I were characterized by high *WNT6* expression and absence of *LHX9* and *KRT19*, whereas OSECs and Pre-GCs II shared a similar transcriptional profile, marked by expression of *LHX9*, *KRT19*, and *TOX3*. This progressive similarity, coupled with their pseudotemporal ordering, suggests that Pre-GC II likely differentiates directly from OSECs, while Pre-GCs I may represent a divergent, parallel branch of development. Together, these results indicate that the three cell types collectively form the supporting cell lineage during early gonadal development in pigs.

References:

1. Wamaitha, S.E. et al. Single-cell analysis of the developing human ovary defines distinct insights into ovarian somatic and germline progenitors. *Dev Cell* **58**, 2097-2111.e2093 (2023).
2. Garcia-Alonso, L. et al. Single-cell roadmap of human gonadal development. *Nature* **607**, 540-547 (2022).
3. Ge, W. et al. Spatiotemporal dynamics of early oogenesis in pigs. *Genome Biol* **26**, 2 (2025).
4. Chen, M. et al. Integration of single-cell transcriptome and chromatin accessibility of early gonads development among goats, pigs, macaques, and humans. *Cell Rep* **41**, 111587 (2022).
6. Niu, W. & Spradling, A.C. Two distinct pathways of pregranulosa cell differentiation support follicle formation in the mouse ovary. *Proc Natl Acad Sci U S A* **117**, 20015-20026 (2020).
7. Mayère, C. et al. Origin, specification and differentiation of a rare supporting-like lineage in the developing mouse gonad. *Sci Adv* **8**, eabm0972 (2022).
8. Saitou, M., Nagano, M. & Mizuta, K. Mechanisms of human germ cell development. *Nat Rev Mol Cell Biol* (2025).

Figure 1d is mentioned after figure 1f.

Response: We have checked and reorganized the order in which the panels of Fig. 1 are

cited in the main text. All panels are now referenced in accordance with their logical sequence of appearance in the Results section, ensuring a coherent narrative flow. We have also verified that the panel order within the supplementary figure itself aligns with the citations in the text.

Lines 231-233

The statement that “WNT signaling is a pivotal regulator of supporting cell lineage specification during early differentiation” is well established in other species. The authors should explicitly acknowledge this prior knowledge and clarify that their results demonstrate that the same regulatory role of WNT signaling is conserved in the pig model.

Response: Thank you for your valuable feedback. We have carefully revised the manuscript to further highlight the evolutionary conservation of the WNT signaling pathway across species.

The modification is as follows (see details in lines 260-263 of the revised manuscript):

These results suggested that the well-established role of WNT signaling as a pivotal regulator of supporting cell lineage specification during early differentiation, previously shown in mice and humans, is conserved in the pig model.

Supplementary figure 4 is not mentioned in the text

Supplementary Figure 4a is referenced after the entire figure description

Response: We thank the reviewer for this insightful comment. We have thoroughly addressed the issues regarding the citation of Supplementary Figure 4.

Line 250

“To validate these spatial relationships, we conducted RNA-FISH targeting the mesonephric marker *RARB* and interstitial marker *MAN1A1*.”

The authors should clarify the biological roles of *RARB* and *MAN1A1* and provide relevant references to support their selection as markers for the mesonephric and interstitial cell populations, respectively.

Response: We are grateful to the reviewer for their thoughtful question, which has prompted a deeper examination of our data.

Initially, we considered *RARB* primarily as a mesonephros region-specific marker. However,

observations of tissue sections from the E24 porcine gonad-mesonephros complex revealed that the mesonephros is, in fact, a remarkably complex organ in terms of cellular composition. Notably, we found that both *RARB*- and *MAN1A1*-expressing cell populations co-express *ARX* and *NR2F2*, which are key markers of mesenchymal cells. Based on these findings, we have sought to reclassify the two cell populations into two functionally distinct mesenchymal subtypes: one defined by high expression of *MAN1A1*, designated as “Mesenchymal Cell 1”, and the other by high expression of *RARB*, designated as “Mesenchymal Cell 2” (**Response Figure 6a-c**). Notably, both subtypes are spatially situated at the gonadal-mesonephric interface. Considering the reported roles of *RARB* and *MAN1A1* in cellular migration processes, we preliminarily suggest that these subtypes may be involved in guiding or otherwise influencing migratory processes at this interface^{1,2}. Accordingly, we have thoroughly revised the manuscript to no longer describe *RARB* and *MAN1A1* simply as markers of the mesonephros and interstitial cell, but rather to underscore their roles as two distinct mesenchymal cell subtypes and their potential functions in cell migration. This refinement has significantly improved the logical flow of the narrative and clarified its biological implications. We would like to once again express our sincere gratitude to the reviewer. Your insightful comments directly prompted this important clarification and deepened our understanding of the story.

The modification is as follows (see details in lines 290-312 of the revised manuscript):

In early gonad sections (E24 and E27), mesenchymal cells constitute a substantial fraction of the cell population; yet, they contribute minimally to gonadogenesis in later developmental stages. To explore their potential functions, we performed an in-depth analysis of mesenchymal cells and uncovered extensive heterogeneity. Through iterative clustering, we identified two distinct subpopulations with divergent distribution patterns: Mesenchymal cells 1 (MC-1) exhibit high expression of *ARX* and *MAN1A1*, whereas Mesenchymal cells 2 (MC-2) express *NR2F1* and *RARB*. Spatial visualization revealed a stratified distribution pattern, with MC-1 localizing to the gonadal tissue boundary region and MC-2 positioned directly beneath them. Given that the gonad-mesonephros interface is a site of active tissue remodeling and cell migration and considering prior evidence that *MAN1A1* silencing can promote cell migration and *RARB* knockout enhances EMT, we

hypothesized that these subpopulations might possess differential migratory capacities. Supporting this notion, Gene Ontology (GO) enrichment analysis showed that MC-2 was enriched in terms like "mesenchyme development," while MC-1 was enriched in both "mesenchyme development" and the "negative regulation of cell migration". Taken together, our findings identify two previously uncharacterized mesenchymal subpopulations located at the periphery of the developing gonad, each exhibiting distinct transcriptional profiles and spatial distributions. Their positioning along the tissue boundary, together with their divergent gene expression signatures, provides a plausible framework for further studies into their potential functions at the gonadal–mesonephric interface, which may involve roles in regulating cell migration.

Response Figure 6: Spatial transcriptional characteristics of mesenchymal cells. **a**, UMAP plot shows that the SPCs are divided into two subpopulations. **b**, Dot plot displays the characteristic gene expression in two mesenchymal cell subgroups. The diameter of each dot representing a gene indicates the proportion of cells expressing the gene, and the color gradient corresponds to the expression intensity. **c**, Spatial visualization illustrates the distribution of two distinct subgroups of mesenchymal cell at different developmental stages, each color-coded for identification. The dashed line indicates the gonad-mesonephros interface. Scale bars: 500 µm (as shown in Fig. 4 a-c).

References:

1. Alonso-Garcia, V. et al. High Mannose N-Glycans Promote Migration of Bone-Marrow-Derived Mesenchymal Stromal Cells. *Int J Mol Sci* **21** (2020).
2. Cai, X. et al. RARB associated with MSI, affects progression and prognosis of gastric cancer. *BMC Gastroenterol* **24**, 285 (2024).

Line 252-255

“RNA-FISH revealed differential localization patterns, with *RARB* predominantly localized to the mesonephros region, while *MAN1A1* expression was restricted to the adjacent interstitial region. These results indicated stratified distribution patterns of multiple cell types at the early gonad-mesonephros interface (Fig. 4b, c)”

The figure would benefit from clearer annotation: please label the gonad and mesonephros regions to help interpret the spatial expression patterns. The purpose of the dashed square is unclear—if it indicates a zoomed area, this should be clearly shown; otherwise, consider removing it. Finally, please elaborate briefly on the biological implications of these differential expression patterns beyond their distinct localization.

Response: Thank you for your valuable insights and suggestions. In the spatial visualization of *RARB* and *MAN1A1* cell types, we have added dashed lines and defined this region as the gonadal-mesonephric interface (**Response Figure 6c**). To provide a clearer view of the RNA-FISH staining images of *RARB* and *MAN1A1*, we have included enlarged areas highlighted with dashed boxes (**Response Figure 7a,b**). We believe these revisions have enhanced the clarity of our findings and strengthened their biological context. Thank you once again for your insightful feedback.

Regarding the biological functions of *RARB* and *MAN1A1*, we have further specified that *RARB* and *MAN1A1* are not merely regional or mesenchymal markers, but are co-expressed in *ARX/NR2F2*-positive mesenchymal cells. Based on this, we have reclassified them into two functionally distinct mesenchymal subsets—“Mesenchymal Subset 1” (high in *MAN1A1* expression) and “Mesenchymal Subset 2” (high in *RARB* expression). Both subsets are localized at the gonad-mesonephros interface, suggesting their potential roles in interface-related physiological processes such as cell migration. The manuscript has been comprehensively revised accordingly to ensure accurate interpretation of the associated biological significance (**see details in lines 290-312 of the revised manuscript**).

Response Figure 7: Spatial characterization of mesenchymal cells. **a**, RNA in situ hybridization for *RARB* (green, probe) and *MAN1A1* (purple, probe) in porcine female gonadal sections from E24. The cell nuclei were counterstained with DAPI (blue). The dashed box indicates the enlarged view of the corresponding area. Scale bar: 250 μm (zoomed-out), 50 μm (zoomed-in). **b**, RNA in situ hybridization for *RARB* (green, probe) and *MAN1A1* (purple, probe) in porcine female gonadal sections from E27. The dashed box indicates the enlarged view of the corresponding area. The cell nuclei were counterstained with DAPI (blue). Scale bar: 250 μm (zoomed-out), 50 μm (zoomed-in) (**as shown in Fig. 4d, e**).

Lines 255_258

“transcriptomic analysis identified several genes (*EBF1*, *ITGA9*, *HBE1*, and *NAV3*) commonly expressed across endothelial cells, interstitial cells, and mesonephric, indicating shared transcriptional features (Fig. 4f)”

Please clarify the biological relevance of the shared gene expression—do these findings suggest common function, origin, or interaction between the cell types?

Response: We sincerely thank you for your valuable comments. Regarding the biological relevance of the shared expression patterns of the *EBF1*, *ITGA9*, *HBE1*, and *NAV3* genes that you pointed out, we have conducted in-depth discussions and re-analysis. We acknowledge the reviewer point that relying exclusively on transcriptomic co-expression data—without additional functional validation such as knockout studies or co-localization assays—limits our ability to conclusively determine whether the observed expression patterns reflect shared functions, common regulatory origins, or potential interactions. Within the current framework of our study, we are unable to draw a definitive conclusion on this matter. Accordingly, we have removed these specific results and carefully revised

the entire manuscript to enhance the logical flow and consistency. We believe these refinements have helped to improve the overall quality and clarity of the paper.

Supp 5 b mentioned after c.

fig 4e mentioned after 4g.

Response: We have checked and reorganized the order in which the panels of Fig. 4 and Supplementary 5 are cited in the main text. All panels are now referenced in accordance with their logical sequence of appearance in the Results section, ensuring a coherent narrative flow. We have also verified that the panel order within the supplementary figure itself aligns with the citations in the text.

Lines 309-314

“Between E24 and E35, Mitotic PGCs are predominantly enveloped by supporting cells, while interstitial cells, endothelial cells, and stromal progenitor cells show minimal involvement (Fig. 5a, b; Supplementary Fig. 6a). In E30–E50 gonads, the proportion of supporting cells surrounding RA Oogonia decreases, potentially due to the substantial expansion of FGCs during this phase. By E50, Meiotic Oogonia are once again primarily surrounded by supporting cells (Fig. 5a, b; Supplementary Fig. 6a, c).”

The current images make it difficult to distinguish cell types, as the colored dots lack clear contrast. Please provide a simplified image with two clearly distinguishable colors highlighting only FGCs and supporting cells, excluding other germ cells. A separate image showing the remaining somatic cell types would also improve clarity. In this context, Supplementary Fig. 6a appears redundant and could be removed.

Response: Thank you very much for your valuable suggestions regarding image clarity optimization. While following your request to highlight the spatial relationship between supporting cells and germ cells, we re-analyzed the data based on the optimized results and found that, in addition to supporting cells, interstitial cells also represent a major component of the germ cell microenvironment, with a proportion comparable to that of the supporting cell lineage. Therefore, focusing solely on the relationship between supporting cells and germ cells would not accurately reflect the true cellular composition and spatial architecture in vivo. To address both the core requirement of "improving image clarity" and

to present these findings completely and accurately, we have adopted a novel presentation method (**Response Figure 8**). We hope this approach provides a more comprehensive and precise representation of the actual germ cell microenvironment.

Response Figure 8: Spatial visualization showing niches surrounding three female fetal germ cell subsets from E24 to E50. Mitotic PGCs, RA Oogonia, Meiotic Oogonia. Different cell types are color-coded. Panels (a) and (b) show magnified views of different regions in the figure. The outer cortical, inner cortical, and medullary regions of the gonads are delineated by dashed lines. Scale bar: 500 μm (zoomed-out), 100 μm (zoomed-in) (as shown in Fig. 5a).

Fig 6 e

A higher-magnification image is needed to clearly show protein co-expression, displaying only the relevant channels. Additionally, please clarify why the LAMA1 expression pattern appears markedly different between Figs. 6d and 6e, despite both corresponding to E50 samples.

Response: Thank you very much for your insightful comments and valuable suggestion.

Your question regarding the consistency of LAMA1 expression in Figures 6d and 6e is

crucial and has prompted us to conduct a thorough re-evaluation of our experimental methods.

Regarding the clarity of Figure 6e and the differences in LAMA1 expression, we fully concur with your observation that there is a notable discrepancy in the LAMA1 signal patterns between Figure 6d (immunofluorescence staining) and Figure 6e (combined immunofluorescence staining and RNA-FISH). To investigate the underlying causes, we have carefully revisited the experimental procedures. We have observed that the RNA-FISH experimental procedure significantly interferes with concurrently performed antibody staining, particularly for certain extracellular matrix components such as LAMA1, leading to compromised antigenicity or loss of signal. As a result, the LAMA1 immunofluorescence results presented in Figure 6e were incomplete and could not be directly compared with the standard staining results in Figure 6d.

Action taken: In light of these methodological limitations, and adhering to the highest standards of scientific rigor, we have decided to remove the data from Figure 6e. We apologize for this adjustment and sincerely appreciate your insightful observations, which have helped us identify and address the limitations of this part of the data. Once again, thank you for contributing to the improvement of our work.

Line 334-335

“Meanwhile, we also observed that LAMA1 was expressed in various cell types (Supplementary Fig. 6d). “ **Figure 6e does not allow the reader to evaluate expression intensity and cell-type identity simultaneously. A higher-magnification inset of the dashed-square region for expression levels may help.**

Response: Thank you for your valuable review comments. As explained in our previous response, due to current limitations of the LAMA1 antibody co-staining approach in adequately visualizing the morphological structure of LAMA1, we have decided to omit these staining results from the final analysis to ensure scientific rigor. We are currently optimizing the staining protocol to address this issue in future experiments. In accordance with your suggestion, we have adopted an alternative approach to more accurately illustrate the distribution of LAMA1 and germ cells within the gonad. We have added dashed boxes to the low-magnification panoramic images and provided corresponding

high-magnification insets of these areas. In the new enlarged insets, we have also included separate single-channel images for DAPI (**Response Figure 9a**). Additionally, to clearly illustrate the spatial distribution of different cell types in the female gonad of pig E50, corresponding schematic diagrams were generated (**Response Figure 9b**). Once again, we appreciate your help in improving the quality of our manuscript.

Response Figure 9: Spatial localization of porcine fetal germ cells. **a**, Immunofluorescence images of DDX4 (green) and LAMA1 (purple) in porcine female gonadal sections in E50 gonad. The dashed box indicates the enlarged view of the corresponding area. Nuclei were counterstained with DAPI (blue). Scale bar: 100 μm (zoomed-out), 50 μm (zoomed-in). **b**, Schematic illustration of a porcine fetal ovary at E50. The dashed box indicates the enlarged view of the corresponding area. (as shown in Fig. 5f, g).

Line 338-340

“Overall, these results collectively underscore the central role of supporting cells in oogonia specification, with hormonal signaling and extracellular matrix remodeling likely contributing to this process.”

This conclusion is overly generic and echoes long-standing literature rather than the author’s data. Please anchor the statement to the specific markers and experimental findings presented here—for example, detail how your supporting-cell signatures (e.g., X, Y) and ECM/hormonal readouts (e.g., Z) directly link to oogonia specification.

Response: We sincerely thank the reviewer for this insightful comment. Upon careful re-evaluation of our manuscript, we recognize that our initial conclusion was indeed overly broad and did not adequately anchor to our specific findings. Based on the reviewers’ comments, we have reorganized and integrated the manuscript content, further clarifying

that the somatic cells primarily surrounding the germ cells belong to the interstitial and supporting cell lineages. In addition, LAMA1, as a key component of the extracellular matrix, shows immunofluorescence staining signals predominantly localized around germ cell clusters, forming an encapsulating structure. Accordingly, we have adjusted the conclusion section based on these findings.

The modification is as follows (see details in lines 349-352 of the revised manuscript):

In summary, these results suggested that the distribution of interstitial cells in the medulla and endothelial cells near the inner cortex supports their potential functional collaboration, which may be critical for supporting female germ cell development.

Line 347

(Fig. 6a; Supplementary Fig. 7d).

If panel 6d is the first cited, it should also be presented first in the figure to maintain a logical and clear order. Please reorder the panels accordingly.

Response: We agree that aligning the panel order with the citation sequence improves the logical flow. Accordingly, we have reordered the panels in Figure 6. The other panels have been rearranged and relabeled accordingly to follow the sequence of their citation in the text. The figure legend and all in-text citations have been updated to reflect these changes.

Line 349-351

“We identified a significant interaction between WNT-ligand WNT3A, expressed by Pre-GCs II, and WNT receptors FZD3, expressed in Mitotic PGCs and RA Oogonia (Fig. 6a, b)” **WNT3A is not shown Fig 6b.**

Response: According to your suggestion, we have now included the expression data of WNT3A in both supporting cell lineage and germ cells **(as shown in Fig. 6b)**.

Lines 354-358

“High levels of membrane-associated CTNNB1 were detected in somatic cells and FGCs of gonads at various developmental stages. However, weak nuclear CTNNB1 was detected exclusively in RA oogonia, suggesting that the canonical WNT signaling mechanism may be transiently active only in RA Oogonia (Fig. 6d).”

Figure 6d does not allow evaluation of CTNNB1 localization; neither the cytoplasmic nor the nuclear signal is discernible. Please include higher-magnification images of

the same region, add a DAPI channel, and provide clear overlays so that nuclear versus cytoplasmic CTNNB1 can be unambiguously assessed.

Response: We thank you for this suggestion, and we have replaced the original Figure 6d with higher-magnification images that include separate DAPI channels and clear overlays. This revised panel is now designated as **Figure 6c** in the updated manuscript, where it more clearly illustrates the subcellular localization of CTNNB1 in RA oogonia (**as shown in Fig. 6c**).

Line 365-368

“Meiotic oogonia exhibited significantly upregulated ID gene expression relative to RA oogonia and Mitotic PGCs, suggesting an important role for BMP signaling in the PGC-to-oogonia transition”

Please, explain the role of ID genes (with references) in this process.

Response: Thank you for the suggestions. We have refined the manuscript accordingly, with particular emphasis on elucidating the relationship between the ID gene family and the BMP signaling pathway. BMPs bind to type I receptors ALK-2, ALK-3, and ALK-6, leading to the selective activation of Smad1/5 rather than Smad2/3 and the subsequent upregulation of *ID* family genes (*ID1*, *ID2*, *ID3*)¹. This represents one of the canonical mechanisms of the BMP signaling pathway. Previous studies have confirmed that during the specification of primordial germ cells (PGCs) into oogonia in both humans^{2,3} and mice^{4,5}, the expression levels of the ID family gradually increase. This study further validates that the ID family, as downstream target genes of BMP signaling, are upregulated during the differentiation of PGCs into oogonia, suggesting that the BMP signaling pathway may play a conserved regulatory role in this process across different species.

The modification is as follows (see details in lines 434-436 of the revised manuscript):

Meiotic oogonia exhibited a significant upregulation in the expression of *ID* genes, which are established downstream targets of the BMP signaling pathway, compared to RA oogonia and Mitotic PGCs.

References:

1. Miyazono, K. & Miyazawa, K. Id: a target of BMP signaling. *Sci STKE* **2002**, pe40

(2002).

2. Li, L. et al. Single-Cell RNA-Seq Analysis Maps Development of Human Germline Cells and Gonadal Niche Interactions. *Cell Stem Cell* **20**, 858-873.e854 (2017).
3. Murase, Y. et al. In vitro reconstitution of epigenetic reprogramming in the human germ line. *Nature* **631**, 170-178 (2024).
4. Miyauchi, H. et al. Bone morphogenetic protein and retinoic acid synergistically specify female germ-cell fate in mice. *Embo Journal* **36**, 3100-3119 (2017).
5. Nagaoka, S.I. et al. ZGLP1 is a determinant for the oogenic fate in mice. *Science* **367** (2020).

Line 359-365

The BMP signaling pathway not only regulates PGCs specification but has also been shown in recent studies to play a pivotal role in oogenesis⁵⁸. We noticed that BMP ligand-receptor interactions between FGC subtypes and surrounding Pre-GCs II revealed progressively increasing BMP ligand (BMP2, GDF5) activity from E24 to E50. At E50, the ligand BMP2 from Pre-GCs II exhibited significant interactions with RA Oogonia and Meiotic Oogonia, whereas GDF5 functioned exclusively in Meiotic Oogonia (Fig. 6a). **Figure 6a is too small to resolve or compare the reported BMP ligand-receptor interactions. Please supply (i) a higher-magnification supplementary panel that clearly shows these contacts and (ii) a separate figure that tracks the interaction within a single cell type across the developmental time points.**

Response: Thank you for your valuable feedback and interest in our analysis of ligand–receptor interactions in this study. Your observation regarding the small size of Fig. 6a is very valid. We fully agree and have rearranged the figure according to different signaling pathways. Through these improvements, we aim to present the ligands and receptors across various stages and types more clearly and intuitively. Once again, we sincerely appreciate your constructive comments, which have helped us enhance the quality of our manuscript **(as shown in Fig. 6 and Supplementary 7 and 8)**.

Given the importance of the BMP signaling pathway, we have supplemented the Results section with interactions between different cell types based on this pathway **(as shown in Fig. 6g)**.

Line 368-372

Since a hallmark of BMP pathway activation is the nuclear translocation of phosphorylated (p) SMAD proteins, we assessed pSMAD1/5 localization in fetal gonads. At E24, no pSMAD1/5 expression was detected in BLIMP1+ cells; at E35, pSMAD1/5 was exclusively localized to the cytoplasm of DDX4+ cells; whereas by E50, distinct nuclear expression was observed in DDX4+ cells⁶⁰ (Fig. 6e; Supplementary Fig. 7e)

The present images do not permit reliable assessment of pSMAD1/5 subcellular localization—E35 and E50 panels appear essentially identical. Please supply higher-magnification images with a DAPI counterstain so that nuclear versus cytoplasmic pSMAD1/5 can be clearly distinguished.

Response: We thank you for this helpful suggestion. To address the concern regarding pSMAD1/5 subcellular localization, we have now included higher-magnification images as well as additional single-channel images for the DAPI stain to enhance readability. The corresponding modifications are outlined in the points below. We provide DAPI single-channel images under high background conditions to enhance credibility. At E35, the results clearly demonstrate that pSMAD1/5 signals are predominantly localized in the cytoplasm of DDX4-positive germ cells. By E50, a marked nuclear enrichment of pSMAD1/5 signals within DDX4-positive germ cells is distinctly observed (**Response Figure 10**).

Response Figure 10: Immunofluorescence images of DDX4 (purple) and p-SMAD1/5 (green) in porcine female gonadal sections from different timepoints. Nuclei were counterstained with DAPI (blue), Scale bar: 20 μm (as shown in Fig. 6 h).

Line 414 and FIGLA-positive oocytes are positioned within

414the medulla³, 21, 38 NO encuentro donde fue estudiado figla

Please explain what are FIGLA-positive oocytes. Please provide reference.

Response: Thank you for your question. As outlined in our study, the selected E24–E50 period covers three germ cell stages, Mitotic PGCs, RA-responsive oogonia, and Meiotic oogonia, but does not include the oocyte stage. Therefore, in the discussion, we integrated existing literature to explore the spatial distribution of oocytes. During human gonad development, immature oocytes are primarily located in the cortical region, whereas mature oocytes are mostly situated in the medullary region¹. *FIGLA*, a marker gene for the oocyte stage², was detected in both 19 PCW and 21 PCW samples, with positive cells predominantly distributed in areas adjacent to the medulla regions (**Response Figure 11a, b**). We hope this clarification addresses your question.

Notes: The figure was cited from references ¹.

Response Figure 11: Spatial localization of human fetal germ cells. **a**, High-resolution imaging of a representative transverse section of a human ovary at 21 post-conceptual weeks (PCW), with intensity proportional to smFISH signal for *POU5F1* (green, primordial germ cells), *DDX4* (red, fetal germ cells), *STRA8* (cyan, pre-meiotic germ cells) and ***FIGLA*** (**yellow, oocytes**); n = 4. The white dashed rectangle highlights the enlarged gonadal region. Scale bars = 100 μ m. **b**, *cell2location* estimated cell abundance (colour intensity) contributed by each germ cell to each Visium spot (colour) shown over the H&E image of a 19 PCW ovary; n = 2. Scale bars = 1 mm. E = embryonic day; Expr = expressed, FGC = fetal germ cells; P = postnatal day; PCW = post-conceptual weeks; PGC = primordial germ cells.

References:

1. Garcia-Alonso, L. et al. Single-cell roadmap of human gonadal development. *Nature* **607**, 540-547 (2022).
2. Li, L. et al. Single-Cell RNA-Seq Analysis Maps Development of Human Germline Cells and Gonadal Niche Interactions. *Cell Stem Cell* **20**, 858-873.e854 (2017).

Discussion

Lines 415-418

At E50, we observed female germline cyst formation, similar to mouse ovarian cysts containing ~6 oocytes supported by 24 nurse cells, suggesting that cortical expansion of porcine germ cells ensures a sufficient pool for subsequent oogenesis.

Where is this shown?

Response: Thanks for the comment. In our staining of LAMA1 and germ cells in E50, we observed that the germ cells were arranged in clustered aggregates, resembling germline cyst formation in mice (**Response Figure 9a, b**). To ensure more precise wording, we have further revised the manuscript accordingly.

The modification is as follows (see details in lines 504-508 of the revised manuscript):

At E50, porcine oogonia were organized into clusters within the cortical region. This spatial distribution is notably similar to that of the female germline cysts in mice, which comprise approximately 24 nurse cells and ultimately yield about 6 oocytes. This similarity suggests

that pigs may employ a comparable mechanism to ensure an ample cellular reserve for subsequent oogenesis.

Response Figure 9: Spatial localization of porcine fetal germ cells. **a**, Immunofluorescence images of DDX4 (green) and LAMA1 (purple) in porcine female gonadal sections in E50 gonad. The dashed box indicates the enlarged view of the corresponding area. Nuclei were counterstained with DAPI (blue). Scale bar: 100 μm (zoomed-out), 50 μm (zoomed-in). **b**, Schematic illustration of a porcine fetal ovary at E50. The dashed box indicates the enlarged view of the corresponding area. (as shown in Fig. 5f, g).

Lines 439-453

This paragraph should be grounded in the author's own results. Please revise it to explicitly reference data generated in your study that support these conclusions; if your experiments do not provide such evidence, the paragraph should be removed. This is extensive to other paragraphs of the discussion.

Response: We agree that the discussion must be firmly grounded in the data generated by the present study. Upon careful review, we recognized that the paragraph in question (previously in lines 439-453) was indeed speculative and pertained to future work, without direct support from our current experimental results. Therefore, in strict adherence to this comment, we have removed that entire paragraph from the discussion. We have also reviewed the entire Discussion section to ensure that all conclusions are supported by our own data.

Methods Line 46-52

PDMS microfluidic chip: Which manufacturer? Please describe in more detail. If no commercial brand is used, the origin should be specified.

well-specific barcode. ¿Which manufacturer? Please describe in more detail. If no

commercial brand is used, the origin should be specified.

Line 532-533 Please change “Statistically analyze” for “Statistical analysis”

Response: Thank you for your valuable comments.

The PDMS microfluidic chip was designed by our research team and manufactured by Suzhou Cchip Scientific Instrument Co., Ltd. The well-specific barcodes were synthesized by Sangon Biotech (Shanghai) Co., Ltd. The manufacturer information is included in *Methods* section **(see details in lines 586-601 of the revised manuscript)**.

The wording “Statistically analyze” has been revised to “Statistical analysis” as suggested **(as shown in Supplementary Fig. 3b legend)**.

Reviewer #2 (Remarks to the Author):

Dear authors,

This paper provided an interesting dataset of spatial transcriptomics with relatively low-resolution regarding bin size and molecular signature of pig fetal female gonads during a period of development (between E24-E50). The data has shortcomings, but it is important and unique to have a broader picture of female gonadogenesis in mammals.

Response: We thank you for the positive comments.

The clusters provided are general, the genes chosen rather standard (any new things compared to human or mouse?),

Response: In our study, initial cell clustering was performed based on conserved classical marker genes in mammals, which enabled the identification of major cell types. Furthermore, we identified and validated two novel mesenchymal cell types situated at the gonad-mesonephros interface, marked specifically by *MAN1A1* and *RARB*, respectively (see details in lines 290-312 of the revised manuscript). To the best of our knowledge, based on a comprehensive review of the literature, this particular phenotype–localization combination has not been explicitly reported in previous studies of mouse or human development.

the clusters are not well defined and restricted (because the resolution bins are relatively large and so each bin is a mixture of cells/signatures). For example, the OSE cluster is clearly broader than just OSE.

Response: Thank you for your constructive comments regarding the resolution, generality of our clusters, and the interpretation of the OSE cluster. We appreciate your recognition of the dataset's unique value to the field of mammalian gonadogenesis. We would like to clarify the point regarding resolution: Our spatial transcriptomics data are composed of 10µm bins. As the typical diameter of a eukaryotic cell is generally around 10µm, we believe that while some bins may contain mixed signatures, our resolution allows us to reasonably define and distinguish major cell populations both in the UMAP and within the preserved spatial context. Therefore, we initially did not rely on single-cell data for deconvolution and based our annotations on the clarity of the spatially-resolved clusters. Regarding your specific concern that the OSE cluster appears broader than expected, we

agree with your critical assessment. We have thoroughly re-examined our clustering approach and hypothesize that our initial use of Harmony for batch correction may have led to an over-correction phenomenon (overfitting), resulting in overly generalized clusters. To resolve this issue and obtain a more biologically faithful cellular map, we have taken the following steps: We have updated our batch correction methodology to use the Reciprocal PCA (RPCA) integration method. RPCA is a more conservative approach that matches cells in a low-dimensional space, proving to be more effective at preserving subtle biological differences while mitigating technical noise. The new clustering result, derived from the RPCA corrected data, now provides more distinct and biologically realistic cell definitions. We are currently refining our cluster annotations and data interpretations based on this improved map. We believe this addresses your concern about the generality of the clusters, particularly the OSE population.

It further provides standard ligand-receptor analysis and spatial neighbourhood analysis, which is adequate. The included validation using immunofluorescence (IF) is valuable, but the images are of low quality morphology (do you have access to paraffin sections?).

Response: Thank you very much for your suggestion. We would like to note, however, that the spatial transcriptomics method employed in this study requires the use of frozen sections for tissue embedding. For this reason, during sample collection, and to ensure we obtained sufficient female gonadal tissue, we only performed fresh-frozen tissue embedding and did not proceed with paraffin embedding. Due to recent impacts of classical swine fever and other factors, our pig farm is unable to provide additional porcine embryo samples for supplementary experiments. We sincerely apologize for this limitation. To enhance the quality of images related to ligand-receptor interactions, we have supplemented with higher-magnification immunofluorescence staining images and provided single-channel DAPI images, thus more clearly illustrating cellular localization features (**see details in lines Fig. 6c and 6h of the revised manuscript**).

The analysis is standard, but the dataset is worth providing as resource, whereas the interpretation of the data needs some improvement and claims need to be refined.

Response: Thank you for your valuable suggestions. We have incorporated the relevant explanations into the manuscript and revised the statements accordingly, following your guidance. We have also thoroughly and carefully revised the entire manuscript to temper the conclusiveness of our language. In addition, we have provided point-by-point responses to all the questions you raised. Here are several examples of statements from our revised manuscript.

Example 1 (see details in lines 349-352 of the revised manuscript): In summary, these results suggested that the distribution of interstitial cells in the medulla and endothelial cells near the inner cortex supports their potential functional collaboration, which may be critical for supporting female germ cell development.

Example 2 (see details in lines 392-396 of the revised manuscript): In summary, these results indicate that the supporting cell lineages and interstitial cells interact with germ cells in a spatial context, suggesting that this cell population may play a more critical role during the specification of PGCs into oogonia. Furthermore, LAMA1, as a key component of the ECM, provides structural support by enveloping the germ cells.

Example 3 (see details in lines 171-177 and 187-194 of the revised manuscript): We also recognize that relying solely on transcriptomic similarity data without additional functional validation, such as knockdown studies or co-localization analyses, limits our ability to determine whether the observed expression patterns reflect shared functions, common regulatory origins, or potential interactions. Within the current framework of our study, we are unable to draw definitive conclusions on this matter. Therefore, we have removed these specific results and have carefully revised the entire manuscript to streamline the logical flow and improve consistency. In addition, we have incorporated background information on germ cell epigenetics to enhance the readability of the text.

In addition to optimizing the content of the manuscript, this study also incorporated published spatial transcriptomic data from human female fetal gonads at 14 weeks and conducted a cross-species comparative analysis based on this data (**see details in lines 209-219 and 274-283 of the revised manuscript**). The results highlight the conservation and divergence in the spatial distribution patterns of germ cells and supporting cell lineages in the gonads between pigs and humans. We believe that these additions and analyses

contribute to enhancing the academic quality of the manuscript. Your comments have significantly enhanced the logic and rigor of our paper, for which we sincerely appreciate your contribution.

Point of improvement:

1.The evidence for preGCs I and II is not robust. Could the authors provide better validation and higher magnification/better quality IF to separate between the two populations?

Response: We sincerely thank you for your valuable suggestion. We fully recognize the importance of high-quality immunofluorescence (IF) data. To that end, we initially attempted IF experiments using TOX3, a marker commonly cited in the literature¹. However, after multiple attempts, we found that the commercially available TOX3 antibodies failed to produce specific signals in porcine fetal gonad tissues. As an alternative approach, we have screened for specific marker genes of Pre-GC I and Pre-GC II from our spatial transcriptomics data and initiated the customization of corresponding antibodies. However, due to the extended timeline required for antibody production and validation, we were unable to obtain reliable results within the current revision period. We have openly communicated this technical challenge and acknowledge it as a limitation of the present study, which we plan to prioritize in future work.

Although direct IF verification posed challenges, our in-depth analysis has provided bioinformatic evidence supporting the existence of both Pre-GC I and Pre-GC II, as well as their developmental relationship. We have supplemented the pseudotemporal analysis. The results suggested that Pre-GC I and OSEC are at an earlier developmental stage, while Pre-GC II is more likely to be derived from OSEC (**Response Figure 5**). This pattern is highly conserved in both mice²⁻⁴ and humans⁵, suggesting that pigs may share a similar developmental mechanism with mice and humans (**as shown in lines 238-243**).

Response Figure 5: Monocle3 was applied to perform UMAP dimensionality reduction on spatial transcriptome data from three types of supporting cell lineages, and pseudotemporal trajectory analysis was used to reconstruct the developmental pathway (as shown in Fig. 3 b).

In particular there are no germ cells in the spatial location of preGC I, these cells seem rather stromal to me. What would preGC I be in the middle of the gonad far away from germ cells? Could this cluster currently include several different cell types?

Response: To indirectly validate the possibility of such spatial distribution of cell types, we reanalyzed the published spatial transcriptomic data of human fetal gonads at 14 weeks (female)⁶ (**Response Figure 12a**). We successfully identified homologous cell populations in humans corresponding to porcine OSEC, Pre-GC I and Pre-GC II. Spatial localization analysis clearly demonstrated that human Pre-GC I homologous cells are predominantly enriched in the medulla, while Pre-GC II homologous cells are enriched in the cortex (**Response Figure 12b, c**). This finding is also highly consistent with reports in mice², suggesting that the "Pre-GC I localized in the medulla, Pre-GC II localized in the cortex" pattern may be a developmentally conserved mechanism across species (as shown in lines 276-283).

Regarding the key issue you raised about Pre-GC I being distant from germ cells, we have carefully reviewed the published literature on human fetal gonads⁶. The function of Pre-GC I is primarily reflected in its interaction with FIGLA-positive oocytes, whereas Pre-GC II mainly acts on RA Oogonia and Meiotic Oocytes^{2, 6}. In human fetal ovaries at 19 and 21 post-conception weeks (PCW), FIGLA-positive oocytes are predominantly localized in the medullary region⁶ (**Response Figure 11a-c**). As shown in this study, the selected E24–

E50 period covers three germ cell stages, Mitotic PGCs, RA Oogonia, and Meiotic Oogonia, but does not include the oocyte stage. Therefore, as the oocytes progressively migrate toward the medulla, Pre-GC I become localized around the germ cells. We hope this explanation and additional clarification adequately address your concerns.

Response Figure 12: Spatial transcriptomic comparison of supporting cell lineages Between Human and Pig. **a**, Timeline of oogenesis and sampling strategy for cross-species analysis in pigs and humans. **b**, Spatial visualization illustrates the distribution of three distinct subgroups of porcine supporting cell lineage at different developmental stages, each color-coded for identification. Panels (a) and (b) show magnified views of different regions in the figure. The cortical and medullary regions of the gonads are delineated by dashed lines. Scale bars: 500 μm (zoomed-out), 100 μm (zoomed-in). **c**, Spatial localization of three human supporting cell lineage subgroups at 14 gestational weeks. The dashed box indicates the enlarged view of the corresponding area. Scale bars: 500 μm (zoomed-out), 100 μm (zoomed-in) (as shown in Fig. 2 g and Fig. 3 f, g).

Notes: The figure was cited from references ⁶.

Response Figure 11: Spatial localization of human fetal germ cells. **a**, High-resolution imaging of a representative transverse section of a human ovary at 21 post-conceptual weeks (PCW), with intensity proportional to smFISH signal for *POU5F1* (green, primordial germ cells), *DDX4* (red, fetal germ cells), *STRA8* (cyan, pre-meiotic germ cells) and *FIGLA* (yellow, oocytes); $n = 4$. The white dashed rectangle highlights the enlarged gonadal region. Scale bars = 100 μm . **b**, *cell2location* estimated cell abundance (colour intensity) contributed by each germ cell to each Visium spot (colour) shown over the H&E image of a 19 PCW ovary; $n = 2$. Scale bars = 1 mm. E = embryonic day; Expr = expressed, FGC = fetal germ cells; P = postnatal day; PCW = post-conceptual weeks; PGC = primordial germ cells.

References:

1. Estermann, M.A. et al. DMRT1-mediated regulation of TOX3 modulates expansion of the gonadal steroidogenic cell lineage in the chicken embryo. *Development* **150** (2023).
2. Niu, W. & Spradling, A.C. Two distinct pathways of pregranulosa cell differentiation support follicle formation in the mouse ovary. *Proc Natl Acad Sci U S A* **117**, 20015-20026 (2020).
3. Mayère, C. et al. Origin, specification and differentiation of a rare supporting-like lineage in the developing mouse gonad. *Sci Adv* **8**, eabm0972 (2022).
4. Sasaki, K. et al. The embryonic ontogeny of the gonadal somatic cells in mice and monkeys. *Cell Rep* **35**, 109075 (2021).
5. Saitou, M., Nagano, M. & Mizuta, K. Mechanisms of human germ cell development. *Nat Rev Mol Cell Biol* (2025).
6. Garcia-Alonso, L. et al. Single-cell roadmap of human gonadal development. *Nature* **607**, 540-547 (2022).

2. Why are OSE cells inside the E50 gonad (Fig S4A)? I think the OSE clusters should be further subclustered to understand what those cells are.

Response: Thank you for your valuable feedback and insightful observation regarding the localization of Cluster 1 (OSECs) at E50. We agree that this point warrants careful consideration and verification.

We have diligently revisited our clustering methodology and critically evaluated our approach to batch correction. We found that the initial clustering using Harmony might have led to over-correction (overfitting), potentially obscuring genuine biological differences.

To address this, we performed a thorough comparison:

1. Clustering samples from different developmental stages independently.
2. Comparing various batch correction strategies.

We ultimately determined that using Reciprocal PCA (RPCA) for batch correction yielded the most robust and biologically faithful result. RPCA works by matching cells based on the principal components in a low-dimensional space, effectively mitigating the influence

of high-dimensional noise. Crucially, compared to Harmony, RPCA is a gentler approach that is better at preserving genuine biological variance during the batch integration process. The resulting map, based on RPCA integration, now provides a more accurate and reliable representation of the cellular architecture. We have updated our figures and analysis accordingly. We appreciate you raising this critical point, as it has allowed us to significantly improve the robustness and confidence in our final cellular map.

3. Having cells expressing similar genes does not mean they share a common origin. Hence the claim that interstitial cells are derived from the mesonephros is not well evidenced. Can the authors provide more convincing data for common origin or perhaps tone down that claim.

Response: We sincerely thank you for your valuable comments. Regarding the biological relevance of the shared expression patterns of the *EBF1*, *ITGA9*, *HBE1*, and *NAV3* genes that you pointed out, we have conducted in-depth discussions and re-analysis. We acknowledge the reviewer point that relying exclusively on transcriptomic co-expression data—without additional functional validation such as knockout studies or co-localization assays—limits our ability to conclusively determine whether the observed expression patterns reflect shared functions, common regulatory origins, or potential interactions. Within the current framework of our study, we are unable to draw a definitive conclusion on this matter. Accordingly, we have removed these specific results and carefully revised the entire manuscript to enhance the logical flow and consistency. We believe these refinements have helped to improve the overall quality and clarity of the paper.

4. Comparison with humans should be more detailed regarding similarities and specifically differences between the different compartments/cell types.

Response: We sincerely thank the reviewer for this insightful suggestion. To precisely compare the development of germ cells between humans and pigs, we selected highly comparable gonad samples, human 14-week¹ and porcine E50, based on key germ cell developmental events, using spatial transcriptomics for systematic analysis (**Response Figure 13a-c**). Our analysis revealed a high degree of similarity in the spatial localization of both germ cells and supporting cell lineages between the two species (**Response Figure 13d**). However, a key difference was identified: Mitotic PGCs are almost

undetectable in porcine E50 (**Response Figure 13a**), whereas in humans, their presence persists at 14 weeks and, according to literature, continues until 20 weeks¹ (**Response Figure 11a and Response Figure 13d**) (as shown in lines 209-219).

Response Figure 13: Comparative analysis of spatial transcriptomes of human and porcine fetal germ cells. a, Spatial visualization displays the spatial localization of three subgroups of porcine FGCs. The germ cells at the three different stages were represented by different colors. The dashed box indicates the enlarged view of the corresponding area. The outer cortical, inner cortical, and medullary regions of the gonads are delineated by dashed lines. The image shown is from a single representative sample (the replicate is presented in the **Supplementary Fig. 3c**). Scale bar: 500 μm (zoomed-out), 100 μm (zoomed-in). **b**, Kernel Density Plot shows the spatial distance of two fetal germ cell subpopulations (RA Oogonia and Meiotic Oogonia) from a defined tissue boundary in the E50 female gonad. The x-axis represents the Distance from Boundary (in μm), and the y-axis represents the Density (probability distribution). **c**, Timeline of oogenesis and

sampling strategy for cross-species analysis in pigs and humans. **d**, Spatial localization of three human supporting cell lineage subgroups at 14 gestational weeks. The dashed box indicates the enlarged view of the corresponding area. Scale bar: 500 μm (zoomed-out), 100 μm (zoomed-in) (as shown in Fig. 2 f, Supplementary Fig. 3f and Fig. 2 g,h).

References:

1. Garcia-Alonso, L. et al. Single-cell roadmap of human gonadal development. *Nature* **607**, 540-547 (2022).

5. Cluster 1 in the DEG table has a low quality set of genes (lots of RPL genes). Are you sure this cluster does not represent low quality cells? I am a bit concerned about the thresholds used for filtering the data. The DEGS from several clusters are unexpected and the usual suspects are absent. Can the authors mention this as limitation of the technology used?

Response: Thank you very much for pointing out the concern regarding the quality of Cluster 1 due to the high expression of ribosomal protein (RPL) genes and the unexpected DEGs. We fully recognize the importance of ruling out technical artifacts or low-quality cell populations.

Following your comment, we have carefully re-examined our data after the revised clustering analysis and found that Cluster 1 indeed continues to show elevated expression of certain RPL genes. However, we firmly believe this cluster does not represent a low-quality or dying cell population for the following compelling reasons:

High Data Quality Metrics: Cells in Cluster 1 exhibit significantly high UMI counts and feature counts (number of expressed genes) compared to the overall dataset median, directly contradicting the typical signatures of low-quality cells (which usually have low counts and few features).

Specific Biological Identity: This cluster expresses a highly specific and canonical set of Ovarian Surface Epithelium Cell (OSEC) marker genes (as previously discussed, e.g., *LHX9* and *KRT19*). Low-quality clusters, in contrast, typically lack such a clear, established biological identity.

Unique Spatial Localization: Cluster 1 demonstrates a highly specific spatial enrichment at

the outermost cortical region of the developing gonad, confirming its biological location and role, which would not be observed in randomly distributed technical noise or debris.

Biological Interpretation (RPL genes): We interpret the high RPL gene expression not as a stress signature, but as a potential hallmark of high translational or transcriptional activity. Furthermore, our subsequent pseudotime analysis consistently supported that these cells possess a high degree of plasticity and differentiation potential (stemness-like properties), which often correlates with high metabolic rates and, consequently, high ribosomal content. In conclusion, based on the superior quality metrics, definitive cell identity markers, specific spatial distribution, and supporting trajectory analysis, we are confident that Cluster 1 represents a genuine, biologically active OSEC population with high biosynthetic activity, rather than a cluster defined by technical noise. We appreciate this scrutiny, as it strengthens our confidence in the biological fidelity of our findings.

6. I think the resolution (bin-size) is a severe limitation and this should also be explicit in the text.

Response: We sincerely thank the reviewer for this insightful suggestion. In our approach, each spot represents a 10 μm bin. Given that the typical diameter of a eukaryotic cell is generally around 10 μm , our method achieves higher resolution and data quality compared to existing commercialized methods. For instance, while BGI's proprietary Stereo-seq technology can achieve a 220 nm spot diameter, subsequent analysis typically pools 50 bins to define a single cell, with median captured gene and UMI counts of 792 and 1910, respectively¹. Likewise, 10X Genomics' recently launched Visium HD technology offers a 2 μm spot; however, it is usually analyzed in 16-bin units and requires prohibitively high costs to achieve a resolution level comparable to ours. Critically, our technology platform is capable of yielding superior data quality at a lower cost, with a resolution that can closely approximate the single-cell level.

References:

1. Chen, A. et al. Spatiotemporal transcriptomic atlas of mouse organogenesis using DNA nanoball-patterned arrays. *Cell* **185**, 1777-1792.e1721 (2022).

7. Title: remove 'high definition'

Response: We sincerely appreciate your valuable focus on the precision of our terminology, especially concerning data resolution. We fully agree with your assessment. We have taken immediate action to revise the manuscript to align with the standards you suggested. This modification serves a dual purpose: it eliminates ambiguity regarding the terms used, and concurrently, we have highlighted **(as reflected in our preceding response 6)** the significant advantage of our 10 μm resolution over other platforms. Furthermore, we adjusted "single-cell resolution" to "near single-cell resolution" in the revised manuscript **(see details in lines 92-93 of the revised manuscript)**. We hope this comprehensive response successfully alleviates your concerns regarding the resolution aspect of our data.

Reviewer #3 (Remarks to the Author):

Oogenesis proceeds in the ovary through the interaction of various cell types. Cell-cell communication between adjacent cells is essential for cell differentiation. In this study, the authors present a spatial transcriptomic dataset that covers the early stage of ovarian development in a porcine fetus. The authors generated a dataset capturing the development of several ovarian cells, including oocytes, supporting cells, endothelial cells, and a part of mesonephric cells. They also demonstrate cell-to-cell communication between supporting cells and oocytes. This dataset is informative and useful resource for the field of reproductive biology. Therefore, it is important that the data are shared publicly. However, the current version of the manuscript needs to be restructured to make data easier to follow and interpret.

Response: Thank you very much for your valuable suggestions. We have revised the manuscript accordingly by improving the overall logical flow and including schematic diagrams to illustrate the spatial distribution of different cell types in the gonad. Corresponding annotations have also been added to the Methods section to clarify the procedures for data generation and analysis. Furthermore, we have made all relevant data and research code publicly available to enhance reproducibility and research transparency. We sincerely appreciate your constructive comments.

Data availability (see details in lines 870-875 of the revised manuscript)

All data from this study have been deposited to GSA (<https://ngdc.cncb.ac.cn/gsa>) with the accession number CRA023499. Raw image files used in the figures that support the findings of this study are available from the corresponding authors upon reasonable request. Pipeline code generated to perform the analysis in this study is available in Prince-Xiaaa/pig-fetal-ovary-st-code (github.com).

Major points to address

- 1. The authors demonstrated the differentiation and spatial distribution of ovarian cells. However, it is unclear whether this is completely novel or if related findings have been reported before. Please include the information about the knowledge shown in mouse and explain the similarities and differences. This will highlight the significance of this paper and relationship in relation to previous studies.**

Response: Thank you for your suggestion. The pig has recently been regarded as an ideal animal model due to its highly conserved embryonic morphology, organ size, and germ cell specification process with humans¹. However, owing to technical constraints, previous investigations into the development of germ cells and supporting cell lineages in pigs have primarily relied on single-cell transcriptomics^{2,3}, which does not allow for the resolution of their spatial characteristics. As a result, the spatial dynamics of germ cells and surrounding somatic cells during the critical transition from primordial germ cells (PGCs) to oogonia remain uncharacterized.

In this study, we revealed that the spatial localization of germ cell and supporting cell lineages is highly conserved in both mice and humans. Within the supporting cell lineage, the spatial positioning of the first two granulosa cell types and ovarian surface epithelial cells (OSECs) is particularly conserved. Furthermore, we supplemented the manuscript with pseudotime analysis, which demonstrated that OSECs and Pre-Granulosa Cells I (Pre-GC I) reside at earlier developmental stages, while Pre-Granulosa Cells II (Pre-GC II) are more likely derived from OSECs (**Response Figure 5**) (see details in lines 238-243 of the revised manuscript). This developmental progression is highly conserved across mice^{4,5} and humans^{6,7}.

Response Figure 5: Monocle3 was applied to perform UMAP dimensionality reduction on spatial transcriptome data from three types of supporting cell lineages, and pseudotemporal trajectory analysis was used to reconstruct the developmental pathway (as shown in Fig. 3 b).

To conduct cross-species comparative analysis, we systematically evaluated existing data resources. Given that gonadal development in mice has been relatively well-studied but lacks available spatial transcriptomic data⁸⁻¹⁰, we cited existing literature to highlight both

similarities and differences between species. Accordingly, this study selected publicly available spatial transcriptomic data from a 14-week human female fetus as the basis for cross-species comparison¹¹ (**Response Figure 14a**). The analysis clearly identified three corresponding cell types: OSEC, Pre-GC I, and Pre-GC II. Importantly, the spatial localization of these cells in the human fetal ovary is highly consistent with our primary findings, which strongly supports the cross-species conservation of this spatial organization between humans and pigs (**Response Figure 14b**). Notably, previous studies on mouse fetal ovaries have similarly reported analogous spatial distribution characteristics of cells: both OSECs and Pre-GC II are predominantly located in the fetal ovarian cortical region, whereas Pre-GC I is more frequently found in the medullary region⁴ (**see details in lines 274-283 of the revised manuscript**). The supporting cell lineage exhibits a high degree of conservation in spatial localization across species. Similarly, the spatial distribution of different germ cell types is also highly conserved¹ (**Response Figure 14c-e**). For instance, in the E50 porcine gonad, Meiotic Oogonia are positioned closer to the medullary region than RA Oogonia (**Response Figure 14d**). However, a key difference was identified: Mitotic PGCs are almost negligible in porcine E50 (**Response Figure 14e**), whereas in humans, these cells persist at 14 weeks and, according to literature, continues until 20 weeks¹¹ (**see details in lines 209-220 of the revised manuscript**). Furthermore, during germ cell specification, the intensity and temporal pattern of the BMP signaling pathway exhibit notable species-specific variations, particularly in the timing of downstream target gene *ZGLP1* expression. In mice^{12, 13} and humans¹⁴, *ZGLP1* primarily functions at the RA-responsive stage, whereas in pigs, its expression persists into the meiotic phase. These findings indicate that although BMP signaling plays a critical role in the differentiation of primordial germ cells into oogonia across multiple species, the duration of its activation may vary among different organisms (**see details in lines 534-539 of the revised manuscript**).

Response Figure 14: Comparative analysis of spatial transcriptomes in female gonads of humans and pigs. **a**, Timeline of oogenesis and sampling strategy for cross-species analysis in pigs and humans. **b**, Spatial localization of three human supporting cell lineage subgroups at 14 gestational weeks. The dashed box indicates the enlarged view of the corresponding area. Scale bar: 500 μ m (zoomed-out), 100 μ m (zoomed-in). **c**, Spatial visualization displays the spatial localization of three subgroups of porcine FGCs. The germ cells at the three different stages were represented by different colors. The dashed box indicates the enlarged view of the corresponding area. The outer cortical, inner cortical, and medullary regions of the gonads are delineated by dashed lines. The image shown is

from a single representative sample (the replicate is presented in the **Supplementary Fig. 3c**). Scale bar: 500 μm (zoomed-out), 100 μm (zoomed-in). **d**, Kernel Density Plot shows the spatial distance of two fetal germ cell subpopulations (RA Oogonia and Meiotic Oogonia) from a defined tissue boundary in the E50 female gonad. The x-axis represents the Distance from Boundary (in μm), and the y-axis represents the Density (probability distribution). **e**, Spatial localization of three human supporting cell lineage subgroups at 14 gestational weeks. The dashed box indicates the enlarged view of the corresponding area. Scale bar: 500 μm (zoomed-out), 100 μm (zoomed-in).

References:

1. Anand, R.P. et al. Design and testing of a humanized porcine donor for xenotransplantation. *Nature* **622**, 393-401 (2023).
2. Zhu, Q. et al. Specification and epigenomic resetting of the pig germline exhibit conservation with the human lineage. *Cell Rep* **34**, 108735 (2021).
3. Chen, M. et al. Integration of single-cell transcriptome and chromatin accessibility of early gonads development among goats, pigs, macaques, and humans. *Cell Rep* **41**, 111587 (2022).
4. Niu, W. & Spradling, A.C. Two distinct pathways of pregranulosa cell differentiation support follicle formation in the mouse ovary. *Proc Natl Acad Sci U S A* **117**, 20015-20026 (2020).
5. Mayère, C. et al. Origin, specification and differentiation of a rare supporting-like lineage in the developing mouse gonad. *Sci Adv* **8**, eabm0972 (2022).
6. Wamaitha, S.E. et al. Single-cell analysis of the developing human ovary defines distinct insights into ovarian somatic and germline progenitors. *Dev Cell* **58**, 2097-2111.e2093 (2023).
7. Saitou, M., Nagano, M. & Mizuta, K. Mechanisms of human germ cell development. *Nat Rev Mol Cell Biol* (2025).
8. Sasaki, K. et al. The embryonic ontogeny of the gonadal somatic cells in mice and monkeys. *Cell Rep* **35**, 109075 (2021).
9. Stévant, I. et al. Deciphering Cell Lineage Specification during Male Sex

Determination with Single-Cell RNA Sequencing. *Cell Rep* 22, 1589-1599 (2018).

10. Nef, S., Stévant, I. & Greenfield, A. Characterizing the bipotential mammalian gonad. *Curr Top Dev Biol* 134, 167-194 (2019).

11. Garcia-Alonso, L. et al. Single-cell roadmap of human gonadal development. *Nature* 607, 540-547 (2022).

12. Nagaoka, S.I. et al. ZGLP1 is a determinant for the oogenic fate in mice. *Science* 367 (2020).

13. Cheung, F.K.M. et al. BMP and STRA8 act collaboratively to ensure correct mitotic-to-meiotic transition in the fetal mouse ovary. *Development* 152 (2025).

14. Li, L. et al. Single-Cell RNA-Seq Analysis Maps Development of Human Germline Cells and Gonadal Niche Interactions. *Cell Stem Cell* 20, 858-873.e854 (2017).

2. Based on Fig. 1e, the authors discuss the migration of several cell types. However, it is difficult for this reviewer to understand the authors' argument only on images of whole region. Could the authors provide magnified or cell-type specific images to make the data easier to interpret?

Response: Thank you for your valuable suggestion. We agree that the panoramic image in Fig. 1e does not clearly demonstrate the migration of different cell types, and this is a valid point. Clear identification of cell types is indeed essential to support our conclusions. To thoroughly address this issue, we have refined the color scheme to enhance contrast. Additionally, magnified views along with annotations of the anterior and posterior, as well as the cortical and medullary regions, have been included to better illustrate the spatial architecture (**Response Figure 1**). We believe these additions and refinements significantly improve the visual presentation and strengthen the support for our findings.

Response Figure 1: Spatial mapping of cell populations: Cellular distributions were reconstituted on the microarray coordinate system through HDst-seq barcode alignment. The dashed box indicates the enlarged view of the corresponding area. The outer cortical, inner cortical, and medullary regions of the gonads are delineated by dashed lines. The image shown is from a single representative sample (the replicate is presented in the **Supplementary Fig. 1e**). Scale bar: 500 µm (zoomed-out), 100 µm (zoomed-in) (as shown in Fig. 1e).

3. Even though the authors have the spatial transcriptomics dataset, visualizing gene expression levels spatially is limited. In the mouse, it is well known that the onset of meiosis exhibits an anterior-posterior wave because the mesonephros is the major source of retinoic acid. Therefore, it is useful if the authors can show the such kind of transcriptomic shift. Additionally, discussion about the link between somatic cell differentiation and anterior-posterior wave is also interesting.

Response: Thank you for your highly insightful feedback. We fully concur with your perspective and have revised our manuscript accordingly. Through an in-depth analysis of the spatial dynamics of signaling pathways, we made an intriguing discovery: BMP and RA signals initially co-localize at the gonad-mesonephros interface during early stages, and subsequently segregate spatially—with BMP becoming restricted to the unilateral cortical region, while RA distributes broadly across both sides (**Response Figure 15**). These findings suggest that BMP may initiate oogonial specification, after which both signals cooperate to promote the progression of meiosis (see details in lines 438-449 of the revised manuscript). This also provides new insights into how signaling pathways

precisely regulate organ development through spatiotemporal dynamics. Of course, these results still require further validation in future studies, and we have now incorporated the relevant analyses and discussions into the corresponding sections of the manuscript.

Response Figure 15: BMP and RA signaling pathway activity. a, Spatial visualization of BMP pathway activity in the developing female pig gonad at different time points. **b**, Spatial visualization of RA pathway activity in the developing female pig gonad at different time points. The white arrows indicate the enriched regions of BMP/RA signaling activation. The image shown is from a single representative sample (the replicate is presented in the **Supplementary Fig. 7d**). Max, maximum; min, minimum. Scale bar: 500 μm (as shown in Fig. 6f).

4. Based on Fig. 6 a and b, the authors discuss about the ligand-receptor expression and cell-to-cell communication, focusing mainly on the effect of ligand from supporting cells on oocytes. However, the effect of oocyte-derived ligand on supporting cells, such as BMP2 at E50, are also observed. The authors should discuss this and explain why the authors have ignored it to help readers more easily follow their logic.

Response: Thank you very much for your insightful and constructive suggestion. You rightly pointed out that our data do indicate the potential role of oocyte-derived ligands (such as BMP2) in acting on supporting cells, which was not sufficiently addressed in the discussion section of our initial draft. This oversight may have affected the readers' comprehensive

understanding of the logical flow of our study. We sincerely apologize for this omission.

The modification is as follows (see details in lines 424-429 of the revised manuscript):

Particularly during the E50 stage, the BMP signaling pathway exhibits a significant bidirectional role between germ cells and Pre-GC II/IC-1, suggesting that this pathway not only serves as a mediator through which supporting cells influence germ cells, but may also function as an important tool for germ cells to provide feedback and regulate their microenvironment.

5. Line 349-350, the authors discuss about the function of WNT3A in cell-to-cell communication between supporting cell and oocytes. The authors showed WNT6, WNT2B, FZD8, WNT5A, ROR2 and FZD3 violin plot but the violin plot for WNT3A is missing. Could the authors provide WNT3A violin plot?

Response: According to your suggestion, we have now included the expression data of WNT3A in both supporting cell lineage and germ cells (as shown in Fig. 6b).

6. Line 365-367, the authors demonstrated the upregulation of ID family genes in meiotic oogonia. In mouse, as reported in Miyauchi et al., EMBO (2017), Nagaoka et al., Science (2020) and others, BMP signals affect oocytes prior to meiotic initiation. Is this due to differences in species or problems in identifying the cell type? Please discuss about this.

Response: Thank you for raising this profound and constructive comment. You pointed out an important discrepancy between our findings and those from classical mouse model studies. We fully agree on the critical role of BMP signaling in mouse germ cell development, particularly during the mitotic proliferation stage prior to meiotic initiation¹⁻³. In response to your question, we have conducted an in-depth reanalysis of the data and a thorough review of the literature.

First, we have re-optimized our cell clustering and annotation criteria. We confirmed that the "Meiotic oogonia" in this study are clearly defined based on a stringent set of marker genes (such as *SYCP1*, *SPO11*, *DMC1*, etc.), effectively ruling out the possibility of erroneous conclusions due to misclassification of cell types. We clearly observed that the *ID* family genes, downstream targets of BMP signaling, are consistently highly expressed throughout the entire process of germ cell specification from PGCs to oogonia and entry

into meiosis. This finding differs from the pattern in mice, where BMP signaling primarily acts prior to meiosis¹⁻³. To validate the universality of this discovery, we carefully examined published single-cell RNA sequencing data from human fetal ovaries and found that *ID* genes are similarly upregulated during the specification of PGCs into Meiotic oogonia in humans, only beginning to decrease at the oogenesis stage⁴ (**Response Figure 16**). This suggests that sustained activation of BMP signaling may be essential for the progression of meiosis in both pigs and humans.

Notes: The figure was cited from references⁴.

Response Figure 16: For the BMP signaling pathways, the relative expression levels of the target *ID* genes are shown. Black arrows indicate Meiotic oogonia. Mitotic (Mitotic PGC), RA responsive (RA Oogonia), Meiotic (Meiotic Oogonia) and Oogenesis (Oocyte).

In addition, please provide the expression of *ZGLP1* that is known as the critical downstream factor of BMP signal.

Response: Your question regarding *ZGLP1* is of critical importance. Consistent with the expression pattern of *ID* genes we observed, *ZGLP1* is expressed in porcine germ cells starting from the PGC specification stage, and its expression level progressively increases through the subsequent RA oogonia stage and into the Meiotic oogonia stage (**Response Figure 17a**). Therefore, in pigs, *ZGLP1*, as a "key downstream factor" of BMP signaling, is indeed continuously expressed, which aligns perfectly with the expression pattern of *ID* genes and forms a complete logical chain. This strongly supports the conclusion that "BMP signaling plays a sustained role in porcine germ cell development." Furthermore, the expression pattern of *ZGLP1*, transient in mice but sustained in pigs, further indicates that there may be distinct species-specific differences during the meiosis stage.

To further verify the reliability of our data, we incorporated published single-cell transcriptomic data from pigs (E45–E75) as supplementary evidence and identified three

stages of germ cells: RA oogonia, Meiotic oogonia, and oocytes⁵ (**Response Figure 17b,c**). Consistent with our findings, *ZGLP1* also showed the highest expression during the meiotic stage and began to decrease in oocytes (**Response Figure 17d,e**). Unfortunately, our data did not cover the oocyte stage. However, the trend observed from RA oogonia to meiotic oogonia is consistent with the data presented in our study. Based on this, we propose that sustained expression of *ZGLP1* in pigs represents a species-specific characteristic. This aspect diverges from observations in mice and humans, a point that has been emphasized in the manuscript (**see details in lines 534-539 of the revised manuscript**).

Response Figure 17: scRNA-seq of the developing porcine female gonad. **a**, Dot plot presents the characteristic genes expressed by germ cells at the three different stages. The dotted box indicates the expression of *ZGLP1* in Mitotic PGCs, RA oogonia, and Meiotic oogonia. The size of the dots represents the percentage of cells expressing each

gene, and the color coding is used to indicate the gene expression level. **b**, UMAP visualization of single-cell clustering: Distinct cell populations of germ and somatic cells in developing female porcine gonads. Dashed area indicates fetal germ cell cluster. **c**, UMAP analysis reveals three fetal germ cell subpopulations: RA Oogonia, Meiotic Oogonia and Oocyte. **d**, Dot plot presents the characteristic genes expressed by germ cells at the three different stages. The size of the dots represents the percentage of cells expressing each gene, and the color coding is used to indicate the gene expression level. **e**, Dot plot shows the expression of BMP-related *ID* genes and *ZGLP1* in FGCs. The dot diameter indicates the proportion of expressing cells, and the color gradient reflects the expression level.

References:

1. Miyauchi, H. et al. Bone morphogenetic protein and retinoic acid synergistically specify female germ-cell fate in mice. *Embo Journal* **36**, 3100-3119 (2017).
2. Nagaoka, S.I. et al. ZGLP1 is a determinant for the oogenic fate in mice. *Science* **367** (2020).
3. Cheung, F.K.M. et al. BMP and STRA8 act collaboratively to ensure correct mitotic-to-meiotic transition in the fetal mouse ovary. *Development* **152** (2025).
4. Li, L. et al. Single-Cell RNA-Seq Analysis Maps Development of Human Germline Cells and Gonadal Niche Interactions. *Cell Stem Cell* **20**, 858-873.e854 (2017).
5. Ge, W. et al. Spatiotemporal dynamics of early oogenesis in pigs. *Genome Biol* **26**, 2 (2025).

Minor points to address

1. Line 158, γ H2AX is not a specific marker for leptotene stage of meiotic cells, it also marks the zygotene stage of meiotic cells. Therefore, “leptotene stage of meiotic...” should be replaced with “leptotene and zygotene stage of meiotic...”.

Response: Thank you for your thorough review of the manuscript and for your valuable feedback. Regarding the description of γ H2AX labeling in line 158, I fully agree with your observation that γ H2AX marks not only the leptotene stage but also the zygotene stage of meiotic cells. Following your suggestion, I have revised the text by replacing “leptotene stage of meiotic...” with “leptotene and zygotene stage of meiotic...”.

The modification is as follows (see details in lines 163-165 of the revised manuscript):

To evaluate meiotic initiation during gonadal development, we next performed immunofluorescence at E35 and E50 for the oogonia marker DAZL as well as the leptotene and zygotene stage marker γ H2AX, respectively.

2. Line 180-192, the figures should be labelled in the order in which they appear in the main text.

Response: We have checked and reorganized the order in which the panels of Fig. 2 are cited in the main text. All panels are now referenced in accordance with their logical sequence of appearance in the Results section, ensuring a coherent narrative flow. We have also verified that the panel order within the supplementary figure itself aligns with the citations in the text.

3. There is no reference about the BMP function on oocyte differentiation. The authors should refer the paper such as Wu et al. PLoS Biology (2016), Miyauchi et al., EMBO (2017), Nagaoka et al., Science (2020) and Cheung et al., Development (2025).

Response: Thank you very much for your valuable suggestion. The references you pointed out regarding the role of the BMP signaling pathway in oocyte differentiation were indeed a significant omission in our initial draft. These references are crucial for strengthening the background and discussion sections of our paper. We have carefully studied all the recommended papers and have revised our manuscript accordingly. Once again, we sincerely appreciate your guidance and assistance.

References (see details in references of the revised manuscript):

27. Nagaoka, S.I. *et al.* ZGLP1 is a determinant for the oogenic fate in mice. *Science* **367** (2020).

64. Wu, Q. *et al.* Sexual Fate Change of XX Germ Cells Caused by the Deletion of SMAD4 and STRA8 Independent of Somatic Sex Reprogramming. *PLoS Biol* **14**, e1002553 (2016).

65. Cheung, F.K.M. *et al.* BMP and STRA8 act collaboratively to ensure correct mitotic-to-meiotic transition in the fetal mouse ovary. *Development* **152** (2025).

67. Miyauchi, H. *et al.* Bone morphogenetic protein and retinoic acid synergistically specify female germ-cell fate in mice. *Embo Journal* **36**, 3100-3119 (2017).

Reviewer #4 (Remarks to the Author):

In this manuscript by He and colleagues, the authors produced a spatial transcriptome dataset of the developing pig ovary at high spatial and transcriptome resolution. This dataset spans multiple developmental timepoints from E24 to E50 that are relevant for ovary development, including PGC specification and entry of oogonia into meiosis. The authors also identified in a spatially resolved manner the two waves of pre-granulosa cells.

With this dataset, the authors interrogated the dynamics of

While the data are high quality and this manuscript promises to become an important resource for the field of ovarian biology, this remains a transcriptome dataset that does not provide functional evidence for the authors' sometimes lofty conclusions, and there are several instances where the manuscript could be improved.

Response: We thank the reviewer for their valuable insights. In response, we have clarified that the primary contribution of this study lies in providing a high-quality spatial transcriptomic data resource. Accordingly, we have revised and toned down any speculative interpretations to ensure that all conclusions are appropriately aligned with the strength of the data evidence.

Suggestions to authors

I have listed below specific suggestions and recommendations to improve the quality of the manuscript and to clarify the impact of the data presented.

Major comments / limitations

- In the abstract and manuscript the authors claim that they “clarified the roles of the two waves of pre-granulosa cells in regulating germ cell development” and that they “identified the mesonephros as the source of endothelial and interstitial cells along with their specific contributions to gonadal development.” They further claim that “cell-cell communication analysis revealed the critical role of BMP signaling (BMP2, GDF5) in driving the specification of PGCs into oogonia and their subsequent entry into meiosis.” I have several problems with these claims:

-- The data presented are descriptive, based on expression of specific transcripts (and some protein level validation) in wild-type pig ovaries. These data are merely

correlations, and do not allow such strong conclusions on the role of specific cell populations or pathways in development of the ovary. I recommend the authors either provide functional evidence for these conclusions (using in vitro systems with small molecule inhibitors or cell ablation methods for example), or that they tone down their conclusions and clearly identify in the results and discussion that these data provide correlations that may indicate function, but this would need to be tested using functional experimental approaches. This should be addressed throughout the text. For example line 301 to 303, the summary suggests functional involvement of angiogenesis, ECM remodeling and myogenesis, but this was not functionally tested, just suggested by transcript expression. Same on line 338-340, with the additional caveat that this has already been shown in other animal models.

Response: Thank you very much for your valuable and insightful comments. You have rightly pointed out a central limitation of our paper—namely, that overly strong functional conclusions were drawn from correlational data. We fully agree that descriptive data alone are insufficient to establish causality. We have thoroughly and carefully revised the entire manuscript to temper the conclusiveness of our language.

The modification is as follows (see details in lines 349-352 of the revised manuscript):

In summary, these results suggested that the distribution of interstitial cells in the medulla and endothelial cells in the inner cortex supports their potential functional collaboration, which may be critical for supporting female germ cell development.

The modification is as follows (see details in lines 392-396 of the revised manuscript):

In summary, these results indicate that the supporting cell lineages and interstitial cells interact with germ cells in a spatial context, suggesting that this cell population may play a more critical role during the specification of PGCs into oogonia. Furthermore, LAMA1, as a key component of the ECM, provides structural support by enveloping the germ cells.

We also recognize that relying solely on transcriptomic similarity data without additional functional validation, such as knockdown studies or co-localization analyses, limits our ability to determine whether the observed expression patterns reflect shared functions, common regulatory origins, or potential interactions. Within the current framework of our study, we are unable to draw definitive conclusions on this matter. Therefore, we have

removed these specific results and have carefully revised the entire manuscript to streamline the logical flow and improve consistency. In addition, we have incorporated background information on germ cell epigenetics to enhance the readability of the text (**see details in lines 171-177 and 187-194 of the revised manuscript**).

In addition to optimizing the content of the manuscript, this study also incorporated published spatial transcriptomic data from human female fetal gonads at 14 weeks and conducted a cross-species comparative analysis based on this data (**see details in lines 209-219 and 274-283 of the revised manuscript**). The results highlight the conservation and divergence in the spatial distribution patterns of germ cells and supporting cell lineages in the gonads between pigs and humans. We believe that these additions and analyses contribute to enhancing the academic quality of the manuscript.

-- Some of these conclusions (2 waves of granulosa cells, mesonephric contribution to vasculature, role of BMP2 in oocyte development) have already been made using functional experiments in mouse ovaries. The fact that these same pathways and developmental events may be conserved between mouse and pig and function in similar ways is novel and exciting, and I would argue that the authors could do a better job recognizing these similarities and using them to make more solid conclusions.

Response: Thank you very much for your insightful and constructive suggestion. To draw more reliable conclusions, we have conducted additional analyses using published data on human fetal female gonads at 14 weeks. The results suggested that the spatial organization of key lineages such as germ cells and supporting cells is highly conserved between pigs and humans. In addition, we have included a discussion on the similarities and differences in the BMP signaling pathway across species (mouse, pig, and human). We hope these refinements will further enhance the quality of our manuscript (**see details in lines 209-219 of the revised manuscript**).

- I did not see a materials and methods section in the manuscript, it seems all methods were pushed to supplemental. I find this extremely confusing and does not favor data accessibility and reproducibility. I recommend putting the materials and methods in the main manuscript.

What is the spatial resolution (size and spacing of spots)?

Response: We sincerely thank the reviewer for this important and constructive comment. We fully agree that placing the entire “Materials and Methods” section only in the Supplementary Information hampers readability and reproducibility. Following your suggestion, we have now moved the complete “Materials and Methods” section to the main text.

Our approach achieves a spatial resolution of 10 μm per channel with 10 μm Interval between channels. Following two orthogonal rounds of array coverage (spatial barcodes in both horizontal and vertical directions), the final size of each analysis unit (bin) is a square of 10 μm \times 10 μm . Crucially, the Interval between these 10 μm bins is also 10 μm (**Response Figure 18**) (see details in lines 94-97 of the revised manuscript).

Response Figure 18 Chip functional zone magnification display: The channels and interval widths of the functional area determine the resolution of the tissue spatial map. The small circular rings represent the sample loading holes and outlet holes; the black lines represent the channel walls through which the solution reagents flow and the interval walls separating the channels; the red dashed arrows indicate the channels and intervals of the functional area; the yellow, blue, and purple lines indicate the interval walls, with two lines forming Interval 1, Interval 2, and Interval 3; the wavy line between Channel 1 and Channel 2 represents the flowing reagent solution. Both the channel width and the intervals are 10 μm .

- Lines 93-95: **“Following stringent quality control, 50,630 high-quality spatial pixels were retained for constructing the porcine female gonadal spatiotemporal**

transcriptomic atlas". The authors should provide more detail about the QC steps, what was defined as "high quality" pixels, what is the difference between the QC-filtered and the raw datasets?

Response: Thank you for your valuable comment regarding the stringency and details of our quality control (QC) procedures. We are pleased to provide a comprehensive explanation of the steps taken to define "high-quality spatial pixels" and the differences between the QC-filtered and raw datasets.

Our QC process was primarily divided into two major components: processing of the raw sequencing data and quality filtering of the resulting gene expression matrix.

Sequencing Data Processing and Transcript Capture: For the raw sequencing data, we performed the following steps: We utilized Cutadapt to remove sequencing adapters. We used UMI-tools to extract reads containing the two-round spatial barcodes (Barcode B & Barcode A) and the Unique Molecular Identifier(UMI). Crucially, we modified the Genome Annotation File (GTF file) to expand the alignment scope from the default coding sequences to the entire transcript. This allowed us to effectively capture unprocessed nuclear mRNA sequences, including those containing introns. The initial gene expression matrix was then generated using the stpipeline processing workflow.

Gene Expression Matrix QC and Spatial Filtering: The second part involved stringent quality control applied to each individual spatial transcriptomics slide: **Pixel-Level Filtering:** For each slide, we filtered out pixels (spots) that had total UMI counts below a range of 300-600 or feature counts (number of expressed genes) below a range of 150-250. The specific threshold varied slightly per slide to optimize data quality.

Rationale for Doublets and Singlets: Given that our sequencing-based spatial transcriptomics data does not represent strict single cells (each pixel is a spot encompassing multiple cells), we deliberately omitted the step of removing potential 'doublet' spots, as this step is less relevant and could lead to unnecessary data loss.

Manual Tissue Cropping (Spatial Filtering): Following the automated filtering, we manually cropped the gene expression matrices based on the corresponding bright-field images of the tissue sections. This step eliminated pixels located on non-tissue regions or background, ensuring that only data points originating from the gonadal tissue were

retained.

Comparison to Raw Data: By implementing these comprehensive steps, particularly the expanded transcript capture and the manual spatial cropping, we obtained a QC-filtered dataset that, compared to the raw data, exhibits: Fewer spurious noise spots. More complete and accurate tissue morphology. Higher data quality and biological relevance, suitable for constructing a robust spatiotemporal atlas. We believe these details fully clarify the rigorous QC procedures applied and the definition of the final 50,630 high-quality spatial pixels.

Spatial and Clustering Visualization Comparison: Furthermore, the necessity of our QC steps is strongly supported by the comparative spatial and clustering visualization analysis. When comparing the spatial display of data before and after QC, it is evident that the raw data contained significant background noise in both UMI counts and feature counts. This background noise would introduce a large number of non-tissue pixels into our subsequent processing, severely compromising data quality (**Response Figure 19a-c**). To further demonstrate this impact, we subjected both the raw (unfiltered) and the QC-filtered datasets to the identical downstream pipeline, including SCT normalization, RPCA batch correction, and dimensionality reduction. The unfiltered UMAP plot exhibits a highly mixed and ambiguous distribution, making it impossible to clearly delineate distinct cell types (**Response Figure 19b**). In contrast, the QC-filtered UMAP plot yields well-separated and biologically meaningful clusters (**Response Figure 19d**), enabling the precise identification of various cell populations. This comparison definitively validates the stringency of our QC procedures, which were essential for obtaining a robust dataset capable of resolving the complex cellular architecture of the fetal porcine gonad.

Response Figure 19 a, Spatial distribution of counts in porcine female fetal gonads (E24–E50) before and after quality control, illustrating the removal of background noise and non-tissue pixels. **b**, UMAP visualization displays the spatial transcriptomics clustering status before quality control QC: The highly disorganized distribution of germ and somatic cells within the developing porcine female gonad. **c**. Spatial distribution of features in porcine female fetal gonads (E24–E50) before and after quality control, illustrating the removal of background noise and non-tissue pixels. **d**, UMAP visualization of spatial transcriptomic clustering after quality control, showing well-separated and biologically distinct cell populations.

- Lines 126-127 “both endothelial cells and interstitial cells shared a corticomedullary migration pattern”. This conclusion seems like a stretch, how did the authors determine this was a migration pattern vs proliferation of endothelial and interstitial cells toward the medulla? These two possibilities should be addressed and/or discussed.

Response: Thank you for raising this important point. We fully agree that distinguishing between cell migration and localized proliferation is crucial for interpreting our observations. Due to current technical limitations, we are unable to directly demonstrate cell migration

behavior through experiments such as live-cell imaging in this study. Therefore, following your suggestion, we have carefully refined our wording to ensure a more rigorous and comprehensive discussion in the manuscript (**see details in lines 520-525 of the revised manuscript**). We truly appreciate your valuable feedback, which has significantly enhanced the quality of our paper.

- Line 166, the authors investigate the methylation of DNA in germ cells. This experiment is interesting but could be better introduced. In its current form it is not immediately clear why this assay is relevant. Authors should also define 5mC and 5hmC. Similar comment for line 172, why is it relevant to investigate H3K27me3 marks?

Response: During mammalian early development and germ cell formation, the entire genome undergoes dramatic "erasure-reestablishment-re-erasure" reprogramming of DNA methylation¹. Specifically, primordial germ cells (PGCs) initiate extensive DNA demethylation to reset the epigenetic state, thereby establishing the foundation for gametogenesis. DNA methyltransferases *DNMT3A* and *DNMT3B* establish de novo DNA methylation patterns, while *DNMT1* functions as the maintenance methyltransferase, propagating methylation patterns following DNA replication². DNA demethylation can occur through passive mechanisms, involving replication-dependent dilution, or active pathways. Active demethylation is primarily mediated by the TET family (*TET1*, *TET2* and *TET3*) of dioxygenases, which catalyze the stepwise oxidation of 5mC to intermediates such as 5hmC, 5fC, and 5caC³. These oxidized derivatives are ultimately replaced with unmodified cytosine through the base excision repair pathway. As a key intermediate in active DNA demethylation, 5-hydroxymethylcytosine (5hmC) plays a dual role in epigenetic reprogramming of PGCs. Studies demonstrate that 5hmC levels exhibit highly dynamic spatiotemporal specificity: during early developmental stages (e.g., human PGCs at week 4, mouse E9.5 PGCs), it is notably enriched and functions as a pioneer in global demethylation; whereas during later reprogramming phases (e.g., human PGCs at week 9, mouse E13.5 PGCs), its global levels are markedly reduced^{4, 5}. Previous studies have reported the coexistence of 5mC and 5hmC signals in migrating PGCs at embryonic day 14 (E14) in pigs⁶. To investigate the dynamics of these modifications after PGCs colonize

the gonads, we further examined the levels of 5mC and 5hmC in gonadal germ cells at E24. The results showed that both 5mC and 5hmC signals were no longer detectable in E24 gonadal germ cells, indicating that their erasure is completed before E24. In PGCs, epigenetic reprogramming is not limited to the erasure and reestablishment of DNA methylation but also encompasses a series of crucial histone modifications, together forming a tightly coordinated regulatory network. Among these modifications, H3K27me3 serves as a key repressive mark and plays distinct and complementary roles during the development of primordial germ cells. During human primordial germ cell (hPGCs) specification, the global DNA hypomethylation is accompanied by extensive reprogramming of histone modifications⁷. These processes occur simultaneously, reflecting the synergy and complexity of epigenetic regulation in germ cell development. Immunofluorescence staining revealed a unique pattern: the repressive marker H3K27me3 exhibited stronger signals in migrating hPGCs than in the surrounding somatic cells, but this signal was no longer detectable between weeks 7 and 9. In this study, we detected H3K27me3 in germ cells at stages E24 and E27 and observed its progressive loss as development proceeded to E50. This dynamic pattern closely resembles the H3K27me3 changes reported in mouse and human germ cells. In response, we have added background information on 5mC and 5hmC in the section on DNA demethylation, as well as context on histone modifications in the H3K27me3 section (**see details in lines 171-177 and 184-194 of the revised manuscript**). We hope this supplementary information will help readers better understand our study.

References:

1. Lee, S.M. & Surani, M.A. Epigenetic reprogramming in mouse and human primordial germ cells. *Exp Mol Med* **56**, 2578-2587 (2024).
2. Zhou, W., Liang, G., Molloy, P.L. & Jones, P.A. DNA methylation enables transposable element-driven genome expansion. *Proc Natl Acad Sci U S A* **117**, 19359-19366 (2020).
3. Hsu, F.M. et al. TET1 facilitates specification of early human lineages including germ cells. *iScience* **26**, 107191 (2023).
4. Hackett, J.A. et al. Germline DNA demethylation dynamics and imprint erasure

through 5-hydroxymethylcytosine. *Science* **339**, 448-452 (2013).

5. Guo, F. et al. The Transcriptome and DNA Methylome Landscapes of Human Primordial Germ Cells. *Cell* **161**, 1437-1452 (2015).

6. Zhu, Q. et al. Specification and epigenomic resetting of the pig germline exhibit conservation with the human lineage. *Cell Rep* **34**, 108735 (2021).

7. Borensztein, M. et al. Contribution of epigenetic landscapes and transcription factors to X-chromosome reactivation in the inner cell mass. *Nat Commun* **8**, 1297 (2017).

8. Saitou, M., Nagano, M. & Mizuta, K. Mechanisms of human germ cell development. *Nat Rev Mol Cell Biol* (2025).

- Line 187: how did authors calculate the distance between FGCs and the cortex, and how did they normalize for dorsal-ventral localization of the section plane. I.e. if the section is closer to the surface of the ovary, the distance will be smaller than if the section is taken on the dorsal side of the ovary which is mostly medulla.

Response: We serially sectioned the entire gonad. Instead of measuring distances from random sections, we specifically selected mid-plane sections for our quantitative analysis. These mid-plane sections are those that pass through or near the central axis of the ovary, typically characterized by the presence of the central medulla and a continuous, well-defined cortex surrounding it. This ensures that the anatomical context (the relative proportion of cortex and medulla) is consistent across all samples being measured.

By analyzing these central sections, we effectively "normalize" for the dorsal-ventral localization. Since all measurements are taken from a similar anatomical plane across all ovaries, the distances measured are directly comparable. This approach minimizes the bias that would arise from comparing a section near the thin ventral surface to a section through the thick dorsal medulla, thereby significantly strengthening the validity of our conclusions. The sections selection and analysis details are described in the method section (see details in lines 816-827 of the revised manuscript).

Figures

- The authors state in line 93, that 2-3 biological replicates were sequenced, but they only show one section per stage. Were these integrated together on a single spatial map as shown in Fig. 1, or is fig 1e depicting a single replicate? I recommend

clarifying this in the text and legend, and showing images of the transcriptome for all biological replicates. A discussion of the variability in spatial transcriptome between individuals would be interesting as well.

For the spatial images in Fig. 1e, I recommend adding some indication of the orientation of the tissue (anterior / posterior pole, cortex vs medulla) to better orient the reader to these images.

Response: We sincerely thank the reviewer for these valuable and constructive suggestions, which have significantly enhanced the rigor and clarity of our manuscript. To fully demonstrate data reproducibility, we have included spatial transcriptomics images from all biological replicates for each stage as supplementary figures, with detailed descriptions **in the corresponding legends**.

Your observation regarding the significance of different slices is highly valuable. In E50, we also noted that the two slices presented exhibit subtle differences. However, the core spatial structural features remain consistently present across all samples, which further demonstrates the robustness of the spatial transcriptomics approach presented in this study (**see details in lines 320-325 of the revised manuscript**).

We have optimized Figure 1e by adding clear directional labels, including anterior/posterior and cortical/medullary regions, to each spatial transcriptome image. With these labels, readers can now more easily correlate the spatial transcriptomic data with the gonadal anatomy, significantly enhancing both the readability and scientific rigor of the figures (**Response Figure 1**).

Response Figure 1: Spatial mapping of cell populations: Cellular distributions were

reconstituted on the microarray coordinate system through HDst-seq barcode alignment. The dashed box indicates the enlarged view of the corresponding area. The outer cortical, inner cortical, and medullary regions of the gonads are delineated by dashed lines. **The image shown is from a single representative sample (the replicate is presented in the Supplementary Fig. 1e).** Scale bar: 500 μm (zoomed-out), 100 μm (zoomed-in) (as shown in Fig. 1e).

- I recommend using colorblind-friendly color schemes in the IF images instead of red and green, to increase accessibility of the manuscript.

Response: We sincerely thank the reviewer for raising this critical point regarding the accessibility of our figures. In direct response to this comment, we have replaced the red color in all immunofluorescence (IF) images with a purple color.

- A few typos: Fig 1a4, “library”; Fig1a3, chip orientation should say “B1-B96” and A1-A96; Fig 1c y-axis title, “fraction”;

Response: We sincerely apologize for these oversights. The expert is absolutely correct. We have carefully corrected all these typos in the revised version of Figure 1: “libary” has been changed to “library”.

The chip orientation labels have been updated to “B1-B96” and “A1-A96”.

The y-axis title has been corrected to “fraction”.

Furthermore, we have taken this opportunity to perform a thorough proofreading of the entire manuscript to ensure no similar errors remain.

- Fig 2D and E, where in the tissue were these panels taken? I recommend showing the entire section in addition to the zoom up to show distribution of BLIMP1, DAZL, H2AX cells across the tissue. Same comment for Fig3c and fig6d

Response: Thank you for your valuable suggestions, and we have systematically supplemented and optimized the relevant figures accordingly. For Figures 2d and 2e, we have supplemented low-magnification images and magnified views to clearly illustrate the different types of germ cells and their spatial distribution patterns (**Response Figure 20a, b**). Similarly, for Figure 3c, a low-magnification image has been provided, with annotations marking the cortical and medullary regions to facilitate readers' understanding of the spatial localization characteristics of the supporting cell lineage(**Response Figure 20c**).

Furthermore, to more intuitively demonstrate the nuclear localization of pSMAD1/5, higher-magnification images along with separate DAPI channel images have been included for Figure 6d (**Response Figure 20d**). These adjustments are intended to further improve the quality of figure presentation and enhance the readability of the results.

Response Figure 20: Optimization of immunofluorescence staining images. **a**, Immunofluorescence images of BLIMP1 (green) and DAZL (purple) in porcine female gonadal sections from different timepoints. Nuclei were counterstained with DAPI (blue). The dashed box indicates the enlarged view of the corresponding area. Scale bars, 100 μm (zoomed-out), 50 μm (zoomed-in). **b**, Immunofluorescence images of DAZL (purple) and γH2AX (green) in porcine female gonadal sections from different timepoints. Nuclei were counterstained with DAPI (blue). The dashed box indicates the enlarged view of the corresponding area. Scale bars, 100 μm (zoomed-out), 50 μm (zoomed-in). **c**, Immunofluorescence images of KRT19 (purple) and GATA4 (green) in porcine female gonadal sections from different timepoints. Nuclei were counterstained with DAPI (blue). The dotted line demarcates the boundary between the gonad cortex and medulla; the white arrow indicates the ovarian surface epithelium (OSE). Scale bars, 100 μm . **d**,

Immunofluorescence images of DDX4 (purple) and p-SMAD1/5 (green) in porcine female gonadal sections from different timepoints. Nuclei were counterstained with DAPI (blue), Scale bar: 20 μm (as shown in Fig. 2d, e, Fig. 3f and Fig. 6h).

Secondary / minor comments

- **Gene / protein annotation seems inconsistent. For example, line 147 IF for *BLIMP1* and *DAZL*, should not be italicized since this is looking at protein.**

Response: Thank you for your valuable observation regarding the inconsistency in gene and protein nomenclature. You are absolutely correct that when detecting proteins using techniques such as immunofluorescence (IF), their names should not be italicized. In accordance with your suggestion, we have systematically checked and revised the entire manuscript (see details in lines 151-154 of the revised manuscript).

- **What method was used for pseudotime analysis? This should be mentioned in the results at line 149.**

Response: Thank you for your suggestion. We have now specified that the pseudotime analysis was performed using Monocle3. As recommended, we have included this information in the Results section. The detailed description of the analysis parameters is provided in the *Methods* section (see details in lines 771-777 of the revised manuscript).

The modification is as follows (see details in lines 154-156 of the revised manuscript):

We also reconstructed the differentiation trajectory of FGCs including Mitotic PGCs, RA Oogonia, and Meiotic Oogonia and computed pseudotime using Monocle 3.

- **What method was used for ligand-receptor analysis? This should be mentioned in the results at line 344.**

Response: Thank you for this valuable feedback. As suggested, we have specified that the ligand-receptor interaction analysis was performed using CellChat. The detailed methodology is now described in the *Methods* section (see details in lines 796-805 of the revised manuscript).

The modification is as follows (see details in lines 401-403 of the revised manuscript):

The ligand-receptor interaction analysis performed with CellChat evaluated interactions between germ cells and both their surrounding supporting cell lineages and interstitial cells.

Point-by-point response to reviewers' comments:

Editor:

Thank you for revising your manuscript entitled "A spatiotemporal transcriptomic atlas of porcine female early gonadal development" in response to referees comments; the revised manuscript has been seen again by the 4 reviewers.

Response: Thank you for your time in reviewing our revised manuscript again and for providing valuable guidance. We are very pleased to learn that the four reviewers have expressed their overall approval of the revised version.

You will see from their comments below that they all largely approve of the new version; however, Reviewers #1 and #3 left a few minor comments to address.

Response: We have carefully addressed the specific suggestions provided by both reviewers and have made corresponding revisions and refinements to the textual expressions, clarification of results, and annotations in figures and tables throughout the manuscript.

Additionally, since one of the reviewers implied in confidential comments to the editor that they felt more of the Response Figures which have currently only been included in the response to reviewers document should be included in the manuscript itself, **we would ask that you please add at least some of the information currently only contained in the Response Figures (it appears to us this would be Response Figures 3, 11, 16, 17, 18, and 19 as well as parts of Response Figure 14) to the further revised manuscript.**

Response: We fully agree with your suggestion and have implemented it by integrating the data and information from the reply materials that provide important support for the main text into both the main manuscript and the supplementary figures.

The content from **Response Figure 3** (analysis of the gonadal- mesonephric complex) has been integrated into **Supplementary Fig. 4 c, d** of the manuscript and cited in the relevant section of the main text.

The content from **Response Figure 17** (validation of BMP downstream target genes and *ZGLP1* expression in published single-cell data) has been incorporated into **Supplementary Fig. 7 d–f** of the manuscript.

The content from **Response Figure 19** (data quality control analysis) has been included in **Supplementary Fig. 1 d** of the manuscript.

All key findings from **Response Figure 14** have been presented in the manuscript as **Fig. 2 f–h, Fig. 3 g, and Supplementary Fig. 3 f**.

As you know, the **Response Figures 11 and 16** were directly sourced from formally published papers. In compliance with copyright norms and to uphold academic integrity by avoiding self-plagiarism, we are unable to directly reproduce these published charts in this article. As an alternative and more rigorous academic approach, we have clearly referenced these published sources in the relevant discussion and results sections of this manuscript, so that readers may trace and access the complete information.

The corresponding citations have been provided below.

3. Garcia-Alonso, L. *et al.* Single-cell roadmap of human gonadal development. *Nature* **607**, 540-547 (2022).

20. Li, L. *et al.* Single-Cell RNA-Seq Analysis Maps Development of Human Germline Cells and Gonadal Niche Interactions. *Cell Stem Cell* **20**, 858-873.e854 (2017).

We would like you to address these remaining minor concerns in the form of a further revised manuscript before we make a final decision on publication.

Please highlight all changes in the manuscript text file.

Response: We sincerely thank the editors and reviewers for their valuable time and constructive comments. We have carefully considered each suggestion and have incorporated corresponding revisions into the manuscript. All modifications have been highlighted with a **yellow background** in the text for the convenience of both you and the reviewers.

Reviewers' comments:

Reviewer #1 (Remarks to the Author):

The authors have successfully addressed the major concerns from the previous round, significantly improving the quality and scientific rigor of the study. The new comparative analyses and structural reorganization are commendable.

Response: We would like to express our sincere gratitude to you for once again taking the time to review our manuscript and for providing such positive and professional comments on the revised version.

Minor issues:

Line 111: "These FGCs initially exhibited a dispersed distribution but progressively migrated toward cortical regions during gonadal development"

Comment: In the figures, the FGCs appear to be always located close to the cortical region. Please verify this description or the figures.

Response: Thank you for pointing out the possible ambiguity between the description of the spatial distribution of fetal germ cells (FGCs) in the text and the visual impression from the figures. We measured the distance between FGCs and the outer cortex at different developmental time points. From E24 to E50, the average distance between FGCs and the outer cortex indeed decreased significantly over the course of development. At the same time, we fully acknowledge your observation that, as shown in the figures, even at the earlier stages of E24 and E27, most FGCs were already located spatially close to the cortical region. Therefore, describing this specific phase as "from dispersed distribution to migration" may not be sufficiently accurate. Based on the above analysis, and to ensure rigor in expression, we have revised the relevant description in the manuscript.

The modification is as follows (see details in lines 110-112 of the revised manuscript):

During gonadal development from E24 to E50, FGCs continue to proliferate and migrate to accumulate in the gonadal cortical region.

Line 117: "Spatial visualization revealed a stratified organization of clusters 2 and 3 relative to OSECs at E24–E27"

Comment: The term "stratified organization" is unclear in this context. Please describe better where and how this stratification occurs, as it is not immediately evident.

Response: Thank you for your thorough and constructive review. We fully agree with your perceptive comment regarding the unclear description of "stratified organization" on line 117. Following your valuable suggestion, we have revised the text to clearly describe the spatial distribution of the three cell types.

The modification is as follows (see details in lines 117-120 of the revised manuscript):

Spatial visualization revealed a distinct stratified organization in the gonads at E24–E27. Specifically, the OSECs occupied the outer cortical region, with cluster 3 adjacent to it medially, and cluster 2 localized to the medullary region.

Line 119: "Subsequent characterization identified cluster 2 as pre-granulosa cells II (Pre-GCs II) through TOX3 expression, while cluster 3 was classified Pre-GCs I based on WNT6 expression.

Comment: There is a contradiction here. In other parts of the text/figures, Cluster 2 is identified as Pre-GCs I and Cluster 3 as Pre-GCs II. Please verify and correct the identification.

Response: We would like to thank you for pointing out the crucial issue of inconsistent cell cluster labeling in our manuscript. We sincerely apologize for this oversight and have now conducted a thorough review and made consistent corrections throughout the manuscript.

The modification is as follows (see details in lines 120-122 of the revised manuscript):

Subsequent characterization identified cluster 2 as Pregranulosa cells-I (PreGCs-I) through *WNT6* expression, while cluster 3 was classified PreGCs-II based on *TOX3* expression.

Line 121: "(Fig. 1d, e)."

Comment: Based on the context, this citation should likely be Fig. 1b-d.

Response: Thank you for your thorough review of the manuscript and for your valuable feedback. Your observation regarding the inconsistent figure citations was precise and professional. We have addressed this issue by revising "(Fig. 1d, e)" to "(Fig. 1b-d)" in accordance with your suggestion.

The modification is as follows (see details in lines 120-122 of the revised manuscript):

Subsequent characterization identified cluster 2 as Pregranulosa cells-I (PreGCs-I) through *WNT6* expression, while cluster 3 was classified PreGCs-II based on *TOX3* expression (**Fig. 1b-d; Supplementary Fig. 1e**).

Line 129: "...interstitial cells marker *NR2F1* and *PDGFRA*, while clusters7"

Comment: Typo. It should read "cluster 7".

Response: Thank you for your careful review and valuable feedback. We completely agree with your observation regarding the writing error on line 129, where "clusters7" should be corrected to "cluster 7." We sincerely apologize for this error. In accordance with your suggestion, we have revised the manuscript to read: "...interstitial cells marker *NR2F1* and *PDGFRA*, while cluster 7..."

The modification is as follows (see details in lines 129-131 of the revised manuscript):

Clusters 4 and 5 both expressed the interstitial cells marker *NR2F1* and *PDGFRA*, while **cluster 7** were identified as endothelial cells due to *PECAM1* and *COL15A1* expression.

Line 185: "trimethylation of histone H3 at lysine 27 (H3K27me3), a repressive histone modification, produced a stronger signal in Mitotic PGCs but sharply declined by E30, becoming undetectable by E35 and E50"

Comment: Please cite the corresponding figure panel immediately after this sentence to support the statement.

Response: Thank you for your careful and professional review. In response to your comment on line 185, we have revised the manuscript in accordance with your suggestion and added the citation to the corresponding experimental results figure at the end of the sentence.

The modification is as follows (see details in lines 186-188 of the revised manuscript):

Interestingly, the trimethylation of histone H3 at lysine 27 (H3K27me3), a repressive histone modification, produced a stronger signal in Mitotic PGCs but sharply declined by E30, becoming undetectable by E35 and E50 (**Supplementary Fig. 3a**).

Line 232: "GATA4 (a zinc finger transcription factor) and WT1 (Wilms tumor 1 protein), key markers of the coelomic epithelium."

Comment: In this context, it would be more precise to indicate that these are markers of the supporting cell lineage, rather than just the coelomic epithelium.

Response: Thank you for your careful and professional review.

In accordance with your suggestions, we have revised the wording at line 232 of the manuscript to more precisely reflect its biological significance. The modified content clearly indicates that GATA4 and WT1 are critical markers for the supporting cell lineage.

The modification is as follows (see details in lines 233-236 of the revised manuscript):

To elucidate supporting cell development, we first performed spatial visualization of GATA4 (a zinc finger transcription factor) and WT1 (Wilms tumor 1 protein), **key markers of the supporting cell lineage**, which revealed their predominant localization in the OSECs, PreGCs-I, and PreGCs-II region.

Line 266: "We observed a distinct stratified distribution pattern of these cells in E24 and E27."

Comment: Similar to the previous comment, please describe this pattern more clearly (where and how the stratification is observed).

Response: Thank you for your valuable feedback. We have revised the manuscript accordingly.

The modification is as follows (see details in lines 274-277 of the revised manuscript):

In E24 and E27, these cells exhibited a distinct stratified distribution pattern, with OSECs positioned in the outer cortical region, cluster 2 located medially adjacent to the OSECs, and cluster 3 located to the medullary region.

Line 271: "The results showed that GATA4 (a coelomic epithelial marker) and KRT19 (a marker for OSECs and Pre-GCs II cells)"

Comment: This statement seems to contradict the figures (or standard literature). Please check if the markers or cell types are swapped, or provide references that support this specific co-expression/identity.

Response: Thank you for your feedback. We observed that PreGCs-II is closer to OSEC in pseudotime analysis. However, further validation revealed that the expression level of KRT19 in PreGCs-II is relatively low, and immunofluorescence staining confirmed weaker

KRT19 expression in PreGCs-II compared to OSEC. Therefore, KRT19 serves as a specific marker for OSECs, rather than for PreGCs-II.

To more clearly illustrate this expression pattern, we have refined the color scheme of the Dot plot to enhance the visual contrast of KRT19 expression differences between the two groups. Additionally, we have revised the previous description regarding the transcriptional features of OSECs and PreGCs-II, with a focus on highlighting the differential expression of KRT19 in these two cell types, thereby improving the accuracy and logical rigor of the discussion (**Response Figure 1**).

The modification is as follows (see details in lines 253-256 of the revised manuscript):

PreGCs-I were characterized by high *WNT6* expression and the absence of *LHX9* and *KRT19*. In contrast, OSECs and PreGCs-II shared a transcriptional similarity but were distinguished by the lower expression of *LHX9* and *KRT19* in PreGCs-II.

The modification is as follows (see details in lines 281-283 of the revised manuscript):

The results showed that *GATA4* (a supporting cell lineage marker) and *KRT19* (a marker for OSECs) colocalized in the inner cortex of the fetal gonad, consistent with the spatial transcriptomics visualization.

Response Figure 1: Dot plot displays the average expression levels of representative genes in the supporting cell types in porcine female gonads. The average expression levels are obtained from the logarithmically scaled normalized counts of single-cell spatial transcriptomes. The percentage of cells expressing each gene is indicated by the size of the dots, and the color shade encodes the expression intensity (**as shown in Fig. 3 c**).

Line 279: "The results showed that the supporting cell lineages in female gonads of both pigs (E50) and humans (14 weeks) exhibit a highly conserved spatial distribution: OSECs

are restricted to the outer cortical region, Pre-GCs II cells are distributed throughout the inner cortex, and the less abundant Pre-GCs I population resides specifically within the medullary region"

Comment: It is difficult to distinguish these specific regions in the human gonad images provided. Please improve the annotation or visualization to clearly differentiate the cortical vs. medullary regions.

Response: Thank you for your thorough and professional review of the manuscript. In accordance with your suggestions, we have used clear dashed lines in high-magnification local views to delineate the boundaries of the "Outer Cortex," "Inner Cortex," and "Medulla," with detailed descriptions provided in the figure legend (**Response Figure 2**).

Response Figure 2: Spatial localization of three human supporting cell lineage subgroups at 14 gestational weeks. The dashed box indicates the enlarged view of the corresponding area. Scale bars: 500 µm (zoomed-out), 100 µm (zoomed-in) (as shown in Fig. 3 g).

Line 361: "Mitotic PGCs are predominantly surrounded by Pre-GCs I"

Comment (General Issue): The abbreviation "GC" is used confusingly for both Germ Cells and Granulosa Cells throughout the manuscript. For example, here "Pre-GCs" refers to granulosa, but elsewhere "GC" implies germ cells. The authors must strictly differentiate these abbreviations.

Response: Thank you for your valuable feedback.

Based on published literature related to human gonad development, we have standardized the terminology from "pre-granulosa cells I/II (Pre-GCs I/II)" to "pregranulosa cells-I/II

(PreGC-I/II)¹." In subsequent descriptions, we will avoid the standalone use of "GC" to minimize potential confusion for readers.

Reference:

1. Garcia-Alonso, L. *et al.* Single-cell roadmap of human gonadal development. *Nature* **607**, 540-547 (2022).

Line 389: "...which female germline cysts develop (Fig. 5e)"

Comment: This citation appears incorrect and likely refers to Fig. 5f. Additionally, Panel 5g is not mentioned in the text.

Response: Thank you for your valuable feedback. Based on your suggestions, we have thoroughly verified and corrected the relevant sections as well as the figure citations in the manuscript. In the revised version, the citation at this location has been updated to "(Fig. 5f)." Furthermore, we have checked the entire manuscript and added corresponding descriptions and citations for Fig. 5g in the Results section.

The modification is as follows (see details in lines 397-399 of the revised manuscript):

Notably, immunofluorescence staining at E50 showed that LAMA1 completely encapsulated numerous FGCs, forming an enclosure within which female germline cysts develop (Fig. 5f).

The modification is as follows (see details in lines 402-405 of the revised manuscript):

In summary, these results indicate that the supporting cell lineages and interstitial cells interact with germ cells in a spatial context, suggesting that this cell population may play a more critical role during the specification of PGCs into oogonia (Fig. 5g).

Supplementary Material: "Fig. 2g"

Comment: The text refers to a Figure 2g, but this panel appears to be missing.

Response: Thank you for the reviewer's keen observation and valuable feedback. You are absolutely correct. Upon checking, we found that this was due to a clerical error—the "Fig. 2g" mentioned in the text should be "Fig. 2f." We sincerely apologize for any confusion this oversight may have caused during the review.

The modification is as follows (see details in lines 181-185 of the revised manuscript):

Alongside low 5mC and 5hmC levels, DNA methylation 3A (*DNMT3A*), *TET1* and *TET3* expression declined, while the base excision repair (BER) pathway was upregulated,

suggesting that active DNA demethylation had been completed, yet epigenetic reprogramming seemed to persist via the BER pathway (**Supplementary Fig. 2f** and **Supplementary Table3**).

Figure 6c (Upper):

Comment: In the merged panel, the purple and green channels appear to be swapped compared to the individual single-channel panels. Please verify.

Response: Thank you for your valuable feedback. We have re-examined the original image data in accordance with your suggestions and corrected the channel merging settings. This ensures full consistency between the individual channel images and the color indicators in the merged image. Once again, we sincerely thank you for helping us rectify this error and improve the accuracy and clarity of the figures (**Response Figure 3**).

Response Figure 3: Immunofluorescence images of DDX4 (green) and CTNNB1 (purple) in porcine female gonadal sections from different timepoints. Nuclei were counterstained with DAPI (blue), Scale bar: 20 μm (as shown in **Fig. 6 c**).

Reviewer #2 (Remarks to the Author):

Dear authors,

The manuscript has gained in quality and clarity.

I now support publication.

Response: We sincerely thank you for reviewing the manuscript titled " A spatiotemporal transcriptomic atlas of porcine female early gonadal development" and for the valuable feedback you provided.

We are very pleased to learn that the revised manuscript has met with your approval in terms of quality and clarity, and that it has gained your support for publication. The insightful comments and constructive suggestions you offered in the previous round of review were crucial to refining this research work.

We wish to reiterate our deepest gratitude for the time and expertise you have dedicated.

Your contribution has significantly enhanced the value of this manuscript.

Reviewer #3 (Remarks to the Author):

The authors have addressed the comments raised by this reviewer, making the current version of the manuscript is easier to follow. Only minor concerns remain as follow.

Response: Thank you very much for your positive feedback on our revised manuscript. We are pleased to note that the current version has achieved greater clarity and readability. Regarding the remaining minor issues you highlighted, we will make further refinements as suggested.

1. Line 180-181, is “DNA methylation (DNMT3A)” not typographical error for “DNA methyltransferase 3a (DNMT3A)”?

Response: Thank you for your careful review of our manuscript and for your valuable comments. The point you raised regarding the inaccurate expression “DNA methylation (DNMT3A)” is absolutely correct. We have revised the text accordingly and changed it to “DNA methyltransferase 3A (DNMT3A)” as suggested.

The modification is as follows (see details in lines 181-185 of the revised manuscript):

Alongside low 5mC and 5hmC levels, **DNA methylation 3A (DNMT3A)**, *TET1* and *TET3* expression declined, while the base excision repair (BER) pathway was upregulated, suggesting that active DNA demethylation had been completed, yet epigenetic reprogramming seemed to persisted via the BER pathway.

2. Line 413-415, the authors cannot not definitively state that these cells are RA oogonia based on these immunofluorescence images. Therefore, this reviewer suggests that the authors revise this point to refer to the embryonic day in order to ensure accuracy.

Response: Thank you for your careful review of our manuscript and your valuable feedback. We fully agree with your perspective. Relying solely on immunofluorescence images indeed cannot provide conclusive evidence that these cells are “RA oogonia.” Using the embryonic developmental age (E35) for description is a more objective and accurate approach. In the relevant section of the manuscript, we have revised “RA oogonia” to “E35 oogonia.” This modification ensures greater rigor in the description of our conclusions, and we sincerely appreciate your suggestion.

The modification is as follows (see details in lines 420-425 of the revised manuscript):

To assess canonical WNT pathway activation during the PGC-to-oogonia transition, we examined CTNNB1 localization. High levels of membrane-associated CTNNB1 were detected in somatic cells and FGCs of gonads at various developmental stages. However, weak nuclear CTNNB1 was detected exclusively in E35 oogonia, suggesting that the canonical WNT signaling mechanism may be transiently active only in E35 Oogonia.

Reviewer #4 (Remarks to the Author):

The authors have done a very thorough job revising the manuscript and incorporating all of the reviewer comments. I have no further comments, the authors have addressed my concerns to satisfaction.

Response: We sincerely thank you for reviewing the manuscript titled "A spatiotemporal transcriptomic atlas of porcine female early gonadal development" and for the valuable feedback you provided. We are pleased to know that the revised version is to your satisfaction, and we are glad that the thorough revisions made based on your and the reviewers' comments have proven fruitful. Once again, we sincerely thank you for your time, effort, and the highly constructive guidance provided throughout the review process.